# Dissolved oxygen as indicator of multiple drivers of the marine ecosystem: the Southern Adriatic Sea case study

Valeria Di Biagio[1], Riccardo Martellucci[1], Milena Menna[1], Anna Teruzzi[1], Carolina Amadio[1], Elena Mauri[1], Gianpiero Cossarini[1]

[1]National Institute of Oceanography and Applied Geophysics - OGS, Trieste, Italy

*Correspondence to*: Valeria Di Biagio (vdibiagio@ogs.it)

**Abstract.** Oxygen is essential to all aerobic organisms and its dynamics in the ocean involve interconnected physical and biological processes that are at the basis for the marine ecosystem functioning. The study of dissolved oxygen (DO) variations under multiple drivers is currently one of the main goals of climate and marine ecological scientific communities, and the quantification of DO levels is essential for the assessment of the environmental status, especially in coastal areas.

We investigate the 1999-2021 interannual variability of DO in the southern Adriatic Sea, a marginal area of the Mediterranean Sea, where deep water formation processes occur, contributing significantly to the ventilation of the Eastern Mediterranean basin. Following the Marine Strategy Framework Directive, which promotes the integration of different observational platforms, we use DO modelled by the Copernicus Marine Mediterranean Sea biogeochemical reanalysis, which assimilates satellite chlorophyll concentrations and to which we apply a bias correction using DO Argo float measurements in 2014-2020. A correlation analysis of the time series of the first three modes of variability (86% of the total variance) of the DO profiles extracted from the bias-corrected reanalysis with key meteo-marine indicators shows a link with (i) net heat fluxes related to oxygen solubility, (ii) vertical mixing, (iii) biological production at the surface and in subsurface layers, and (iv) circulation associated with the entrance of northern Adriatic waters. The alternating entrance of Levantine and Atlantic Waters through the North Ionian Gyre (NIG) appears to be the driver of the fourth mode of variability, which explains 8% of the total variance. Moreover, we find that the first temporal mode of variability is the main driver of the negative anomaly of DO in the 0-600 m layer in 2021 with respect to the 1999-2020 climatology. We ascribe the lower content of DO in 2021 to a negative anomaly of the subsurface biological production in the same year, in agreement with the previous correlation analysis, but not to heat fluxes. Indeed, in agreement with previous studies, we observe a sharp increase in salinity favoured by the cyclonic circulation of NIG from 2019 onwards. We interpret this as a possible regime shift that is not captured by the time series analysis, and whose possible consequences for Ionian-Adriatic system ventilation and for marine organisms should be carefully monitored in the near future.

# 1 Introduction

Dissolved oxygen (DO) is a key indicator for monitoring the marine ecosystem functioning, because it is the result of several atmospheric, hydrodynamic and biogeochemical driving processes (such as air-sea fluxes, vertical convection and mixing, horizontal transport, biological production and consumption; Keeling and Garcia, 2002; Oschlies et al, 2018; Pitcher et al., 2021). Indeed, DO is currently being studied under the global warming scenarios by climate and marine ecological scientific communities (e.g. Pörtner et al., 2019; Kwiatkowski et al., 2020; Garcia-Soto et al., 2021), as oxygen depletion has been observed in the global ocean as well as at local scale (Breitburg et al., 2018). Climate models predict a reduction in global ocean dissolved oxygen content (Matear et al., 2000; Oschlies et al., 2008; Stramma et al., 2010, Reale et al., 2021), so this parameter is of primary interest especially in those areas where oceanic processes connect surface and deep layers.

The southern Adriatic Sea (SAdr, Fig. 1a) is one of these areas, as it is an area of deep water formation (Gačić et al., 2002; Pirro et al., 2022) and represents the deep engine of the eastern Mediterranean thermohaline circulation (Malanotte-Rizzoli et al., 1999), which is crucial for the eastern basin ventilation. The Adriatic Sea (Fig. 1a) is an elongated, semi enclosed and roughly north-south oriented basin characterized by a shallow northern shelf (shallower than 80 m) and a deep pit in its southern part (maximum depth of approximately 1200 m) which is connected to the Ionian Sea (central Mediterranean basin) through the Otranto Strait (with a maximum depth of 800 m). The Adriatic Sea is characterized by a cyclonic circulation governed by several drivers: river runoff, wind stress, surface buoyancy fluxes and mass exchanges through the Otranto Strait (Cushman-Roisin et al., 2013).

The SAdr is strongly influenced by the inflow of water masses from the northern Adriatic Sea (i.e., North Adriatic Dense Water, Querin et al., 2016) and the Ionian Sea. In particular, the inflow of southern water masses is triggered by the periodic reversal of Northern Ionian Gyre circulation (Gačić et al., 2002; Civitarese et al., 2010, Menna et al., 2019). This oscillating system, called the Adriatic - Ionian Bimodal Oscillating System (BiOS), changes the circulation of the Northern Ionian Gyre from cyclonic to anticyclonic and vice versa, modulating the advection of water masses in the Adriatic Sea (Gačić et al., 2010, Rubino et al., 2020). The cyclonic circulation of the Northern Ionian Gyre causes the advection of saline water masses of Levantine origin (i.e., Levantine Intermediate Water, Cretan Intermediate Water, Ionian Surface Water and Levantine Surface Water, Manca et al., 2006), while the anticyclonic circulation favours the inflow of Atlantic Water and a relative decrease of salinity in the SAdr (Gačić et al., 2011, Menna et al., 2022a). This feature has a strong influence on the biogeochemical properties of the SAdr, affecting nutrient availability (Civitarese et al., 2010), phytoplankton blooms (Gačić et al., 2002; Civitarese et al., 2010), and species composition (Batistić et al., 2014, Mauri et al., 2021).

While hydrodynamic and biogeochemical properties of SAdr have been widely described in several studies (e.g., Civitarese et al. 2010; Cushman-Roisin et al., 2013; Lipizer et al., 2014; Kokkini et al., 2018, 2019; Mavropoulou et al., 2020; Mihanović et al., 2021; Menna et al., 2022b), at the best of our knowledge DO dynamics in the area in connection with relevant driving processes over decadal time scales have not been addressed yet.

Investigating the DO multidecadal variability is crucial for quantifying the state of the marine environment (Marine Strategy
Framework Directive, MSFD; Oesterwind et al., 2016) and for understanding anthropogenic impacts on the marine
environment (Pörtner et al., 2022). The emerging ecosystem-based management method proposed by the MSFD (2008/56/EC)
promotes the use of different observational platforms, allowing to synoptically collect information on the space-time
distribution of important parameters related to water quality (Martellucci et al., 2021).
In this context, the present work integrates the state-of-the-art approach of in *situ* measurements (in 2014-2020, distributed by
Copernicus In Situ TAC) with the Copernicus biogeochemical reanalysis in the Mediterranean Sea at high resolution (Cossarini
et al., 2021), with the aim of characterizing the DO dynamics in the SAdr in the 1999-2021 time period. In particular, we aim
to assess DO inter-annual variability in an area (SAdr) sensitive to multiple drivers (e.g., atmospheric forcing, Mediterranean
circulation, and biological processes) and to evaluate the relative importance of the different drivers in this area.

## 2 Data and methods

In the present study the DO concentration in the SAdr area (Fig. 1a) was assessed by combining data from the Copernicus
reanalysis in the Mediterranean Sea (prod. ref. no. 1 in Table 1; Cossarini et al., 2021) in 1999-2021 and the Copernicus in situ
dataset (prod. ref. no. 2, https://doi.org/10.13155/75807), available for the period 2014-2020 (Figs. 1b-c). The temporal
evolution of the combined model-in *situ* DO concentration profile in 1999-2021 time period is shown in Fig. 1d.
In particular, we used the BGC-Argo float measurements of in *situ* DO to compute a bias correction to the daily DO
concentrations simulated by the biogeochemical reanalysis at 1/24° horizontal resolution. In fact, the biogeochemical
reanalysis does not include BGC-Argo float DO assimilation and displays an average RMSD of 15 mmol m$^{-3}$ for DO in the 0-
600 m depth layer with respect to the observations in the area (Cossarini et al., 2021, Teruzzi et al., 2022). Quantile Mapping,
a technique largely used for climate simulations (e.g., Hopson and Webster, 2010; Themeßl et al., 2011; Gudmundsson et al.,
2012), was adopted to perform the reanalysis bias correction. The Quantile Mapping technique adjusts the cumulative
distribution of the data simulated for the past or future period by applying a transformation between the quantiles of the
simulated and observed data in the present. In our application, we adapted the code publicly provided by Beyer et al. (2020)
at https://doi.org/10.17605/OSF.IO/8AXW9 and included available in *situ* data of daily DO (Fig. 1c) within a representative
area (Fig. 1b) of the southern Adriatic in the period 2014-2020, and DO reanalysis data for the same days of measurements.
The representative area was identified by applying a spatial cross-correlation analysis (Martellucci et al., 2021) to the
biogeochemical reanalysis centered on the SAdr pit and selecting the correlation threshold of 0.9 (Fig. 1b). Specifically, we
considered the cross-correlation between the surface data of DO, nitrate and chlorophyll concentrations in the central point of
the pit and those at each spatial grid point in the domain, to identify the area that displayed the same dynamics at the surface
from a phenomenological perspective. Further details on the Quantile Mapping bias correction are included in Appendix A.
We then applied the Empirical Orthogonal Function (EOF) analysis (e.g. Thomson and Emery, 2014) to the vertical profiles
in Fig. 1d to describe DO variability in the SAdr area in the period 1999-2021. The EOF analysis allows us to identify the
spatial patterns of variability (i.e., EOF vertical patterns), describe how they change in time by means of time series (i.e.,EOF
time series), and associate the explained variance with each mode.
Finally, we performed a Pearson correlation analysis between the EOF time series in 1999-2021 and the following series of
forcing indexes (reported in Fig. 2) providing evidence of the mechanisms driving oxygen concentration and dynamics in the
area:
- heat fluxes in the SAdr as a proxy for thermal and mixing and stratification cycles (from prod. ref. no. 6 in Table 1; Fig. 2a);
- mixed layer depth in the SAdr as a proxy for both local vertical mixing and water residence times in the pit (prod. ref. no. 3;
Fig. 2b);
- chlorophyll concentration at surface and in subsurface in the SAdr as a proxy for biological production in spring and late
spring-summer, respectively (prod. ref. no. 1; Figs. 2c-d);
- heat fluxes in the northern Adriatic Sea (NAdr), as a proxy for dense water oxygen-rich formation in the NAdr and its
transport into the pit (prod. ref. no. 6; Fig. 2e);
- Northern Ionian Gyre (NIG) vorticity derived from satellite altimetry, as a proxy of the inflow of Levantine waters and
Atlantic Water (AW) (prod. ref. no 4 and 5; Fig. 2f).
In particular, the temporal phases of the NIG are defined as cyclonic and anticyclonic, respectively, when the vorticity field is
positive and negative, as highlighted by the de-seasonalized time series in Fig. 2f.
Mixed layer depth (computed in prod. ref. no. 3 considering the 0.03 kg m$^{-3}$ density difference with respect to the near-surface
value at 10 m depth) and the chlorophyll at surface and in subsurface (30-80 m, where the deep chlorophyll maximum is
located) were spatially averaged in the SADr area (41.6°-42.1°N; 17.6°-18.1°E, to consider the whole volume of the pit); heat
fluxes were calculated in both the SAdr area and in the NAdr area (44.5°-45.5°N; 13°-13.5°E), while current vorticity was
computed in the Northern Ionian Sea (37°–39°N; 17–19.5°E).
In the correlation analysis, the time series of the heat fluxes in NAdr Sea (Fig. 2e) has been temporally lagged by 2 months, as
an estimated mean time for the entrance in the SAdr pit of waters originated in the Northern Adriatic area (Vilibić et al., 2013;
Querin et al., 2016; Mihanović et al 2021). Moreover, we tested the significance of the correlation coefficients between EOF
and driver time series using a parametric t-test (with a reference significance level of 0.05).
**3 Results**
**3.1 Temporal scales of variability in connection with drivers**
Dissolved oxygen in the southern Adriatic area (Fig. 1a) shows in the subsurface layers an alternation between periods of
enrichment (in 2004-2006, 2010-2013, 2016-2017) and sharp declines that impacted the Oxygen Minimum Layer (OML),
located between 100 and 300 m. Low concentration values are observed also in the years between 1999 and 2003.
The EOF analysis was performed on the vertical profiles of the oxygen anomaly, derived by removing the mean profile in the
period 1999-2021, and then normalized dividing by their standard deviation.
The time series of the first four EOF modes, which explain up to 95% of the oxygen variability in the water column are shown
along with the corresponding vertical patterns in Figs. 3a,c,e,g and Figs. 3b,d,f,h, respectively.
The EOFs are interpreted considering the correlation of the EOF time series (Figs. 3a,c,e,g) with the time series of the forcing
indicators shown in Fig. 2 (see Table 2), with heat fluxes in the northern Adriatic Sea time-lagged by two months as the
estimated time of entry of the NAdr dense water in the SAdr pit.
The first mode (Figs. 3a-b), accounting for 48.9% of the explained variance, can be associated with the seasonal cycle of
oxygen concentration in the upper layers: its vertical pattern mainly affects the first levels (Fig. 3b), the corresponding time
series shows relative maximum values in spring (Fig. 3a) and it shows a statistically significant but moderate correlation
(r=0.56) with heat flux and a lower correlation with the subsurface chlorophyll concentration (r=0.43) in the SAdr area (first
column in Table 2).
The second and third modes (Figs. 3c-d and 3e-f, respectively), describing 19.7% and 17.7% of the variance respectively,
affected both the upper and deeper layers. Both modes display relative maximum values in summer, but they have different
correlation coefficients with the explanatory factors. The time series of the second mode (second column in Table 2) shows a
significant but low correlation with multiple drivers, exceeding 0.4 only for surface chlorophyll (r=-0.41) and waters from the
NAdr area (r=0.48). The time series of the third mode (third column, same Table) is moderately correlated with both surface
chlorophyll (r=-0.61) and NAdr waters (r=0.68 correlation), but also with heat fluxes in the area (r=0.51), and, to a lower
extent, with mixed layer depth (r=-0.41) and subsurface chlorophyll (r=0.48).
The fourth mode (Figs. 3g-h), that describes 8% of the variance, can be ascribed mainly to the vorticity of the NIG (r=-0.37,
last column in Table 2), which affects the oxygen concentration in the intermediate layer (100-500 m depth), filled by LIW,
and acts in the opposite direction in the upper and deeper layers (Fig. 3h).
Analysing the four modes in order of decreasing explained variance, we ascribe the seasonal variability connected with
solubility mainly to the first mode, whereas we associate the biological contribution to oxygen dynamics to multiple interacting
modes. In fact, the first mode explains the onset of the subsurface oxygen maximum (SOM) in spring, while the summer
dynamics of the SOM is partially related to the third mode. The second mode, whose time series is correlated with surface
chlorophyll evolution among other factors, can explain that part of the oxygen variability that is related to winter surface
productivity.
The SOM, evident in summer oxygen profiles in Figs. 1c-d at about 40 m depth, is a feature that has already been observed in
a great part of oligotrophic oceans (Riser and Johnson, 2008; Yasunaka et al., 2022) and of the Mediterranean Sea (e.g., Kress
and Herut, 2001; Copin-Montégut and Bégovic, 2002; Manca et al., 2004; Cossarini et al., 2021; Di Biagio et al., 2022) and it
represents an emerging property resulting from multiple interacting ecosystem processes (i.e., air-sea interactions, transport,
mixing and biological production and consumption) and is, indeed, captured by multiple modes.
The third mode, which also describes the high concentration values in the deep layers in the period 2005-2006 and 2012-2014,
is also moderately associated with a multi-annual signal of the inflow of deep denser and oxygenated water from the northern
Adriatic Sea (r=0.68, third column in Table 2; Querin et al., 2016). Finally, it is worth noting that an EOF analysis of detrended
DO time series (not shown) yields fairly similar results but with the third mode only weakly correlated with the forcing indexes
(r<0.4). Indeed, we can conclude that the third mode captures a signal of long-term evolution of oxygen concentration
associated with changes in heat fluxes and chlorophyll concentration.

**3.2 The 2021 anomaly**

The year 2021 shows an overall negative anomaly in the oxygen concentration profile (Fig. 4b) compared to the 1999-2020
climatological  profiles (Fig. 4a). In particular, the anomaly affects a layer that thinned during the year, moving from 0-600 m
depth in winter-early spring season to 30-400 m in late spring-summer and 0-80 m in fall. The (absolute) maximum values
correspond to 25-30 mmol m$^{-3}$ at the surface in spring and at the SOM depth in summer.
We verified that, among the EOF modes, the negative anomaly of the first mode is the main contributor to the 2021 negative
oxygen anomaly (not shown). The time series of the first mode (Fig. 3a) is actually negative from 2019 and corresponds to
the negative anomaly of only one of its drivers (Table 2), i.e. subsurface chlorophyll (Fig. 2d), and not heat fluxes (Fig. 2a).
In particular, we estimated a mean negative anomaly approximately equal to 6% with respect to the climatological mean (1999-
2020) for subsurface chlorophyll in 2021.
One of the causes of the decrease in total oxygen concentration in the SAdr could be due to the exceptional salinization
observed in the SAdr since 2017 (Mihanović et al., 2021, Menna et al., 2022b). This increase was related to the inflow of new,
warmer and noticeably saltier water masses from the northeastern Ionian Sea (Mihanović et al., 2021, Menna et al., 2022b).
The inflow of saltier and warmer water masses is also evident by observing the temporal evolution of these parameters through
the Strait of Otranto (Fig. B1). In particular, in the upper layer (0-150 m) both temperature and salinity show an overall positive
trend throughout the period 1999-2021, whereas the decrease observed in 2006-2011 and 2017-2018 can be associated with
the inflow of less saline AW, triggered by the anticyclonic circulation of the NIG (Fig. 2f). In the intermediate layer (150-600
m), salinity shows a positive trend in 1999-2021, while no clear trend is observed for temperature. Moreover, a sharp increase
in salinity (~ 0.1) is observed in 2019. This increase occurred after the NIG inversion from anticyclonic to cyclonic (Fig. 2f),
resulting in a further increase in salinity due to both the decrease in AW advection and the increase in LIW inflow.

**4 Conclusions**

Merging the Copernicus biogeochemical reanalysis in the Mediterranean Sea with in *situ* TAC data of biogeochemical Argo
floats allowed us to characterize the interannual variability of dissolved oxygen in the southern Adriatic Sea in the 1999-2021
time period and the 2021 anomaly with respect to the mean over 1999-2020. This study enriches our knowledge of the dissolved
oxygen state and long-term dynamics in the area, by proposing a seamless time and space perspective that is complementary
to previous climatologies and data aggregation information (e.g. Lipizer et al., 2014) and adding an explanatory framework
for the driving mechanisms in the marine environment.

The EOF statistical analysis that we conducted on the vertical oxygen profiles yielded two key results. First, in contrast with a climatological view, the analysis was able to capture most of the inter-annual oxygen variability associated with variability of the main drivers (i.e., heat fluxes affecting solubility; biological productivity; vertical mixing). We do not detect a clear deoxygenation trend in the subsurface layer, while the multiannual variability is characterised by an alternation of enrichment and reduction phases, whose dominant correlations with the drivers for each EOF time series are in the (absolute) range 0.40-0.70. The possibility to observe such cyclic signals is enhanced by the relatively small volume and short residence time of the SAdr pit waters (Querin et al., 2016) with respect to other Mediterranean areas (Coppola et al., 2018). This feature makes the SAdr a potential efficient probe to detect a rapid response to changes in its meteo-marine drivers, i.e., circulation and atmospheric patterns.

Indeed, as our second result, the variability that is not explained by the EOF decomposition appears to be connected with a possible regime shift, associated with the entrance of new water masses, warmer, markedly saltier and less oxygenated, that were not previously observed in the analyzed time period.

The exceptional increase in salinity occurring after 2019 has been already documented (Mihanović et al., 2021; Menna et al., 2022b) and also observed north of the SAdr pit. Further monitoring of such anomalously high salinity values and assessment of their potential impact on the marine food web is of great importance, as picoplankton groups are sensitive to this environmental variable (Mella-Flores et al., 2011) and changes in biomass and production due to salinity have already been observed in previous studies in the Adriatic Sea (Beg Paklar et al., 2020; Mauri et al., 2021). Moreover, if such a strong negative oxygen anomaly as observed in 2021 were to persist, it could have direct impacts on local marine organisms, as well as on the cycling of dissolved chemical elements (Conley et al., 2009) potentially altering the energy flux towards the higher trophic levels (Ekau et al., 2010). The importance of the relationship between dissolved oxygen and the catch distribution of some marine species has already been proved in the Adriatic Sea (Chiarini et al., 2022).

By integrating model and in *situ* data, our study demonstrates the importance of following up the oxygen content in a seamless spatial and temporal way, as it is a fundamental indicator of good environmental status (GES, Oesterwind et al., 2016) and a factor that significantly affects fishing activities and economy.

**Appendix A: Quantile Mapping bias correction of DO concentration profiles**

Figures A1 and A2 show the modelled DO concentration profiles and histogram distributions before and after the Quantile Mapping bias correction, respectively, conducted by using the BGC-Argo float measurements available in 2014-2020 (Fig. 1c). The Quantile Mapping, better than other methods (i.e. Additive Delta Change, Multiplicative Delta Change and Variance Scaling; results not shown), acted on the profiles by modifying the values of the concentrations (as indicated by the different colorbars in Figs. A1a and A1b) but, at the same time, maintaining the main dynamics observed before the correction: mixing and stratification at the surface during the year, subsurface oxygen maximum onset in spring and development in summer, and interannual variability related to the mixed layer depth dynamics in the intermediate layers. The distributions of the values of the model output before and after the Quantile Mapping bias correction and the values from BGC-Argo floats are displayed in

Fig. A2. The correction actually changed the modelled values (Fig. A2a) to reproduce the shape of the distribution of the observations (Fig. A2c). In particular, after the correction (Fig. A2b) the modelled data show higher variability and a more skewed distribution toward the higher values, similarly to the observations.

**Appendix B: Time series of surface and intermediate temperature and salinity at the Otranto Strait**

**Data availability**

Publicly available datasets were analyzed in this study. Modelling and in *situ* data can be found at the Copernicus Marine Service, with references and DOIs indicated in the Table 1 of the manuscript.

**Author contribution**

VDB and GC conceived the idea. VDB, RM and MM conducted the analysis. VDB, RM and GC wrote the first draft, with contributions from the other co-authors. All the authors discussed and reviewed the submitted manuscript.

**Competing interests**

The authors declare that they have no conflict of interest.

**Acknowledgements**

This study has been conducted using EU Copernicus Marine Service Information.

**Financial support**

This study has been partly funded by the Mediterranean Copernicus Monitoring and Forecast Center (contract LOT REFERENCE: 21002L5-COP-MFC MED-5500  issued by Mercator Ocean) within the framework of Marine Copernicus Service.

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

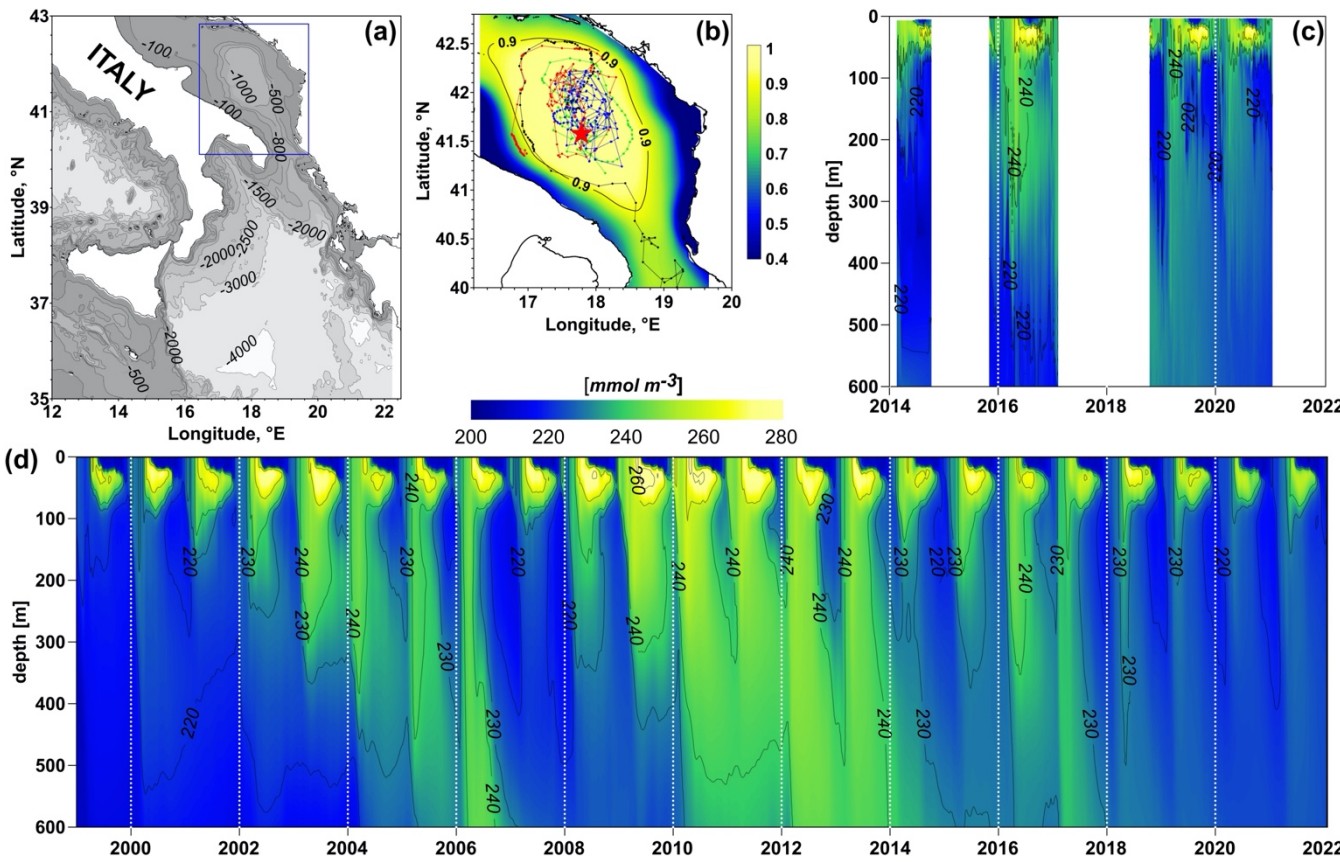


Figure 1: (a) Southern Adriatic area (blue square) within Mediterranean Sea;  (b) Cross-correlation map of surface oxygen, nitrate and chlorophyll concentration provided by Copernicus biogeochemical reanalysis (prod. ref. no. 1, Table 1) in the southern Adriatic area with respect to the central point of the pit indicated by the red star; the black contour line delimits the area with cross-correlation equal or higher than 0.9; dashed lines indicate the trajectories of BGC-Argo floats (In Situ TAC data, prod. ref. no. 2) passing the area. (c) Hovmöller diagrams of the dissolved oxygen concentration from In Situ TAC data (prod. ref. no. 2) within the 0.9 cross-correlation area (panel (b)). Data have been interpolated for readability of the plot. (d) Hovmöller diagrams of dissolved oxygen concentration from Copernicus biogeochemical reanalysis (prod. ref. no. 1), spatially averaged within the area of cross-correlation equal to 0.9 (panel (a)) in 1999-2021 time period, after the bias correction procedure based on In Situ TAC data (prod. ref. no. 2).

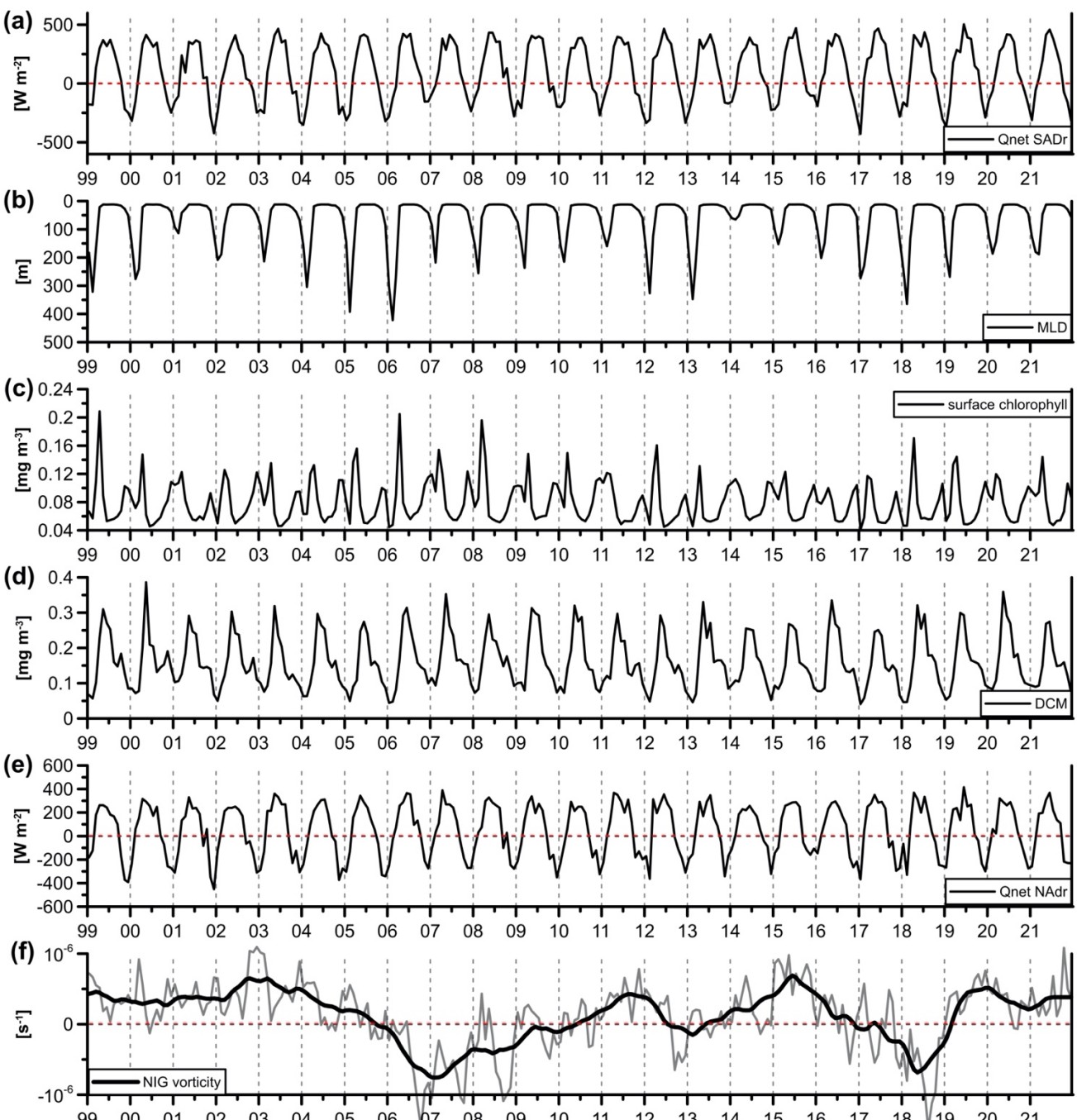

Figure 2: Time series of the main forcing in the 1999-2021 time period: (a) net heat fluxes in SAdr (prod. ref. no. 6 in Table 1), (b) mixed layer depth (prod. ref. no. 3), (c) surface chlorophyll concentration (prod. ref. no. 1), (d) subsurface chlorophyll concentration (30-80 m layer in which deep chlorophyll maximum (DCM) is located, prod. ref. no. 1), (e) net heat fluxes in NAdr (prod. ref. no. 6), (f) NIG current vorticity (gray line) and de-seasonalized time series as obtained by applying a low-pass filter of 13 months (black thick line) (prod. ref. no. 4 and 5).

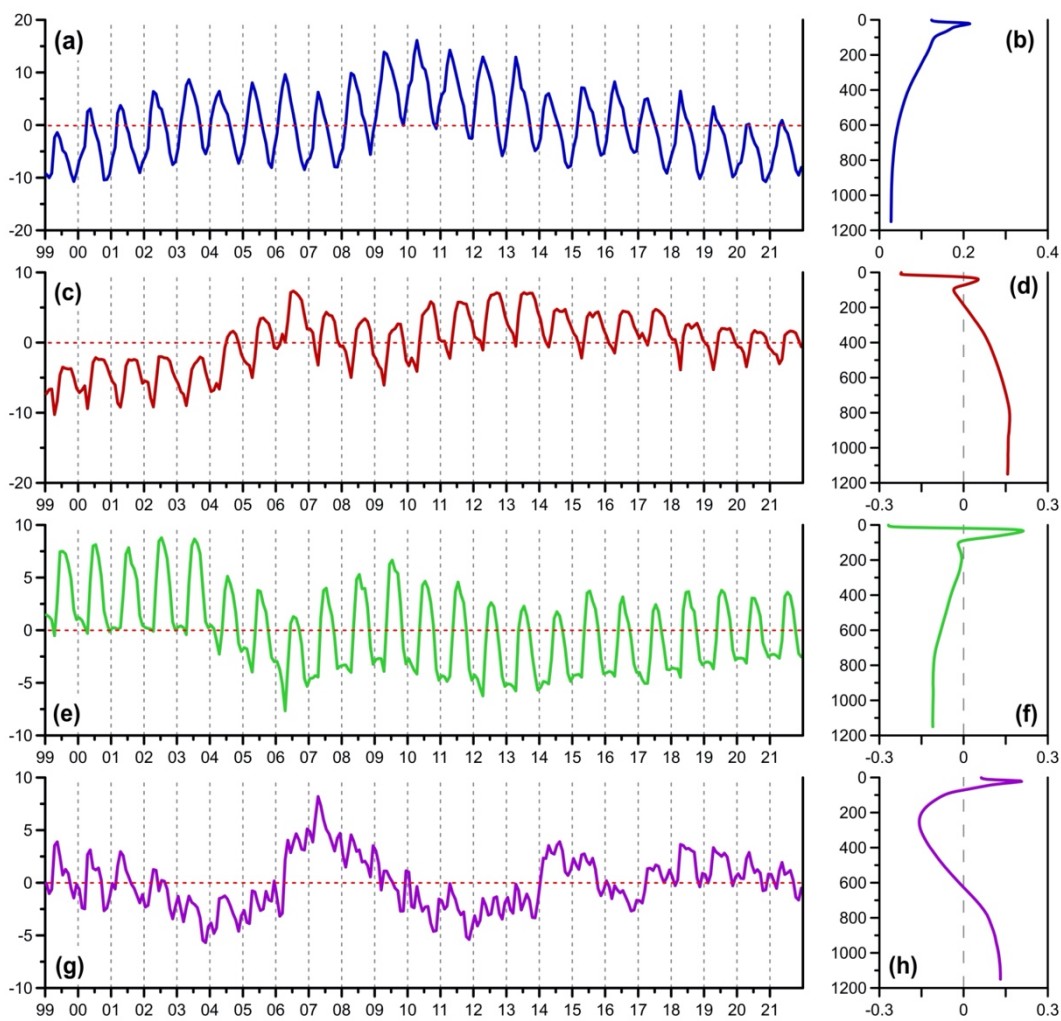

**Figure 3: EOF time series (a, c, e, g) and vertical patterns (b, d, f, h) of the first four modes computed on the bias-corrected dissolved**
**oxygen concentration in the southern Adriatic area shown in Fig. 1d.  The explained variances of the four modes are: 48.9%, 19.7%,**
**17.7% and 8.4%.**

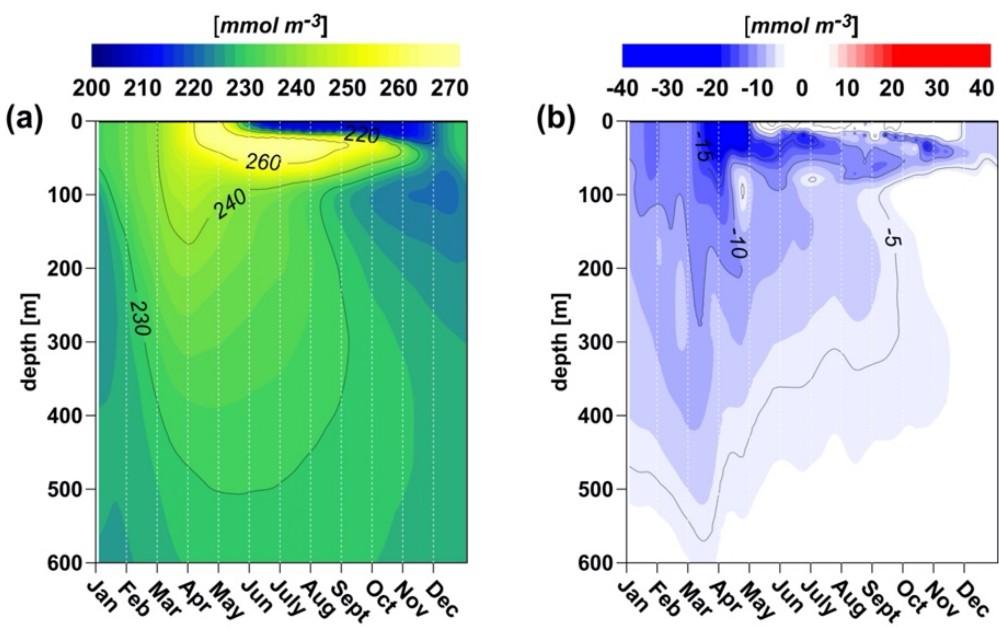



Figure 4: Hovmöller diagrams of: mean over 1999-2020 of daily oxygen concentration computed from Copernicus biogeochemical reanalysis (prod. ref. no. 1, Table 1) after the bias correction procedure based on In Situ TAC data (prod. ref. no. 2) (a) and anomaly in 2021 with respect to the 1999-2020 period (b).



| Prod. ref. no. | Product ID & type | Data access | Documentation |
|---|---|---|---|
| 1 | MEDSEA_MULTIYEAR_BGC_006_008 Mediterranean Sea Biogeochemistry Reanalysis | https://doi.org/10.25423/CMCC/MEDSEA_MULTIYEAR_BGC_006_008_MEDBFM3 (EU Copernicus Marine Service Product, 2022a)  https://doi.org/10.25423/CMCC/MEDSEA_MULTIYEAR_BGC_006_008_MEDBFM3I | Quality Information Document (QUID): Teruzzi et al., (2022) Product User Manual (PUM): Lecci et al., (2022a) Cossarini et al., (2021) |

| | | (EU Copernicus Marine Service Product, 2022b) | |
|---|---|---|---|
| **2** | INSITU_MED_PHYBGCWAV_ DISCRETE_MYNRT_013_035 Mediterranean Sea-In-Situ Near Real Time Observations | https://doi.org/10.48670/moi-00044 (EU Copernicus Marine Service Product, 2022c) | Quality Information Document (QUID): Wehde et al., (2022) Product User Manual (PUM): In Situ TAC partners (2022) |
| **3** | MEDSEA_MULTIYEAR_PHY_ 006_004 Mediterranean Sea Physics reanalysis | https://doi.org/10.25423/CMCC/MEDSE A_MULTIYEAR_PHY_006_004_E3R1 (EU Copernicus Marine Service Product, 2022d) https://doi.org/10.25423/CMCC/MEDSE A_MULTIYEAR_PHY_006_004_E3R1 I (EU Copernicus Marine Service Product, 2022e) | Quality Information Document (QUID): Escudier al., (2022) Product User Manual (PUM): Lecci et al., (2022b) Escudier et al., (2021) |
| 4 | SEALEVEL_EUR_PHY_L4_MY _008_068 European Seas Gridded L 4 Sea Surface Heights And Derived Variables Reprocessed 1993 Ongoing | https://doi.org/10.48670/moi-00141 (EU Copernicus Marine Service Product, 2022f) | Quality Information Document (QUID): Pujol al., (2022a) Product User Manual (PUM): Pujol et al., (2022b) |
| 5 | SEALEVEL_EUR_PHY_L4_NR T_OBSERVATIONS_008_060 | https://doi.org/10.48670/moi-00142 | Quality Information Document (QUID): Pujol al., (2022c) |

| | mode 1 | mode 2 | mode 3 | mode 4 |
|---|---|---|---|---|
| **HFlux (SAdr)** | 0.56 | 0.15 | 0.51 | 0.32 |
| **MLD (SAdr)** | n.s. | -0.28 | -0.41 | -0.25 |
| **surf chl (SAdr)** | n.s. | -0.41 | -0.61 | n.s. |
| **subsurface chl (SAdr)** | 0.43 | 0.13 | 0.48 | 0.34 |
| **Hflux NAdr (2-months lagged)** | n.s. | 0.48 | 0.68 | 0.16 |
| **NIG vorticity (NIon)** | n.s | -0.40 | n.s. | -0.37 |

Table 2: Correlations between the first four temporal modes of EOFs of DO (Figs. 3a,c,e,g) and the forcing fields (Fig. 2, with heat fluxes in the northern Adriatic Sea time-lagged by two months). Not statistically significant correlations are identified by a significance level higher than 0.05 and indicated by "n.s." acronym in the table.

| 6 | European Seas Gridded L 4 Sea Surface Heights And Derived Variables Nrt | (EU Copernicus Marine Service Product, 2022g) | Product User Manual (PUM): Pujol et al., (2022d) |
|---|---|---|---|
| | ERA5 hourly data on single levels from 1940 to present Global climate and weather reanalysis | https://doi.org/10.24381/cds.adbb2d47 (EU Copernicus Climate Change Service Product, 2022) | Hersbach et al., (2018) |

Table 1: Products used in the present work. Prod. ref. no. 3 is a forcing for prod. ref. no. 1 and prod. ref. no. 6 is a forcing for prod. ref. no. 3. Complete references for prod. ref. no 1, 3 and 6 are reported in the bibliography.

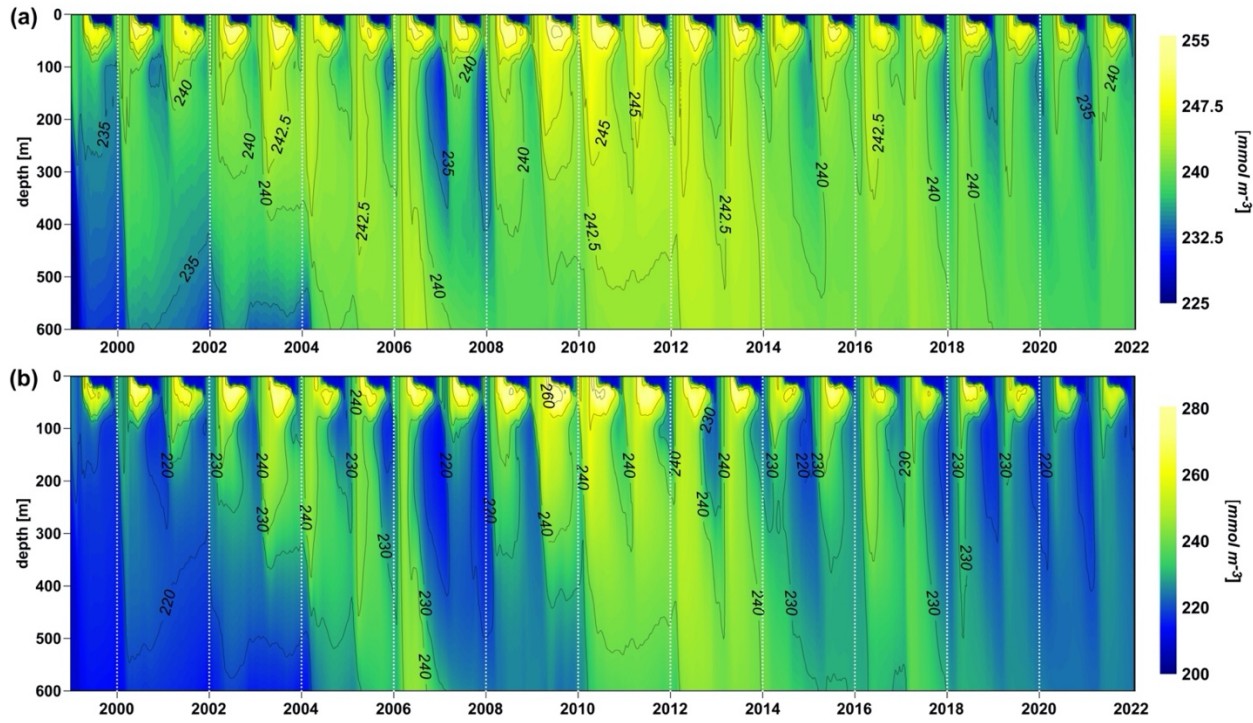

550

Figure A1: Hovmöller diagram of the modelled oxygen concentrations spatially averaged within the area of autocorrelation equal
to 0.9 indicated in Fig 1b, before the bias correction by Quantile Mapping (a) and after the procedure (b).

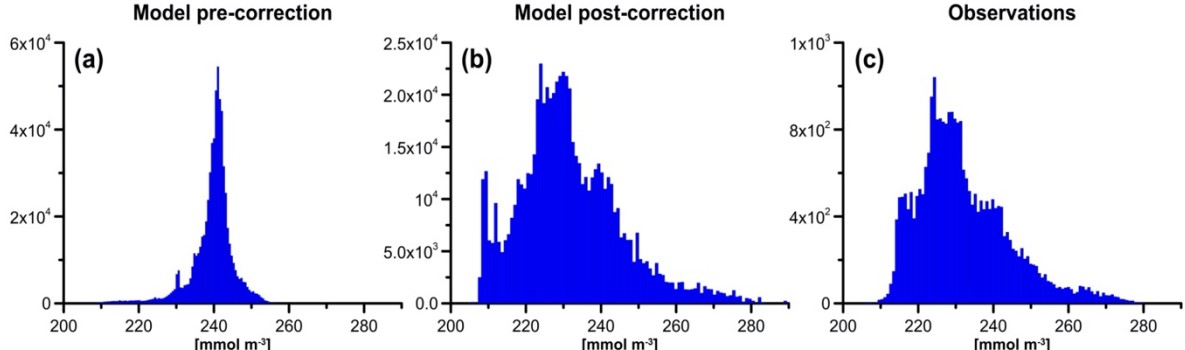

555

Figure A2: Frequency histogram of modelled oxygen concentrations before the bias correction by Quantile Mapping (a) and after
the procedure (b), compared with BGC-Argo observations (c).

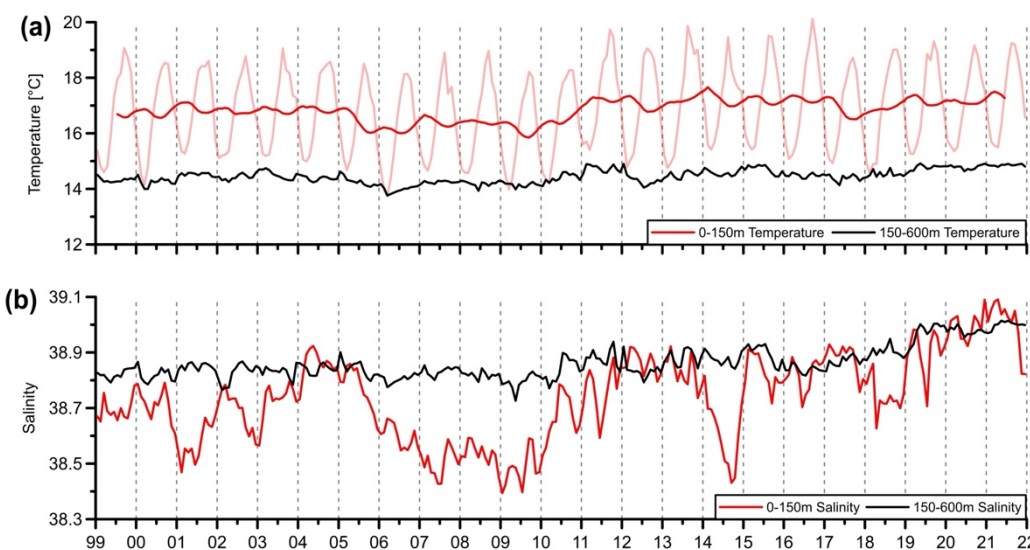

559

**Figure B1: Time series of temperature (a) and salinity (b), averaged in the vertical layers 0 - 150 m (red lines) and 150-600 m (black lines) of the Otranto Strait (39.8°N, 18.5° - 19.5° E) in the 1999-2021 time period. In the top panel, light red and dark red indicate data before and after de-seasonalization, respectively. Data are provided by Copernicus physical reanalysis (prod. ref. no. 3, Table 1).**