# Peer review of "Dissolved oxygen as indicator of multiple drivers of the marine ecosystem: the Southern Adriatic Sea case study"

_State of the Planet, 2022_

## Author Comment (AC1)

**Index**

**Reviewer#1's comments**

This paper presents an analysis of the interannual variability of dissolved oxygen in the Southern Adriatic Sea, based on analysis of the CMEMS Mediterranean reanalysis for 1999-2021, bias corrected using in situ observations made by BGC-Argo floats. EOF analysis is used to separate the main components of variability, which are then linked to different drivers. The analysis is thorough and the results are novel and interesting, but a few points are worth attention before publication.

We thank Reviewer#1 for the consideration and evaluation of our manuscript. We will carefully revise it according to the suggestions. Our answers are indicated in blue and the proposed changes to the manuscript are in red italics.

I am confused by the use of the terms temporal mode and spatial mode. The description in the methods section implies to me that the EOF analysis has been applied separately to identify variability over (vertical) space and variability over time – this would give a different set of modes in each case. But in the results the two sets are presented together, with the "spatial modes" being used to show the vertical variation seen in the "temporal modes". Wouldn't it be more accurate to refer to spatial and temporal aspects of the four modes? Or have I misunderstood the method? I suggest that either a fuller description of the methods or a revision of the language used in the results would help.

We thank Reviewer#1 for this comment, that allows us to clarify the used terminology and the EOF analysis.

In general, the terminology relevant to such a statistical method is not universally agreed: EOF analysis is also indicated as Principal Component Analysis or Eigen Analysis in literature, and also the temporal and spatial functions involved in the computations can be defined in different ways. However, the method itself is well defined (e.g., Thomson and Emery (2014), also cited in the manuscript) and widely used by the scientific community.
Following the notation in Thomson and Emery (2014), the EOF analysis allows to decompose a spatio-temporal function $\psi$ (referring to $M$ spatial locations) in a combination of orthogonal spatial functions $\phi$, whose amplitudes are weighted by time-dependent coefficients $a$:

$$\psi(x_m, t) = \psi_m(t) = \sum_{i=1}^{M} [a_i(t)\phi_{im}]$$

which are uncorrelated over the sample data. Finding $a$ and $\phi$ is equivalent to solve an eigenvalue problem, with $\phi_{im}$ eigenvectors and $\lambda = \overline{a_i(t)^2}$ eigenvalues, corresponding to the variance associated with each eigenvector. In Thomson and Emery (2014), $\phi_{im}$ are called "statistical modes" or "spatial modes" and $a_i(t)$ are called "time amplitudes" of the $i$th statistical mode.

In the manuscript, we have used the term "spatial modes" in such a sense, i.e. referred to $\phi_{im}$. Moreover, we used the expression "temporal modes" for the time amplitudes $a_i(t)$, as done in other studies (e.g., Baldacci et al., 2001; Alvera-Azcárate et al., 2009; Martellucci et al., 2021). However, we recognise that a more accurate terminology can help to better illustrate method and results.

Therefore, following Reviewer#1's suggestion, and also integrating the suggestion by Reviewer#3 of replacing "spatial" by "vertical", we propose to use the expressions: "EOF vertical patterns" and "EOF time series", "vertical pattern of the first/second/… mode" and "time series of the first/second/… mode", as done e.g. in Baldwin (2001), Espinosa-Carreon et al. (2004), Folland et al. (2009), Christoudias et al. (2012), Lee et al. (2020).
For the new version of the manuscript we will carefully revise the text associated with these expressions.

Alvera-Azcárate, A., Barth, A., Sirjacobs, D., and Beckers, J.-M.: Enhancing temporal correlations in EOF expansions for the reconstruction of missing data using DINEOF, Ocean Sci., 5, 475–485, https://doi.org/10.5194/os-5-475-2009, 2009.

Baldacci, A., Corsini, G., Grasso, R., Manzella, G., Allen, J.T. , Cipollini, P., Guymer, T.H. ,Snaith, H.M. A study of the Alboran sea mesoscale system by means of empirical orthogonal function decomposition of satellite data, Journal of Marine Systems, Volume 29, Issues 1–4, Pages 293-311, ISSN 0924-7963, https://doi.org/10.1016/S0924-7963(01)00021-5, 2001.

Baldwin, Mark P. Annular modes in global daily surface pressure, Geophysical Research Letters, 28, 21 - 0094-8276, https://doi.org/10.1029/2001GL013564, 2001.

Christoudias, T., Pozzer, A., and Lelieveld, J.: Influence of the North Atlantic Oscillation on air pollution transport, Atmos. Chem. Phys., 12, 869–877, https://doi.org/10.5194/acp-12-869-2012, 2012.

Espinosa-Carreon, T. L., Strub, P. T., Beier, E., Ocampo-Torres, F., and Gaxiola-Castro, G., Seasonal and interannual variability of satellite-derived chlorophyll pigment, surface height, and temperature off Baja California, *J. Geophys. Res.*, 109, C03039, doi:10.1029/2003JC002105, 2004.

Folland, C. K., Knight, J., Linderholm, H. W., Fereday, D., Ineson, S., & Hurrell, J. W. The summer North Atlantic Oscillation: past, present, and future. Journal of Climate, 22(5), https://doi.org/10.1175/2008JCLI2459.1, 1082-1103, 2009.

Lee, G., Ho, C. H., Chang, L. S., Kim, J., Kim, M. K., & Kim, S. J. Dominance of large-scale atmospheric circulations in long-term variations of winter PM10 concentrations over East Asia. Atmospheric Research, 238, 104871, https://doi.org/10.1016/j.atmosres.2020.104871, 2020.

Martellucci, R., Salon, S., Cossarini, G., Piermattei, V., & Marcelli, M. Coastal phytoplankton bloom dynamics in the Tyrrhenian Sea: Advantage of integrating in situ observations, large-scale analysis and forecast systems. Journal of Marine Systems, 218, 103528, https://doi.org/10.1016/j.jmarsys.2021.103528, 2021.

I am also uncertain why 2021 is picked out for particular discussion. In Figure 1d and Figure 3 it looks to me like the continuation of a trend that goes back to 2016, rather than an unusual year – why focus on this year in particular? It also appears to have some similarity to 1999 – is there some possibility of a cyclical pattern? I don't think that a major reworking of the analysis is needed, but I suggest the authors consider a shift in the way they present the 2021 results. The reference to the entrance of new water masses from the Levantine Basin is interesting – was this the reason for looking at 2021? It would be good to see more detail than is given in lines 155-157 about how unusual this change is.

The analysis conducted for the 2021 year is actually a request for the Copernicus Ocean State Report 7 (OSR7). In fact, according to the OSR7 guidelines: (i) the core-period to be covered is 1993-2021 or earlier/later, depending on product availability and limitations; (ii) the inclusion of data during the year 2021 is mandatory (see for example the explanation of the scheme of OSR in https://marine.copernicus.eu/access-data/ocean-state-report/ocean-state-report-6) .

In our case, (i) we started the analysis from 1999, since Mediterranean biogeochemical reanalysis is available from that year, and (ii) we analysed 2021 year with respect to the 1999-2020 climatology providing a discussion of the 2021 anomaly with respect to the climatology.
Given this preamble, we agree that the analysis of 2021 needs some more details and to be included in a more general discussion of the temporal dynamics of the Southern Adriatic Sea.

In particular, regarding a possible cyclical pattern, it is known that the Southern Adriatic Sea displays quite periodical behaviours on multiannual scales associated with the reversal of the Northern Ionian Gyre (e.g. Civitarese et al., 2010) regulating the waters entering from Levantine/Ionian Sea. Moreover, focusing on the vertical pattern of oxygen concentration during the year, Southern Adriatic pit is an open-ocean convection site, characterised by different phases of the convection (preconditioning, water mixing, spreading). In our case, during the convection period the surface oxygen is mixed down to the basis of the mixed layer depth, producing the high values of concentration observed in the subsurface layers (Fig. 1c) and destroying the previous vertical structure of the water masses. After the convection, the newly formed water spreads into the Ionian Sea, while the incoming Levantine water produces the minimum values of oxygen concentrations observed in the pit, restoring the typical vertical profile.
The inflow of the Northern Adriatic Dense Water (NAdDW) can furtherly increase the oxygen content in the Southern Adriatic pit; in the deepest layers of the pit the "saw tooth" mechanism (involving alternating long-lasting mixing processes and sudden density increases due to the intrusion of very dense Northern Adriatic water, e.g. in 2012, Querin et al., 2016) and more sporadic events at intermediate depths, the double salinity maximum, as indicated in Kokkini et al., 2020 for 2015-2016 years.

Also following similar suggestions by Reviewer#2 and Reviewer#3, we propose to add at the beginning of the Results Section a brief comment on this cyclicity, being the investigation of the drivers the object of the correlation analysis with EOF time series:

*Dissolved oxygen in the Southern Adriatic area (Fig. 1a) shows a cyclicity in the subsurface layers with enrichment periods (in 2004-2006, 2010-2013, 2016-2017) and sharp declines that impacted the Oxygen Minimum Layer (OML), located between 100 and 300 m. Low concentration values are observed also in the years between 1999 and 2003.*

Considering 2021 as requested from OSR guidelines, its behaviour was only partially explained by our analysis conducted with EOF and correlation with drivers. In fact, the negative anomaly of oxygen, associated with the negative anomaly of the EOF time series of the first mode, is only correlated with one of its drivers (i.e. negative anomaly in subsurface chlorophyll). In addition, we observed in the last 3 years the entrance of water masses which are much saltier than usual (as reported in Fig. R1.1 and in Mauri et al. 2021; Mihanović et al., 2021; Menna et al. 2022, as indicated e.g. in Fig. R1.2) and we hypothesise that it can be a regime shift that we will carefully monitor, as mentioned in the Discussion section.

We did not enter in more details on this part because the focus of our analysis was to identify and analyse the principal modes of variability displayed by the area and also because of the short length of the paper (the recommended number of figures and words of the paper was indicatively limited to a maximum of 4 figures and 3000 words). A more complete investigation on the origin of such waters will be the object of future work.

However, we agree with Reviewer#1 that the paper can be improved by providing more details on this particular aspect. Thus, including also a suggestion by Reviewer#2, we propose to show the time series of temperature and salinity at the surface and intermediate layer through the Otranto strait in a new appendix and to summarise the previously published results, to support such a preliminary hypothesis about the strong anomaly starting from 2019.

Civitarese, G., Gačić, M., Lipizer, M., & Eusebi Borzelli, G. L.: On the impact of the Bimodal Oscillating System (BiOS) on the biogeochemistry and biology of the Adriatic and Ionian Seas (Eastern Mediterranean). Biogeosciences, 7(12), 3987- 3997. https://doi.org/10.5194/bg-7-3987-2010, 2010.

Kokkini, Z., Mauri, E., Gerin, R., Poulain, P. M., Simoncelli, S., & Notarstefano, G. (2020). On the salinity structure in the South Adriatic as derived from float and glider observations in 2013–2016. Deep Sea Research Part II: Topical Studies in Oceanography, 171, 104625.

Mauri E., Menna M., Garić R., Batistić M., Libralato S., Notarstefano G., Martellucci R., Riccardo Gerin R., Pirro A., Hure M., Poulain P.M. in von Schuckmann et al., Copernicus Marine Service Ocean State Report, Issue 5, Journal of Operational Oceanography, 14:sup1, 1-185, DOI: 10.1080/1755876X.2021.1946240, 2021.

Menna M., Martellucci R., Notarstefano G., Mauri E., Gerin R., Pacciaroni M., Bussani A., Pirro A., Poulain P.M. Record-breaking high salinity in the South Adriatic Pit in 2020 in Copernicus Ocean State Report, issue 6, Journal of Operational Oceanography, 15:sup1, 1-220, DOI: 10.1080/1755876X.2022.209516, 2022.

Mihanović, H., Vilibić, I., Šepić, J., Matić, F., Ljubešić, Z., Mauri, E., ... & Poulain, P. M.: Observation, Preconditioning and Recurrence of Exceptionally High Salinities in the Adriatic Sea. Frontiers in Marine Science, 8, 834. https://doi.org/10.3389/fmars.2021.672210, 2021.

[Figure]

**Figure R1.1: Hovmöller diagram of seawater salinity east of Otranto Strait in the period 1999-2021 according to Mediterranean biogeochemical reanalysis (Prod3 indicated in the manuscript).**

[Figure]

**Fig. R1.2: T-S diagram of water in the southern Adriatic Pit during the 2013-2020 period (colour scale indicates time) according to BGC-Argo float observations, from Menna et al., 2022 (Figure 4.7.3.d of the reference).**

**Specific points**

line 19, 46: I don't understand "alternate" in the description of the North Ionian Gyre.

The Northern Ionian Gyre is a circulation structure that shows a reversal from cyclonic to anticyclonic and vice versa at multiannual scales (e.g Civitarese et al., 2010; Menna et al., 2019, Gačić et al 2021). In particular, circulation was cyclonic in 1999-2005, 2011-2016, 2019-2021 and anticyclonic in 2006-2010, 2017-2019. It can be also visualised from the vorticity time series in Fig. 2f, where positive (negative) vorticity indicates cyclonic (anticyclonic) circulation. We will add more details about Northern Ionian Gyre circulation in the Introduction of the revised manuscript.

Menna, M.; Reyes-Suarez, N.C.; Civitarese, G.; Gačić, M.; Poulain, P.-M.; Rubino, A.; Decadal variations of circulation in the Central Mediterranean and its interactions with the mesoscale gyres. Deep Sea Res. Part II Top. Stud. Oceanogr., 164, 14-24, https://doi.org/10.1016/j.dsr2.2019.02.004, 2019.

Gačić, M.; Ursella, L.; Kovačević: V., Menna, M.; Malačič, V.; Bensi, M.; Negretti, M.E.; Cardin, V.; Orlić, M.; Sommeria, J.; Barreto, R.V.; Viboud, S.; Valran, T.; Petelin, B.; Siena, G.; Rubino, A.;

Impact of dense-water flow over a sloping bottom on open-sea circulation: laboratory experiments and an Ionian Sea (Mediterranean) example. Ocean Sci., 17, 975–996, https://doi.org/10.5194/os-17-975-2021, 2021.

line 22-23 "We ascribe the lower content of DO in 2021 to a negative anomaly of the subsurface production in the same year, in agreement with the previous correlation analysis, but not to heat fluxes": this point made in abstract is not stated explicitly in the main body of the paper (though it is implied).

We agree with Reviewer#1. We will explicitly indicate this point in the revised paper in the section dedicated to 2021 anomaly as following:

*The first EOF temporal mode (Fig. 3a) is actually negative from 2019, and it corresponds to the negative anomaly of one of its drivers (Table 2), i.e. subsurface chlorophyll, and not heat fluxes. In particular, we estimated a 2021 mean negative anomaly approximately equal to 6% with respect to the climatological mean (1999-2020) for subsurface chlorophyll.*

line 23-25 "we observe the entrance of warmer and exceptionally saltier waters favored by the cyclonic circulation of NIG from 2019 onwards": cyclonic circulation is also observed in 1999-2005 and most of 2011-2016; no evidence is presented about the temperature and salinity of the water.

We thank Reviewer#1 for this comment, that allows us to improve the text of our manuscript.
In Menna et al. (2022), i.e. the reference cited at line 154 of the submitted manuscript, the salinity data coming from World Ocean Database and Argo floats in the subsurface (100-200 m) and intermediate (200-800 m) layers in the Southern Adriatic pit are displayed in the 1990-2020 period and shows that maximum values (approximately equal to 38.95 psu) are recorded in 2019–2020 in the sub-surface layer and in 2020 at intermediate depth.
As we indicated in our reply to the general comments, we propose to show in a new appendix of the revised manuscript the time series of salinity and temperature at the surface and intermediate layer through the Otranto Strait in the considered period (1999-2021) and to summarise the previously published results.

line 67-74: regarding the bias correction, was the bias consistent across the three periods where float data was available? Did the bias correction allow for uncertainty in the in situ observations?

We employed the bias correction as a statistical procedure over all available data. The uncertainties in observation, provided by the Quality Control procedure of BGC-Argo as indicated in Product 2 (https://doi.org/10.13155/75807), were not explicitly accounted for in the bias correction procedure.

line 97: the dimensions of the SAdr box are much smaller than the region from which DO data was taken (i.e. the 0.9 autocorrelation contour). Why was a smaller region used for the forcing indices?

We thank Reviewer#1 for raising this point, that allows us to clarify an aspect of our method.

We actually used two criteria to spatially delimit the domain under study. The first one is the spatial autocorrelation analysis (>0.9) of biogeochemical reanalysis data, that allowed us to identify the area displaying the same dynamics at the surface from a phenomenological point of view. The second criterion was instead related to the bathymetry (i.e. depths), to consider the whole volume of the Southern Adriatic pit. The choice of the box for the forcing fields derives from this second criterion, and the identified area at surface is included in the previous one.

We recognise that this part should be added to the manuscript and we will indicate it in the section Data and methods in the revised manuscript.
Moreover, we will specify that the biogeochemical reanalysis data considered in the spatial autocorrelation analysis are dissolved oxygen, nitrate and chlorophyll concentrations at surface and, also accounting for a comment by Reviewer#2, we will clarify that the autocorrelation analysis should be actually named "cross-correlation" (between the data in the central point of the pit and the ones at every spatial gridpoint in the domain, as in Jones et al., 2015; Martellucci at al., 2021).

Jones E.M., Doblin M.A., Matear R. , King E. Assessing and evaluating the ocean-colour footprint of a regional observing system J. Mar. Syst., 143 (2015), pp. 49-61, https://doi.org/10.1016/j.jmarsys.2014.10.012

line 119 Fig.3c-e: This should be 3c,e if it's referring to the temporal modes only or 3c-f if it's referring to both temporal and spatial modes.

We agree and we will correct the reference to the Figures.

line 129 "the first mode explains the variability (e.g. seasonal) connected with solubility": I'm not sure why the third mode is not involved too – it has a correlation of 0.51 with heat flux and hence, presumably, to temperature and solubility.

We agree with Reviewer#1, since also the third mode has a correlation with heat fluxes in the area. However, the first mode explains a higher percentage of total variance, i.e., we ascribe the variability connected with solubility mainly with the first mode.
We will rephrase this part including also the third mode as a further contribution.

line 167-8 "We do not recognise a clear deoxygenation trend in the subsurface layer": there seems to be a very clear decline for 2010-2021. Is the point that there is no overall decline in the period 1999-2021 (though there is a rise then a fall)?

The oxygen time series (Fig. 1d) does not show a negative trend in the subsurface layer in the whole considered period (1999-2021) and its dynamics are characterised mainly by the inter-annual variability related to the drivers, rather than to an overall trend. We will better specify this part in the revised paper and we will also improve the Figure 1d by adding ticks identifying the years and contour lines to favour readability and interpretation of the figure.

line 168-169 "the multiannual variability is characterized by a sort of cyclicity": I'm not sure what is meant here.

We thank Reviewer#1 for this comment, that allows us to clarify the text of the manuscript. Here we referred to the cyclical pattern of the oxygen concentrations in the subsurface layers, that are higher in the period 2004-2006, 2010-2013, 2016-2017 than in the other years. Please see also our reply to the general comments on page 3 about the cyclical pattern of the oxygen concentration in the Southern Adriatic pit.
We will better specify this part in the text in the revised version.

line 170-172: I'm not sure that I understand the point being made here – is it that the SAdr responds quickly to change because of the small water volume and residence time?

We thank Reviewer#1 for this comment, that allows us to clarify the text of the manuscript. The Southern Adriatic pit is actually characterised by a short residence time of water, due to its small volume and to the strength of its meteo-marine drivers. This feature allows to identify possible recurrent behaviours and also results in a quick reply to changes in the drivers themselves.
We will rephrase this part in the revised paper.

Overall I think this is a good paper and I enjoyed reading it. But I advise the authors to go through the text carefully to make sure that their main findings are presented really clearly and backed up by enough information. I also recommend an English language edit – the paper is generally well written but in a few places I was unclear about the meaning.

We thank Reviewer#1 for the feedback. We will modify the manuscript as indicated in our replies and will check the English Language by a Professional Service.

**Reviewer#2's comments**

The manuscript investigates the interannual variability of dissolved oxygen (DO) over the last 23 years driven by multiple mechanisms (atmospheric, circulatory, and biological processes) in the southern Adriatic Sea. The aim is to demonstrate the importance of DO as indicator of current changes and environmental status.

The study is based on modelled and observational data from the Copernicus Marine Service: the biogeochemical reanalysis for the Mediterranean Sea covering the 1999-2021 period and BGC-Argo float measurements for the 2014-2020 period. DO estimated by BGC-Argo floats are used to bias-correct the modelled DO along the entire reanalysis time series using a quantile mapping technique.

The bias-corrected reanalysis signal is then decomposed using EOF analysis. Then a correlation analysis between the first four EOF temporal modes of variability and key drivers allows evaluating the relative importance of the different drivers to explain DO variability. Finally, year 2021 is compared to the 1999-2020 climatology to identify the main contributor(s) in 2021.

This is a very interesting study which has multiple interests in the context of actual changes. And DO is a good candidate to detect and monitor changes. However, I have some concerns about the foundations of the study. They are detailed below.

We thank Reviewer#2 for the consideration and evaluation of our manuscript. We will carefully revise it according to the suggestions. Our answers are in blue and the proposed changes to the manuscript are in red italics.

1. The ability of the QM method in correcting the bias of modelled DO.

QM covers a variety of methods that do not necessarily have a similar ability to correct model bias for a given domain or a given variable. Some QM methods may even deteriorate the model outputs compared to the non-corrected form. So it is therefore important to choose an appropriate bias correction method.

- Have you compared different QM methods? Does the performance vary significantly from each other? Please describe strengths and weaknesses of the QM method used for this study.

We thank Reviewer#1 for this comment, that allows us to clarify the method we used.

We adopted a bias correction procedure to integrate observations and modelled data of oxygen in the area, with the aim of improving the representation of oxygen concentrations at the local scale. In fact, the biogeochemical reanalysis does not assimilate oxygen observations, whose availability from BGC-Argo profiles is particularly large in the area, and the validation of the biogeochemical reanalysis (Cossarini et al., 2021 and Teruzzi et al., 2021) showed some bias errors.

Therefore, we decided to apply the bias correction procedure, as is commonly done (as illustrated e.g. in the review by Teutschbein and Seibert, 2012). We tested several methods for bias correction: (i) Additive delta change and (ii) Multiplicative delta change (e.g. Hempel et al., 2013; Maraun and Widmann, 2018); (iii) Variance scaling (e.g., Rocheta et al., 2014); (iv) Quantile mapping (e.g., Hopson and Webster, 2010; Themeßl et al., 2011; Gudmundsson et al., 2012, with references indicated in the bibliography of the manuscript) and we performed visual inspection of

the Hovmöller diagrams. We chose the Quantile Mapping because it allows us not only to characterise the bias over the entire distribution of the variable under study (e.g. Li et al., 2010), but also to not modify the main dynamics related to the dissolved oxygen, that are quite satisfactorily reproduced by the model in the considered period.

In Fig. R2.1 we report the Hovmöller diagram of dissolved oxygen concentration before the bias correction by Quantile Mapping (top panel) and after the procedure (bottom panel). The main dynamics observed before the procedure are maintained also after the correction: surface mixing and stratification during the year, subsurface oxygen maximum onset in spring and development in summer (Di Biagio et al., 2022) as an emerging feature of the marine ecosystem, interannual variability related to the mixed layer depth dynamics (i.e. higher values of concentrations under 300 m in 2004-2006, 2010-2013, 2016-2017, as explained in this reply at page 14). In fact, the effect of the bias correction is mainly related to the absolute values of the oxygen concentrations (please see the two different colorbars in top and bottom panels).

[Figure]

**Figure R2.1: Hovmöller diagram of the modelled oxygen concentrations spatially averaged in the area of autocorrelation equal to 0.9 indicated in Fig 1b of the manuscript, before the bias correction by Quantile Mapping (top panel) and after the procedure (bottom panel).**

The difference between the distribution of the absolute values of the model output (before and after the correction) and the ones from BGC-Argo floats can be visualised also in Fig. R2.2, displaying the frequency histogram of oxygen concentrations. The Quantile Mapping correction allows modifying the modelled values in order to reproduce the shape of the distribution of the observations. After the correction, modelled data have higher variability and the shape of the distribution displays a clear tail toward the higher values, similarly to the observations.

[Figure]

**Figure R2.2: Frequency histogram of modelled oxygen concentrations before the bias correction by Quantile Mapping (a) and after the procedure (b), compared with BGC-Argo observations (c).**

Moreover, in order to have an independent validation of the procedure, we made a comparison between the model output (before and after the bias correction) with independent observations in the area, coming from the "EMODnet_int" dataset, defined in Cossarini et al., 2021 as EMODnet collection of in situ data (1999-2016 time period) integrated with additional oceanographic cruises (references in Cossarini et al., 2015; Lazzari et al., 2016). As reported in Fig. R2.3, the bias correction procedure by Quantile Mapping allowed us to reduce the differences between modelled and observed oxygen concentrations at surface and under 100 m, without modifying the depth of subsurface oxygen maximum. Despite a residual bias under 400 m depth, that we ascribe to some underestimated processes in the model (related to the biological respiration), this result further supports the choice of the Quantile Mapping as a proper bias correction procedure in our case study.

In conclusion, it appears that the model could be too energetic in the vertical processes, and it produces too homogeneous oxygen vertical profiles. Indeed, Quantile Mapping successfully improves the data distribution of oxygen data and its comparison with independent observations without modifying the temporal succession of maxima and minima in the timeseries of the Southern Adriatic area.

We recognise that this part should be furtherly extended and explained more in detail, therefore we propose to add in a new appendix additional details on the used Quantile Mapping method, including Figs. R2.1 and R2.2 or some equivalent graphics.

[Figure]

**Figure R2.3: Vertical profile of modelled oxygen concentrations before the bias correction by Quantile Mapping (left panel, blue line) and after the procedure (right panel, black line), with values horizontally averaged in the Southern Adriatic Sea (within the cross-correlation area r>0.9, Fig.1 of the manuscript), compared with observations from Emodnet_int dataset (Cossarini et al., 2021) in the same area, vertically averaged in the layers 0-25 m, 25-50 m, 50-75 m, 75-100 m, 100-125 m, 125-150 m, 150-200 m, 200-400 m, 400-600 m (red dots). Dashed lines for model and observations profiles indicate the standard deviation of data.**

Cossarini, G., Lazzari, P., and Solidoro, C. Spatiotemporal variability of alkalinity in the Mediterranean Sea. Biogeosciences 12, 1647–1658, 2015.

Cossarini, G., Feudale, L., Teruzzi, A., Bolzon, G., Coidessa, G., Solidoro, C., Di Biagio, V., Amadio, C., Lazzari, P., Brosich, A., and Salon, S.: High-Resolution Reanalysis of the Mediterranean Sea Biogeochemistry (1999–2019), Front. Mar. Sci., 8, 1537, https://doi.org/10.3389/fmars.2021.741486, 2021.

Di Biagio, V., Salon, S., Feudale, L., and Cossarini, G.: Subsurface oxygen maximum in oligotrophic marine ecosystems: mapping the interaction between physical and biogeochemical processes, Biogeosciences, 19, 5553–5574, https://doi.org/10.5194/bg-19-5553-2022, 2022.

Hempel, S., Frieler, K., Warszawski, L., Schewe, J., and Piontek, F.: A trend-preserving bias correction – the ISI-MIP approach, Earth Syst. Dynam., 4, 219–236, https://doi.org/10.5194/esd-4-219-2013, 2013.

Lazzari, P., Solidoro, C., Salon, S., and Bolzon, G. Spatial variability of phosphate and nitrate in the Mediterranean Sea: a modelling approach. Deep Sea Res. 108, 39–52. doi: 10.1016/j.dsr.2015.12.006, 2016.

Li, H. Sheffield J., Wood E.F. Bias correction of monthly precipitation and temperature fields from Intergovernmental Panel on Climate change AR4 models using equidistant quantile mapping. J. Geophys. Res. 115: D10101, doi: https://doi.org/10.1029/2009JD012882, 2010

Maraun, D., & Widmann, M. . Statistical downscaling and bias correction for climate research. Cambridge University Press., DOI: 10.1017/9781107588783, 2018

Rocheta E., Evans J.P. and Sharma A. Assessing atmospheric bias correction for dynamical consistency using potential vorticity, Environ. Res. Lett. 9 124010, DOI 10.1088/1748-9326/9/12/124010

Teruzzi, A., Di Biagio, V., Feudale, L., Bolzon, G., Lazzari, P., Salon, S., Coidessa, G., and Cossarini, G.: Mediterranean Sea Biogeochemical Reanalysis (CMEMS MED- Biogeochemistry, MedBFM3 system) (Version 1) Copernicus Monitoring Environment Marine Service (CMEMS) [data set], https://doi.org/10.25423/CMCC/MEDSEA_MULTIYEAR_ BGC_006_008_MEDBFM3, 2021.

Teutschbein, C., & Seibert, J. Bias correction of regional climate model simulations for hydrological climate-change impact studies: Review and evaluation of different methods. Journal of hydrology, 456, 12-29, https://doi.org/10.1016/j.jhydrol.2012.05.052, 2012.

Thrasher B., Maurer E.P., McKellar C., Duffy P.B. Technical Note: Bias correcting climate model simulated daily temperature extremes with quantile mapping. Hydrol. Earth Syst. Sci. 16:3309–3314. doi: 10.5194/hess-16-3309-2012, 2012.

- In the current draft, it is unclear to me how the reanalysis product is corrected with the BGC-Argo profiles. More details are needed on this correction.

The Copernicus reanalysis has been corrected with BGC-Argo profiles only in retrospect, using the Quantile Mapping as a bias correction procedure on the oxygen concentration in the Southern Adriatic Sea.

In fact, the current version of the Copernicus reanalysis (Cossarini et al., 2021 and Teruzzi et al., 2021) only features the assimilation of surface chlorophyll concentration (observations from Ocean Color product based on ESA-CCI data). The data assimilation is performed once a week during the reanalysis run through a variational scheme (3DVarBio, see details in Teruzzi et al., 2014, 2018, 2019). The Copernicus reanalysis assimilates chlorophyll concentration to correct the four phytoplankton functional groups (17 state variables including carbon, chlorophyll, nitrogen phosphorus and silicon internal quotas) of the BFM biogeochemical model.

However, we will add more details on the used Quantile Mapping method as bias correction procedure in a new appendix (please see our reply to the previous comment).

- Are there no other in-situ data (other than Copernicus In-Situ TAC) to evaluate the bias-corrected DO time series (before BGC-Argo area) and confirm the ability of the QM method to correct the modelled DO ?

Please see our reply to a previous comment at pages 11-12 about the comparison of model output before and after the bias correction by Quantile Mapping with an independent dataset of in situ oxygen data.

- To validate the QM method, it would be very helpful to provide the Hovmoller diagram of modelled DO (same as Figure 1d) before applying the bias-correction procedure.

We agree. Please see our reply at pages 10-11 about the new appendix we propose to add to the manuscript.

- The bias-corrected DO time series (Fig 1d) shows that the deep layers are progressively enriched with oxygen over a period of 7-8 years (1999 to 2006), and then a mechanism re-initialize the oxygen at depth. Another cycle starts again (2007 to 2013), and then a third one seems to start and then perhaps disarmed by the bias-correction. This cyclic deep enrichment in oxygen is not really discussed in the paper. Is it an observed feature ? Is it already present in the non-corrected time series ? Is it the result of the bias-correction ? It could even be interpreted like a drift of the modelled DO, with a relaxation to initial conditions each 7-8 years. A lot of questions can arise from this cyclic enrichment in DO... So more clarification is needed.

As already explained in our reply to Rev#1's comments, the initial draft of this contribution followed the guideline of the Ocean State Report in terms of number of figures, length of the text and focus on the last available year. However, we agree that the multiyear cycle is an interesting feature worth to be discussed with more details.

As we replied also to Reviewer#1 (page 3), the cyclicity of oxygen concentrations in the water column is mainly associated with the mixed layer depth (Fig. 2b) on the annual scale (leading to a deep enrichment in oxygen) and with the NIG circulation (Fig. 2f) on the multiannual scale (generally an enrichment during anticyclonic phase and reduction during the cyclonic phase). These features make the statistical analysis in terms of EOF particularly suitable for our study. Moreover, the inflow of the Northern Adriatic Dense Water (NAdDW) can furtherly increase the oxygen content in the Southern Adriatic pit; in the deepest layers of the pit the "saw tooth" mechanism (involving alternating long-lasting mixing processes and sudden density increases due to the intrusion of very dense Northern Adriatic water, e.g. in 2012, Querin et al., 2016) and more sporadic events at intermediate depths, the double salinity maximum, as indicated in Kokkini et al., 2020 for 2015-2016 years.

The higher values of oxygen concentration in subsurface layers in years 2004-2006, 2010-2013, 2016-2017 with respect to the other years is clearly visible also in the Hovmöller plot of oxygen concentration before the bias correction, as indicated in Figure R2.1 (that we propose to add in a new appendix related to the Quantile Mapping bias correction method).

Since the investigation of the drivers is the object of the correlation analysis with EOF time series, we propose to add at the beginning of the Results Section a brief comment on this cyclicity:

*Dissolved oxygen in the Southern Adriatic area (Fig. 1a) shows a cyclicity in the subsurface layers with enrichment periods (in 2004-2006, 2010-2013, 2016-2017) and sharp declines that impacted the Oxygen Minimum Layer (OML), located between 100 and 300 m. Low concentration values are observed also in the years between 1999 and 2003.*

Querin, S., Bensi, M., Cardin, V., Solidoro, C., Bacer, S., Mariotti, L., ... & Malačič, V. (2016). Saw-tooth modulation of the deep-water thermohaline properties in the southern Adriatic Sea. Journal of Geophysical Research: Oceans, 121(7), 4585-4600.

Kokkini, Z., Mauri, E., Gerin, R., Poulain, P. M., Simoncelli, S., & Notarstefano, G. (2020). On the salinity structure in the South Adriatic as derived from float and glider observations in 2013–2016. Deep Sea Research Part II: Topical Studies in Oceanography, 171, 104625.

2. The necessity to bias-correct the modelled DO

Modelled DO is corrected ? But why is it necessary?

As we have replied to a comment at page 13, modelled dissolved oxygen concentration has been corrected only in retrospect (i.e. the Copernicus product does not include such a correction). However, since this study has been conducted on a local scale in which there is a large availability of BGC-Argo floats, we also integrated observations from floats to reduce the model bias (Cossarini et al., 2021; Teruzzi et al., 2021) and to better reproduce local dynamics.

It is difficult to appreciate the bias correction. Need to see the temporal evolution of modelled DO before applying the bias-correction (hovmoller diagram).

As we have replied before, we will include in a new appendix more details on the Quantile Mapping method, including the Hovmöller diagram of dissolved oxygen before the bias correction.

Bias-correction modifies the signal and may alter the link between DO and its drivers and thus reduce the correlation between the decomposed DO time series and forcing indexes... In fact, all correlations are quite "low", at best "moderate". Can't it come from this bias-correction? Have you tried the same analysis without bias-correction ?Or maybe, the correction does not modify the signal... but in this case, more clarification is needed in the text.

We applied the bias correction in order to "align" model results to the observations, thus correcting possible model errors. Then, the EOF analysis has been applied to the best time series available, that is, indeed, the timeseries corrected by the bias correction technique.

Furthermore, as shown in the figures R2.2 and R2.3, the bias correction increases variability of data distribution without modifying the temporal succession of maxima and minima and the shape of vertical oxygen profiles. Thus, EOF analysis should not be substantially impacted by the bias correction. Indeed, for sake of curiosity and completeness, we replicated the same EOF analysis on the Hovmöeller diagram before the bias correction (top panel in Figure R2.1). Results are pretty analogous to the ones obtained with the bias correction.

In particular, both EOF vertical patterns and EOF time series of the modes of dissolved oxygen before the bias correction (Fig. R2.4b,d,f,h) are quite similar to the analysis done on the bias corrected output (Fig. 3b,d,f,h in the manuscript). We can spot some differences in the intermediate part of the EOF vertical pattern of mode 1 only. We believe that this reflects the less variability of data in the model result before the bias correction (e.g., model profiles before bias correction are vertically more homogeneous in the subsurface part, Fig. R2.3).

[Figure]

**Figure R2.4:** EOF time series (a, c, e, g) and vertical pattern (b, d, f, h) of the first four modes computed on the dissolved oxygen concentration in the Southern Adriatic area shown in the top panel of Fig.R2.1. The explained variances of the four modes are: 41.7%, 30.3%, 18.9%, 4.5%.

|  | mode 1 | mode 2 | mode 3 | mode 4 |
|---|---|---|---|---|
| **HFlux (SAdr)** | -0.19 | 0.35 | 0.63 | 0.31 |
| **MLD (SAdr)** | 0.41 | -0.18 | -0.15 | -0.2 |
| **surf chl (SAdr)** | 0.62 | -0.14 | -0.25 | n.s. |
| **subsurface chl (SAdr)** | n.s. | 0.25 | 0.49 | 0.35 |
| **Hflux NAdr (2-months lagged)** | -0.71 | 0.26 | 0.30 | 0.13 |
| **NIG vorticity (NIon)** | n.s | -0.40 | 0.27 | -0.39 |

**Table RT2.1:** Correlations between the EOF time series of the first four modes (Fig.R2.4a,c,e,g) and the forcing fields (Fig. 2 of the manuscript, with heat fluxes in the Northern Adriatic Sea time-lagged by two months). Not significant correlations are identified by a significance level higher than 0.05 and indicated by "n.s." acronym in the table.

Correlation analysis (Table RT2.1 and Table 2 in the manuscript) shows that the signs of the relationship between EOF time series and drivers remain the same for modes 2, 3 and 4. Also, the correlation values are pretty similar. Only mode 1 shows some differences: it captures mostly the dynamics of the very surface layer while the remaining part of the profile shows very small variability (Fig. R2.4b) and correlations become higher for northern Adriatic heat fluxes, surface chlorophyll and mixed layer depth drivers. The difference in the intermediate and deeper layers between the two first modes (Fig. 3 of the original manuscript and Fig. R2.4) could be ascribed to the uncorrected model. In fact, the model appears to be too energetic in the subsurface layer producing too homogeneous vertical profiles below 50-100m (Fig. R2.3). Thus, we think that the EOF analysis applied to the bias corrected model results provided more reliable indications.

As a conclusion, since the output of the statistical analysis is not modified in a relevant way, and the bias correction validation produced encouraging results (please see our reply at pages 11-12), we consider that the method we applied is scientifically valid.

3. The focus on the interannual variability

The draft states that this study focuses on DO interannual variability related to multiple drivers.

But, 5 over the 6 forcing indexes presented on Figure 2 mainly highlight a seasonal pattern. Seasonal pattern is also a main feature of the 3 first EOF temporal modes. And a major part of the results (section 3.2) aims to characterize the seasonal variability.

If the paper is intended to address DO inter-annual variability and long-term dynamics, shouldn't the seasonal cycle be removed from the signal before starting the analysis ? Have you tried this ?

We thank Reviewer#2 for having raised this point, that allowed us to clarify a point of our paper.

We have actually tried to identify all the temporal components present in the profile time series (i.e. Hovmöeller diagram) of oxygen concentrations by EOF decomposition, rather than decompose the time series by classical time series decomposition (e.g., as T = interannual + seasonal + irregular by means of methods like X-11 (e.g. Colella et al., 2016)) and then extract the interannual component and compare it with the deseasonalized time series of the drivers.

As noticed also by Reviewer#1, the time series shows a cyclicity over a time period of some years, which might not be revealed by classical time series decomposition methods. Thus, we preferred to apply EOF analysis on the not deseasonalized time series. In fact, we considered that, on one hand, the seasonality would have emerged in the EOF time series and, on the other hand, that we could verify if multiannual cyclicity could have affected the seasonal cycle.

Colella, S., Falcini, F., Rinaldi, E., Sammartino, M., & Santoleri, R. (2016). Mediterranean ocean colour chlorophyll trends. PloS one, 11(6), e0155756.

4. The interpretation of the correlation analysis

The study is based on an analysis of correlation between the first four EOFs temporal modes and the six main forcing in the area. On the 24 correlations calculated, only 4 are greater than 0.5 (with

a maximum correlation of 0.68), 15 are less than 0.5 and 5 are not significant. I find the link between EOFs modes and forcing indexes a bit week, and the message sometimes confusing, with the term "significant correlation". A statistically significant correlation must not be confused with relevant correlation. Please make a clear distinction between the two.

We agree and we will carefully revise the used terms.

The scale used to classify correlations is not really appropriated. It is somewhat exaggerated to mention a correlation of r=-0.61 as "strongly correlated".

A reasonable classification would be:     r < 0.5 à low correlation

0.5 < r < 0.7 à moderately correlated

0.7 < r < 0.9 à highly/strongly correlated

We agree and we will modify the used terminology with expressions like: "a significant but low correlation", "significant and moderate correlation between…"

Maybe, scatter plots (EOF temporal mode vs forcing index) would be helpful to appreciate the relationship between the 2 signals.

We appreciate this suggestion by Reviewer#2, but, since we have considered the correlations between the EOF time series of (the first) four modes (Fig. 3a,c,e,g) and six drivers (Fig. 2), it would lead to adding 24 plots (or panels). In the perspective of a short paper (in line with the general OSR7 guidelines), we think that there would be too much detailed information to show and comment.

 Specific Comments

L14: "we used DO modelled by the latest Copernicus Marine biogeochemical reanalysis", please add "for the Mediterranean Sea".

We agree.

L68: why is it necessary to correct the reanalysis? please add a few words

We agree. In the new manuscript version, we will better explain why it is necessary to introduce the correction.

L77: what does "auto correlation" mean ?

We thank Reviewer#2 for this comment, that allows us to clarify the used terminology.

The "autocorrelation" is actually a cross-correlation between the central point of the Southern Adriatic pit and every spatial gridpoint in the domain (Jones et al., 2015, Martellucci et al., 2021). As we replied to Reviewer#1, we will also specify in the revised manuscript that the biogeochemical reanalysis data considered in the spatial cross-correlation analysis are dissolved oxygen, nitrate and chlorophyll concentrations at surface.

Jones E.M., Doblin M.A., Matear R. , King E. Assessing and evaluating the ocean-colour footprint of a regional observing system J. Mar. Syst., 143 (2015), pp. 49-61, https://doi.org/10.1016/j.jmarsys.2014.10.012

Martellucci, R., Salon, S., Cossarini, G., Piermattei, V., & Marcelli, M. Coastal phytoplankton bloom dynamics in the Tyrrhenian Sea: Advantage of integrating in situ observations, large-scale analysis and forecast systems. Journal of Marine Systems, 218, 103528, https://doi.org/10.1016/j.jmarsys.2021.103528, 2021.

L85: Is Pearson correlation used ?

Yes, we used Pearson correlation. We will specify it in the revised manuscript.

L97: why is the SAdr box used to average forcing (41.6-42.1°N / 17.6-18.1°E) different from the SAdr area used to average modelled DO (area of autocorrelation 0.9) ?

As also replied to Reviewer#1, we recognise that an explicit explanation about this choice is missing in the submitted manuscript and we will add it in the revised one.

In fact, we actually used two criteria to spatially delimit the domain under study. The first one is the spatial cross-correlation analysis (>0.9) of biogeochemical reanalysis data, that allowed us to identify the area displaying the same dynamics at the surface from a phenomenological point of view. The second criterion was instead related to the bathymetry (i.e. depths), to consider the whole volume of the Southern Adriatic pit. The choice of the box for the forcing fields derives from this second criterion, and the identified area at surface is included in the previous one.

L102-103: why has the vorticity been filtered ? why other forcing indexes have not been filtered ? please clarify

The 13-month moving average is a procedure commonly done on NIG vorticity (e.g. Menna et al., 2019) to highlight the cyclonic and anticyclonic periods in the northern Ionian Sea circulation.

However, we recognise that the use of not filtered time series should be applied also in case of NIG vorticity, also in the perspective of including all the temporal components as illustrated in our reply to a previous comment at page 17.

We applied the correlation analysis also to the not filtered NIG vorticity time series and we obtained similar values to the ones reported in the manuscript: -0.40 for the second mode and -0.37 with the fourth mode. The main difference was that the correlation with the third mode was not significant. We propose to replace the values in the last row of the Table 2 in the manuscript with the ones here indicated, to modify the lines in the text associated with this part (lines 102-103, 126-128) and to modify Fig. 2f displaying NIG vorticity time series.

L117: change "significantly correlated…" to "statistically significant but moderate correlation with the heat flux (r=0.56) and low correlation with the subsurface chlorophyll concentration (r=0.43)"

We agree, but we would prefer to change "low" with "lower".

L123: "strongly correlated" for r=-0.61 and r=0.68 is somewhat exaggerated… please change to "moderately correlated"

We agree.

L139-141: I would not say that Fig 1d shows 2 distinct periods 2005-2006 and 2012-2014, but rather 2 periods of 7-8 years each with progressive oxygenation of the deep layers (1999-2006 and 2007-2013). Is this the third mode that explains this deep oxygenation ? Which mechanism re-initialize the oxygen content at depth in 2007 and 2014 ?  Is it confirm by analysis of DO advection from northern Adriatic sea ? here a time series of DO advection would confirm the hypothesis.

As we have already replied at page 14, the cyclicity of the oxygen concentration in the subsurface layers is related to multiple factors: the mixed layer depth dynamics (Fig. 2b), the NIG circulation (2f), and in deeper layers to deep water inflow from Northern Adriatic Sea (heat fluxes in Northern Adriatic Sea in Fig. 2e).

Regarding the inflow of NAdDW, it causes an enrichment of oxygen in the intermediate and deep layer.  The inflow of NAdDW in the Southern Adriatic pit can be visualised e.g. at BB mooring site (Fig. R2.5, data from Paladini de Mendoza et al., 2022), that shows potential density higher than 29.2 kg m$^{-3}$ in 2012-2013 and in 2017-2018. Since dense waters coming from the Northern Adriatic Sea are rich in oxygen, such density values support our interpretation of enrichment in oxygen in the deeper layers due to this forcing: in Table 2, r=0.68 for heat fluxes in the Northern Adriatic Sea (Fig. 2e) and the third mode of EOFs (Fig. 3e,f), even if the enrichment in years 2017-2018 (recognisable in Fig. 1d) is not clearly captured by the time series of the third mode of EOFs.

Furthermore, the pronounced NAdDW formation that occurred in 2005-2006 is well documented by observations illustrated in other studies (e.g. Socal et al., 2008).

[Figure]

**Figure R2.5: Potential density of seawater recorded in the BB mooring site in the Southern Adriatic Sea from 2012 to 2020.**

On the other hand, other processes (e.g., entrance of less oxygenated water from Otranto) can be the most relevant drivers in specific years causing the observed decrease of concentration. For example, as shown by our analysis results (e.g., positive values of EOF time series of mode 4 in 2015-2016 and 2019-2021 and correlation of mode 4 with NIG) we hypothesise that the mechanism of oxygen decrease in the intermediate layer is associated to the NIG and the inflow of less oxygenated waters through the Otranto Strait.

Unfortunately, oxygen advection flows were not part of the output of the Copernicus Marine Service reanalysis of Mediterranean Sea and a reconstruction in retrospect of the advection from the model output can be problematic. Therefore, we proposed to identify the relevant processes and drivers indirectly with our EOF and correlation analysis.

Paladini de Mendoza F., Schroeder K., Langone L., Chiggiato J., Borghini M., Giordano P., Verazzo G., & Miserocchi S.. Moored current and temperature measurements in the Southern Adriatic Sea at mooring site BB and FF, March 2012-June 2020 (1.0) [Data set]. Zenodo. https://doi.org/10.5281/zenodo.6770202, 2022.

Socal, G., Acri, F., Bastianini, M., Bernardi Aubry, F., Bianchi, F., Cassin, D., Coppola, J., De Lazzari, A., Bandelj, V., Cossarini, G. and Solidoro, C. Hydrological and biogeochemical features of the Northern Adriatic Sea in the period 2003–2006. Marine Ecology, 29: 449-468, https://doi.org/10.1111/j.1439-0485.2008.00266.x, 2008.

L142: The analysis performed on detrended DO time series provides pretty similar results.But did you try to perform the analysis on de-seasonalized DO time series ?

Please see our reply to a previous comment on deseasonalized time series at page 17.

L151-152: I am sorry, I am not able to see the anomaly (lower than average) in subsurface chlorophyll in Fig 2d… Could you give more details please?

We recognise that the negative anomaly of subsurface chlorophyll is not so evident from Fig. 2d. We estimated that this negative anomaly averaged during 2021 year is equal to 6% with respect to the climatological mean (1999-2020) and we propose to modify this sentence as:

*The first EOF temporal mode (Fig. 3a) is actually negative from 2019, and it corresponds to the negative anomaly of one of its drivers (Table 2), i.e. subsurface chlorophyll, and not heat fluxes. In particular, we estimated a 2021 mean negative anomaly approximately equal to 6% with respect to the climatological mean (1999-2020) for subsurface chlorophyll.*

L152: please add the reference to Fig. 2d at the end of the sentence.

We agree.

L153-155: 2021 anomaly is an important result of the paper, and a central aspect for the Ocean State Report. I think that figure(s) is/are missing to support the text of Section 3.3. For exemple, a time serie of Temperature, salinity and oxygen content at entrance of the SAdr trough the Otranto Strait would confirm the regime shift mentioned in the text, with the entrance of new water masses (warmer, saltier and less oxygenated) well reproduced by the physical and biogeochemical models.  And the reasons why circulation has changed could be developed further.

We thank Reviewer#2 for this comment, that allowed us to better present our results.

As we have replied also to Reviewer#1, we will include in a new appendix a plot to show the time series of temperature and salinity of water entering the Adriatic Sea through the Otranto Strait. Moreover, since an increase in salinity in the Southern Adriatic pit has been already detected and described in previous studies (Mauri et al. 2021; Mihanović et al., 2021; Menna et al., 2022, please see the Fig. R2.5 taken from the study at page 5), in the reviewed version of the manuscript we will also include a paragraph summarising those published results which help to understand the regime shift of the oxygen we observed.

Finally, regarding the change in circulation, a more complete investigation on the origin of such saltier Levantine waters is the object of an ongoing work, that is out of topic with respect to our study.

L169-170: "the cyclicity" is briefly mentioned here while it is an obvious feature in figure 1. More discussion about this point is necessary.

We agree. Please see our reply at page 14.

Figure 1a: please add the bathymetry.

We agree and we will provide the bathymetry in a new version of Fig. 1a.

Figure 2: is Product 3 the forcing of Product 1 ? and are Products 4, 5, 6 used as atmospheric forcing or data assimilation in Product 3 ? please mention the links between the products.

We confirm that Product 3 (The Mediterranean Sea physical reanalysis, Escudier et al., 2021) is the forcing of Product 1 (The Mediterranean Sea biogeochemical reanalysis, Cossarini et al., 2021; Teruzzi et al., 2021) and that Product 6 (Global climate and weather analysis, ERA5, Hersbach et al., 2018) is the forcing of Product 3. On the other hand, Products 4 and 5 (Reprocessed and near real-time altimeter satellite gridded Sea Level Anomalies, SEALEVEL_EUR_PHY_L4_MY_008_068 and SEALEVEL_EUR_PHY_L4_NRT_OBSERVATIONS_008_060) are not forcing or assimilated in Product 3. In fact, Product 3 assimilates a quite different product, i.e. SEALEVEL_EUR_PHY_L3_REP_OBSERVATIONS_008_061.

We will mention the link among the products in the Section Data and Methods.

Table 2: please change "Not significant correlations…" to "Not statistically significant correlations…"

We agree.

**Reviewer#3's comments**

GENERAL COMMENTS

In this study, the authors combine various datasets (in situ and remote observations, and numerical simulations) and various variables (physical and biogeochemical data) to analyse the dissolved oxygen variability over 1999-2021 in the Southern Adriatic Sea through an EOF decomposition. In order to estimate the contribution of a set of drivers, the correlations between four first modes of variability and the drivers are computed. The study is interesting and the authors show that the dissolved oxygen is an relevant indicator of multiple drivers of the marine ecosystem.

The manuscript is very nice example of ocean data integration and provides a new relevant indicator for the Copernicus Ocean State Report. The manuscript is relatively well written although some corrections are required (but I am not English native). Indications about the datasets and computations of some derived variables are strongly missing but the authors could easily add them. Precisions on methodology are also required. The figures appear with low quality and most of them need improvements. I would recommend the publication of this manuscript after some clarifications and improvements (see my comments below).

We thank Reviewer#3 for the consideration and feedback on our manuscript. We will carefully revise it according to the suggestions. In particular, we will check the English language by means of a professional service and we will provide figures at higher resolution. We reply to the specific points about dataset, computations and methodology below.
Our answers are in blue and the proposed changes to the manuscript are in red italics.

MAJOR COMMENTS

Period of bias-correction

Could the authors explain why applying a bias correction using profiling floats over 2014.-2020? Why not over the whole period? Why excluding the last year? Which is the impact of such temporal sub-sampling in bias-correction on the bias-corrected reanalysis time series? In addition, fig. 1c shows several long temporal gaps? Which impacts?

We used all BGC-Argo float data available in a qualified mode (i.e., delay mode after a PI quality check analysis) at the time of the manuscript preparation. In particular, in 2021 only few profiles were available and, thus, we excluded them from the analysis.

Anyway, the available float profiles cover all seasons, i.e., potentially reproduce the typical annual dynamics of the oxygen in the water column (i.e. alternating mixing/stratification cycle at surface, subsurface oxygen maximum feature) and the distribution of oxygen values on a basis of four years, and the bias correction procedure has been applied considering the model output in the same days in which observations were available (i.e., there was a direct correspondence between model outputs and observations).

The temporal gaps that can be noted in Fig. 1c did not allow us to have a complete reference for the multiannual dynamics in 1999-2019, but, as we have replied to Reviewer#2, we tested

different bias correction procedures and also validated the output of the chosen method (i.e. Quantile Mapping) obtaining encouraging results.

Analysis and results

In the introduction, the authors state that they analyse the DO interannual variability (l.60). But in section 3., they also dedicate a sub section in the 2021 anomaly. Why? Because of the OSR7 that also focuses on 2021 event? Or this year has a specificity highlighted thanks to the analyses over 1999-2021? Please, clarify.

The analysis conducted for the 2021 year is actually a request for the Copernicus Ocean State Report 7 (OSR7). In fact, according to the OSR7 guidelines: (i) the core-period to be covered is 1993-2021 or earlier/later, depending on product availability and limitations; (ii) the inclusion of data during the year 2021 is mandatory (see for example the explanation of the scheme of OSR in https://marine.copernicus.eu/access-data/ocean-state-report/ocean-state-report-6) .

In our case, (i) we started the analysis from 1999, since Mediterranean biogeochemical reanalysis is available from that year, and (ii) we analysed 2021 year with respect to the 1999-2020 climatology providing a discussion of the 2021 anomaly with respect to the climatology.

Correlation

- I don't agree with the level of correlation in general. For example, 0.5 is not a high correlation. Please rephrase/moderate/modify your statements (l.117, l.123).

  We agree. We will modify the statement in the revised manuscript, also accounting for some suggestions by Reviewer#2 (please see page 18 of this reply).

- Could you detail the method used (and provide reference) for testing the significance of the correlation coefficient?

  We used the function *corrcoef* in Matlab, that provides as output both (Pearson) correlation coefficients and p-values for testing the hypothesis (parametric t test) that there is no relationship between the observed phenomena (null hypothesis). If the p-value is smaller than the significance level (that we fixed equal to 0.05), then the corresponding correlation coefficient is considered significant. We will specify the test of significance in the revised manuscript.

Datasets

All products have to be described (briefly) in section 2., not only prod1 and prod2. For each driver of the study, indicate which product is used and how the indicator is computed (in particular, MLD, which criteria is used).

We will add such information in the revised manuscript.

In addition, it is important to provide to complete references and DOIs (as indicated in the Copernicus Marine Service website, how to cite: https://help.marine.copernicus.eu/en/articles/4444611-how-to-cite-or-reference-copernicus-marine-products-and-services).  For the dataset, do not write "the latest" in the text. I recommend

to provide the complete references and to indicate the date of access in the references associated with dataset [access Month Day, Year]. Please find below the information:

Prod1: The Mediterranean Sea biogeochemical reanalysis at 1/24° of horizontal resolution and daily temporal resolution (Cossarini et al., 2021, Teruzzi et al., 2021)

- Cossarini, G., Feudale, L., Teruzzi, A., Bolzon, G., Coidessa, G., Solidoro C., Amadio, C., Lazzari, P., Brosich, A., Di Biagio, V., and Salon, S., 2021. High-resolution reanalysis of the Mediterranean Sea biogeochemistry (1999-2019). Frontiers in Marine Science.
- Teruzzi, A., Feudale, L., Bolzon, G., Lazzari, P., Salon, S., Di Biagio, V., Coidessa, G., & Cossarini, G. (2021). Mediterranean Sea Biogeochemical Reanalysis INTERIM (CMEMS MED-Biogeochemistry, MedBFM3i system) (Version 1) Data set. Copernicus Monitoring Environment Marine Service (CMEMS) https://doi.org/10.25423/CMCC/MEDSEA_MULTIYEAR_BGC_006_008_MEDBFM3I (accessed November 17, 2022)

Prod2: The Mediterranean Sea in situ quality-controlled observations, distributed by Copernicus In Situ TAC

https://doi.org/10.48670/moi-00044

Prod 3: The Mediterranean Sea physical reanalysis at 1/24° of horizontal resolution and daily temporal resolution (Escudier et al., 2021)

Escudier, R., Clementi, E., Omar, M., Cipollone, A., Pistoia, J., Aydogdu, A., Drudi, M., Grandi, A., Lyubartsev, V., Lecci, R., Cretí, S., Masina, S., Coppini, G., & Pinardi, N. (2020). Mediterranean Sea Physical Reanalysis (CMEMS MED-Currents) (Version 1) Data set. Copernicus Monitoring Environment Marine Service (CMEMS). https://doi.org/10.25423/CMCC/MEDSEA_MULTIYEAR_PHY_006_004 (accessed November 17, 2022)

Prod4 and prod 5:Reprocessed and near real-time altimeter satellite gridded Sea Level Anomalies (SLA) computed with respect to a twenty-year 1993, 2012 mean (prod4 and prod5, respectively). The product gives additional variables (i.e. Absolute Dynamic Topography and geostrophic currents).

Prod4: DOI (product): https://doi.org/10.48670/moi-00141 (accessed November 17, 2022)

Prod 5: DOI (product): https://doi.org/10.48670/moi-00142 (accessed November 17, 2022)

Prod6: Global climate and weather analysis (ERA5, Hersbach et al., 2018) with ¼º of horizontal resolution and hourly/monthly (?) temporal resolution.

We thank Reviewer#3 for these suggestions and we will indicate the complete references and DOIs in the revised manuscript.

Figures

All the figures have low resolution and too small characters (xlabel, ylabel and colorbar). Please improve and enlarge the characters of all the figures.

We will provide figures at high resolution and higher readability in the revised manuscript.

MINOR COMMENTS

In my opinion, the abstract is too much detailed (in particular with percentage of variance, correlation numbers).

We agree that the abstract should be not too detailed and we will delete the correlation number from the text (line 17 of the submitted manuscript) in the revised manuscript. On the other hand, we would prefer to leave the percentage of variance, to show that the first four modes capture a great part of variance (i.e., 94%), and that mode 4 explains a percentage lower than 10%.

In all the manuscript:

- Replace all "associated to" by "associated with"

We agree.

- Replace "Levantine/Modified Atlantic Waters" by "Levantine and Modified Atlantic Waters"

We agree with the formal correction. However, we will indicate Modified Atlantic Waters simply as "Atlantic Waters" in the revised manuscript, following the guidelines on Mediterranean water mass acronyms formulated at 36th CIESM Congress (2001). We will apply this correction anywhere else in the manuscript.

- I would write in situ (in italic, and without "–")

We agree.

- Avoid "/" in the text. To be replaced by a word.

We agree. In particular, we propose to replace "past/future" by

*past and future*

(line 71 of the submitted manuscript), "mixing/stratification" by

*mixing and stratification*

(line 88), "production/consumption" by

*production and consumption*

(line 138).

- In the text, do not indicate "first/second/etc column". Only reference to the figure

We agree. In particular, we propose to replace "Fig. 3 (left and right columns, respectively)" by:

*Figs. 3a,c,e,g and Figs. 3b,d,f,h, respectively*

(line 110) and "first column in Fig.3" and "first column of Fig. 3" by

*Figs. 3a,c,e,g*

at line 112 of the text and in the caption of Table 2 (line 387), respectively.

Abstract

l.14: Precise Mediterranean Sea reanalysis

We agree.

l.15: satellite chlorophyll concentration

We agree.

l.15: I would prefer profiling floats that Argo floats

We used the expression "Argo floats" to explicitly make reference to the Argo program (https://argo.ucsd.edu/), whose data in the Mediterranean Sea are delivered by Copernicus In Situ Thematic Assembly Centre. Therefore, we would like to maintain such a reference, unless Reviewer#3 still suggests replacing it.

l.16 and 20: of total variance

We agree.

l.23: biological production?

Yes. We will add the term "biological" as suggested.

l.31: I would replace "i.e." by "such as"

We will replace the term as suggested and, more in general, we will have the English language of the manuscript revised by a professional service.

Introduction

- Add the reference to IPPC 2021?

  We thank Reviewer#3 for this suggestion. We will add the reference "Pörtner et al., 2019" in the text (lines 33-34):

  *Indeed, DO is currently investigated under the global warming scenarios by climate and marine ecological scientific communities (e.g. Pörtner et al., 2019 …*

  and in the bibliography:

  *Pörtner, H. O., Roberts, D. C., Masson-Delmotte, V., Zhai, P., Tignor, M., Poloczanska, E., & Weyer, N. M. The ocean and cryosphere in a changing climate. IPCC Special Report on the Ocean and Cryosphere in a Changing Climate. Cambridge University Press, Cambridge, UK and New York, NY, USA, 755 pp., https://doi.org/10.1017/9781009157964, 2019.*

- 38: I would delete "Despite being a marginal Sea". Not necessary and "despite" introduce a negative aspect…

  We agree. We will delete such an expression.

- 2nd paragraph: the authors could reorganize the geographical description: first the Adriatic Sea, secondly the Southern part.

  We thank Reviewer#3 for this comment, that allowed us to improve the text of the manuscript.

In this point of the submitted manuscript, we introduced before the Southern Adriatic Sea and after the Adriatic Sea since the Southern Adriatic Sea (i.e. the domain under study) is an example of an area in which "oceanic processes connect the surface with deep layers", as written in the previous sentence. However, we recognise that the text should be revised to make it clearer. We propose to modify this part as:

*making this parameter of primary interest especially in those areas where oceanic processes connect the surface with deep layers.*

*The Southern Adriatic Sea (SAdr, Fig. 1a) is one of these areas, since it is a site of deep water formation (Gačić et al., 2002; Pirro et al., 2022), which has a crucial importance for the Eastern Mediterranean Sea ventilation.*

53:" Marine Strategy Framework Directives (MSFD)" rather than "sensu MFSD"

We agree.

- 53: refer to IPPC 2021?

We thank Reviewer#3 for this suggestion. We will add the citation "Pörtner et al., 2022":

*to understand anthropogenic impacts on marine environment (Pörtner et al., 2022).*

and in the bibliography:

*Pörtner H.O. , Roberts D.C. , Tignor M., Poloczanska E.S. , Mintenbeck K., Alegría A. , Craig M., Langsdorf  S., Löschke S., Möller V., Okem A., Rama B. Climate Change 2022: Impacts, Adaptation and Vulnerability. Contribution of Working Group II to the Sixth Assessment Report of the Intergovernmental Panel on Climate Change. Cambridge University Press. Cambridge University Press, Cambridge, UK and New York, NY, USA, 3056 pp., doi:10.1017/9781009325844, 2022*

- 54: replace "Marine Strategy Framework Directives" by " MSFD"

We agree.

- 57: replace "i.e." by "distributed by the"

We propose to modify this part as:

*the present work integrates the state-of-the-art approach of in situ measurements (in 2014-2020, distributed by Copernicus In Situ TAC)*

- 58: reanalysis in the Mediterranean Sea

We agree.

- 58: I would replace "at 1/24º horizontal resolution" by "high-resolution" (such precision is given in section 2)

We agree. We will replace the suggested expression.

Data and method

See my previous comment concerning the dataset descriptions (references and DOIs) + all datasets have to be introduced.

For sure we will account for it.

- 65: reanalysis in the Mediterranean Sea

We will modify this part as:

*by combining data from Copernicus reanalysis in Mediterranean Sea*

- 65: replace "in" by "over"

We agree.

- 66 I would replace "in the time" by "over the"

We agree.

- 67: replace CMEMS by Copernicus

Please see our reply to the next comment.

- I suggest to rewrite the beginning of this section, combining l.64-66 and 67-69 (to avoid repetitions)

We thank Reviewer#3 for this comment, that allows us to improve the text of the manuscript. We would prefer to leave at first a general sentence about the integration of in situ data and modelled ones and then another sentence introducing more details about the method. Nevertheless, we recognise that there are some repetitions. We propose to modify the part at lines 67-69 of the submitted manuscript as:

*In particular, we employed the BGC-Argo float measurements of in situ DO to compute a bias correction to the daily DO concentration simulated by the biogeochemical reanalysis at 1/24° horizontal resolution.*

- 70: what is the reference to ThemeXX et al.?

We thank Reviewer#3 for having noticed this oversight. In fact, we forgot to indicate the reference in the bibliography and we will add it in the new version of the manuscript:

*Themeßl J., M., Gobiet, A. and Leuprecht, A. Empirical-statistical downscaling and error correction of daily precipitation from regional climate models. Int. J. Climatol., 31: 1530-1544. https://doi.org/10.1002/joc.2168, 2011.*

- 71: past/future: past and future

We will replace this expression by:

*past or future*

- 89: delete "the"

We agree.

- 90 and 96: at surface and in subsurface, (removing level)

We agree.

- 94 Levantine Water should be Levantine Intermediate Water (LIW)

Here we indicated a more general "Levantine Water", since it actually includes Levantine Intermediate Water, Levantine Surface Water,  Surface Ionian Water and Cretan Intermediate Water (Civitarese et al., 2010; Schroeder et al., 2017; Table 2 in Menna et al., 2021). Therefore, we would prefer to maintain this expression, or, if Reviewer#3 suggests us to be more precise, to include the three water masses.

Schroeder, K.; Chiggiato, J.; Josey, S.A.; Borghini, M.; Aracri, S.; Sparnocchia, S. Rapid response to climate change in a marginal sea. Sci. Rep. 7, 4065, https://doi.org/10.1038/s41598-017-04455-5, 2017.

Menna, M., Gerin, R., Notarstefano, G., Mauri, E., Bussani, A., Pacciaroni, M., & Poulain, P. M. On the Circulation and Thermohaline Properties of the Eastern Mediterranean Sea. Frontiers in Marine Science, 903, https://doi.org/10.3389/fmars.2021.671469, 2021.

103: could you explain why 13 months?

The 13-month moving average is applied in order to remove the seasonal and intra-annual variations and it is a commonly done procedure to highlight the sense of rotation of the Northern Ionian Gyre (NIG), i.e., cyclonic or anticyclonic (e.g. Menna et al., 2019). The use of 13 months instead of 12 is related to the use of a centered average, since odd orders (i.e., number of periods, here years, over which the moving average is calculated) allows to avoid phase shift phenomena in the smoothed time series.

Menna, M., Suarez, N. R., Civitarese, G., Gačić, M., Rubino, A., & Poulain, P. M. Decadal variations of circulation in the Central Mediterranean and its interactions with mesoscale gyres. Deep Sea Research Part II: Topical Studies in Oceanography, 164, 14-24, https://doi.org/10.1016/j.dsr2.2019.02.004, 2019.

110: "vertical pattern" rather than "spatial pattern"

We agree.

Results

- I suggest to delete the 3.1 Subtitle and move this paragraph in section 3.2. Then, Section 3.2 will become 3.1, and 3.3 will become3.2.

We thank Reviewer#3 for this suggestion. We will modify the structure of the manuscript as suggested.

- l.115: replace "almost 50%" by "48.9%"

We agree.

- l-119: 20%, be precise here, indicate 19.7 and 17.7 % respectively. In general, be more precise in section 3. results

We agree.

-l.123-125: such results raise the questions about the dependency between the drivers. I may be not a problem. Do you have an opinion about that?

The analysis of the correlation among time series of EOFs and meteo-marine drivers gave worthwhile information about dynamics influencing oxygen concentration in the Southern Adriatic pit, but drivers acting on the oxygen concentration in the area are unavoidably interrelated.

In the specific case of mode 3, to which lines 123-125 are referred, it partly explains the intensity of the vertical gradient between surface and subsurface layers (Fig. 3f). Thus, the intensity of the gradient is influenced by drivers impacting either the surface and subsurface dynamics.

-l.138: multiple modes

We agree.

- l. 140: where the seasonal signal is strong, you could remove it to better highlight and quantify the interannual variability, no? Have you test it?

As we replied to Reviewer#2 (page 17), we have actually tried to identify all temporal components present in the profile time series (i.e. Hovmöeller diagram) of oxygen concentrations by EOF decomposition, rather than conduct a classical time series analysis (e.g., X-11 method after de-seasonality; Colella et al., 2016).

This is because the time series shows a cyclicity on a time period of some years (as also noticed by Reviewer#1) and thus we believed that EOF analysis on the not deseasonalized time series should be the most appropriate approach to reveal different frequencies of the time series signal (e.g., seasonal and multiannual). In fact, we considered that, on one hand, the seasonality would have emerged in the EOF time series and, on the other hand, that we could verify if multiannual cyclicity could have affected the seasonal cycle.

Colella, S., Falcini, F., Rinaldi, E., Sammartino, M., & Santoleri, R. (2016). Mediterranean ocean colour chlorophyll trends. PloS one, 11(6), e0155756.

2021 Anomaly

l.152: r=?

As we also indicated in our replies to Reviewer#1 (page 6) and Reviewer#2 (page 21), we will clarify this sentence as:

*The first EOF temporal mode (Fig. 3a) is actually negative from 2019, and it corresponds to the negative anomaly of one of its drivers (Table 2), i.e. subsurface chlorophyll, and not heat fluxes. In particular, we estimated a 2021 mean negative anomaly approximately equal to 6% with respect to the climatological mean (1999-2020) for subsurface chlorophyll.*

1. 152 "seems to be connected"… coincide with? With what it is written in the literature?

   This sentence illustrates a first hypothesis about the 2021 anomaly, that should be carefully verified and monitored in future works. As replied also to Reviewer#1 and Reviewer#2, we did not enter in more details on this part in the submitted manuscript because of the short length of the paper (the recommended number of figures and words of the paper was indicatively limited to a maximum of 4 figures and 3000 words). However, we recognise that the paper can be improved by providing more details on this particular aspect (i.e., the link of this 2021 oxygen anomaly with the already observed and described

incoming of saltier Levantine water).  Thus, we will add in appendix a figure on salinity and temperature through the Otranto Strait and we will summarise the results of previously published papers (e.g., Menna et al., 2022) where the salinity evolution in the Southern Adriatic pit in the latest years has been investigated.

2. 159: remove "latest" (see my comment above)

   We agree.

l.160: rephrase

If Reviewer#3 refers to the mention of 2021 year analysis in addition to the interannual variability characterisation, we agree and we will add the focus on 2021 in the revised manuscript.

l.161: long-term

We agree.

l.165: that we conducted ?

We agree.

Figures

See my comments above.

We will account for them in the revised manuscript.

Fig 3: add the 0-value line in all panels (a,c,e,g). In legend replace "spatial" by "vertical", and ",8.4%" by "and 8.4%".

We agree with the changes indicated between quotation marks.

On the other hand, in Fig. 3 we have already indicated the 0-value line in a,c,e,g panels and maybe it can be a problem of graphical visualisation. Anyway, we will provide a new version of Fig. 3 at higher quality in the revised manuscript.

Fig 4: legend: "climatological mean […] in 1999-2020 " could be replace by "Mean over 1999-2020" and "reference period" by "1999-2020 period".

We agree.

Table 1: see my previous comments

We will account for them in the revised manuscript.

Table 2: EOFs of DO

We agree.

---

## Author Response (AR1)

**Authors' reply to the Editor**

Dear Valeria Di Biagio and co-authors,

I have read you answers to the reviewers comments and based on that I invite you to proceed with the preparation of a revised version of your work along the lines mentioned in your answer, addressing point by point the reviewers' comments including a mark-up copy and detailed description of the changes made.

Thank you so much for your efforts,

Kind regards,

Marilaure Grégoire
* * *
Dear Marilaure Grégoire,

Thank you for your consideration and feedback on our submitted manuscript. We have carefully considered all the reviewers' comments and addressed them point by point.

In particular, we addressed two major topics, namely the high salinity observed in the southern Adriatic Sea in the last years (for which all reviewers suggested more information) and the bias correction method used (for which Reviewer#2 recommended a more detailed discussion). In fact, we added two appendices to the revised manuscript:

- Appendix A, which contains the Hovmöller diagrams of dissolved oxygen before and after the Quantile Mapping bias correction and the frequency histograms of the values associated with the observations and the model;
- Appendix B, which shows the time series of temperature and salinity in the Otranto Strait and supports our interpretation about the negative anomaly in oxygen in 2021, which was discussed in more detail in the main text of the manuscript (Section 3.2).

In our responses we also added further discussion about all individual comments.

In addition, we included new versions of all figures, that take into account Reviewer#3's suggestion about character size and Reviewer#2's suggestion about the addition of the bathymetry in Fig. 1a.

In this document, we indicate the reviewers' comments in black, our responses in blue and the corrections we made to the text of the manuscript in italic red. Line numbers refer to the revised version of the manuscript in the track-change version.

Best regards,

Valeria Di Biagio and co-authors

**Response to Reviewer#1's comments**

This paper presents an analysis of the interannual variability of dissolved oxygen in the Southern Adriatic Sea, based on analysis of the CMEMS Mediterranean reanalysis for 1999-2021, bias corrected using in situ observations made by BGC-Argo floats. EOF analysis is used to separate the main components of variability, which are then linked to different drivers. The analysis is thorough and the results are novel and interesting, but a few points are worth attention before publication.

We thank Reviewer#1 for the consideration and evaluation of our manuscript. We have carefully revised it according to the suggestions. Our answers are indicated in blue and the changes to the manuscript are in red italics. Line numbers refer to the revised version of the manuscript in the track-change version.

I am confused by the use of the terms temporal mode and spatial mode. The description in the methods section implies to me that the EOF analysis has been applied separately to identify variability over (vertical) space and variability over time – this would give a different set of modes in each case. But in the results the two sets are presented together, with the "spatial modes" being used to show the vertical variation seen in the "temporal modes". Wouldn't it be more accurate to refer to spatial and temporal aspects of the four modes? Or have I misunderstood the method? I suggest that either a fuller description of the methods or a revision of the language used in the results would help.

We thank Reviewer#1 for this comment, that allowed us to clarify the used terminology and the EOF analysis.

In general, the terminology relevant to such a statistical method is not universally agreed: in literature EOF analysis is also indicated as Principal Component Analysis or Eigen Analysis, and also the temporal and spatial functions involved in the computations can be defined in different ways. However, the method itself is well defined (e.g., Thomson and Emery (2014), also cited in the manuscript) and widely used by the scientific community.
Following the notation in Thomson and Emery (2014), the EOF analysis allows to decompose a spatio-temporal function $\psi$ (referring to *M* spatial locations) in a combination of orthogonal spatial functions $\phi$, whose amplitudes are weighted by time-dependent coefficients $a$ which are uncorrelated over the sample data:

$$\psi(x_m, t) = \psi_m(t) = \sum_{i=1}^{M} [a_i(t)\phi_{im}]$$

Finding $a$ and $\phi$ is equivalent to solve an eigenvalue problem, with $\phi_{im}$ eigenvectors and $\lambda = \overline{a_i^2(t)}$ eigenvalues, corresponding to the variance associated with each eigenvector. In Thomson and Emery (2014), $\phi_{im}$ are called "statistical modes" or "spatial modes" and $a_i(t)$ are called "time amplitudes" of the *i*th statistical mode.

In the manuscript, we have used the term "spatial modes" in such a sense, i.e. referred to $\phi_{im}$. Moreover, we used the expression "temporal modes" for the time amplitudes $a_i(t)$, as done in other studies (e.g., Baldacci et al., 2001; Alvera-Azcárate et al., 2009; Martellucci et al., 2021). However, we recognise that a more accurate terminology can help to better illustrate method and results.

Therefore, following Reviewer#1's suggestion, and also integrating the suggestion by Reviewer#3 of replacing "spatial" by "vertical", in the revised manuscript we used the expressions: "EOF vertical patterns", "EOF time series", "time series of the first/second/… mode", as done in Baldwin (2001), Espinosa-Carreon et al. (2004), Folland et al. (2009), Christoudias et al. (2012), Lee et al. (2020).
In particular, we modified the definition at lines 108-111 as:

*The EOF analysis allows us to identify the spatial patterns of variability (i.e., EOF vertical patterns), describe how they change in time by means of time series (i.e., EOF time series), and associate the explained variance with each mode.*

the caption of Fig. 3 as:
*Figure 3: EOF time series (a, c, e, g) and vertical patterns (b, d, f, h) of the first four modes computed on the bias-corrected dissolved oxygen concentration in the southern Adriatic area shown in Fig.1d. The explained variances of the four modes are: 48.9%, 19.7%, 17.7% and 8.4%.*

and changed the text accordingly.

Alvera-Azcárate, A., Barth, A., Sirjacobs, D., and Beckers, J.-M.: Enhancing temporal correlations in EOF expansions for the reconstruction of missing data using DINEOF, Ocean Sci., 5, 475–485, https://doi.org/10.5194/os-5-475-2009, 2009.

Baldacci, A., Corsini, G., Grasso, R., Manzella, G., Allen, J.T. , Cipollini, P., Guymer, T.H. ,Snaith, H.M. A study of the Alboran sea mesoscale system by means of empirical orthogonal function decomposition of satellite data, Journal of Marine Systems, Volume 29, Issues 1–4, Pages 293-311, ISSN 0924-7963, https://doi.org/10.1016/S0924-7963(01)00021-5, 2001.

Baldwin, Mark P. Annular modes in global daily surface pressure, Geophysical Research Letters, 28, 21 - 0094-8276, https://doi.org/10.1029/2001GL013564, 2001.

Christoudias, T., Pozzer, A., and Lelieveld, J.: Influence of the North Atlantic Oscillation on air pollution transport, Atmos. Chem. Phys., 12, 869–877, https://doi.org/10.5194/acp-12-869-2012, 2012.

Espinosa-Carreon, T. L., Strub, P. T., Beier, E., Ocampo-Torres, F., and Gaxiola-Castro, G., Seasonal and interannual variability of satellite-derived chlorophyll pigment, surface height, and temperature off Baja California, *J. Geophys. Res.*, 109, C03039, doi:10.1029/2003JC002105, 2004.

Folland, C. K., Knight, J., Linderholm, H. W., Fereday, D., Ineson, S., & Hurrell, J. W. The summer North Atlantic Oscillation: past, present, and future. Journal of Climate, 22(5), https://doi.org/10.1175/2008JCLI2459.1, 1082-1103, 2009.

Lee, G., Ho, C. H., Chang, L. S., Kim, J., Kim, M. K., & Kim, S. J. Dominance of large-scale atmospheric circulations in long-term variations of winter PM10 concentrations over East Asia. Atmospheric Research, 238, 104871, https://doi.org/10.1016/j.atmosres.2020.104871, 2020.

Martellucci, R., Salon, S., Cossarini, G., Piermattei, V., & Marcelli, M. Coastal phytoplankton bloom dynamics in the Tyrrhenian Sea: Advantage of integrating in situ observations, large-scale analysis and forecast systems. Journal of Marine Systems, 218, 103528, https://doi.org/10.1016/j.jmarsys.2021.103528, 2021.

I am also uncertain why 2021 is picked out for particular discussion. In Figure 1d and Figure 3 it looks to me like the continuation of a trend that goes back to 2016, rather than an unusual year – why focus on this year in particular? It also appears to have some similarity to 1999 – is there some possibility of a cyclical pattern? I don't think that a major reworking of the analysis is needed, but I suggest the authors consider a shift in the way they present the 2021 results. The reference to the entrance of new water masses from the Levantine Basin is interesting – was this the reason for looking at 2021? It would be good to see more detail than is given in lines 155-157 about how unusual this change is.

The analysis conducted for the 2021 year is actually a request for the Copernicus Ocean State Report 7 (OSR7). In fact, according to the OSR7 guidelines: (i) the core-period to be covered is 1993-2021 or earlier/later, depending on product availability and limitations; (ii) the inclusion of data during the year 2021 is mandatory (see for example the explanation of the scheme of OSR in https://marine.copernicus.eu/access-data/ocean-state-report/ocean-state-report-6) .

In our case, (i) we started the analysis from 1999, since Mediterranean biogeochemical reanalysis is available from that year, and (ii) we analysed 2021 year with respect to the 1999-2020 climatology providing a discussion of the 2021 anomaly with respect to the climatology. Given this preamble, we agree that the analysis of 2021 needs additional details and to be included in a more general discussion of the temporal dynamics of the Southern Adriatic Sea.

In particular, regarding a possible cyclical pattern (i.e., alternation of periods with low and high values of oxygen), it is known that the Southern Adriatic Sea displays quite periodical behaviours on multiannual scales associated with the reversal of the Northern Ionian Gyre (e.g., Civitarese et al., 2010) regulating the waters entering from Ionian Sea. Moreover, focusing on the vertical pattern of oxygen concentration during the year, Southern Adriatic pit is an open-ocean convection site, characterised by different phases of the convection (preconditioning, water mixing, spreading). In our case, during the convection period the surface oxygen is mixed down to the basis of the mixed layer depth, producing the high values of concentration observed in the subsurface layers (Fig. 1c) and destroying the previous vertical structure of the water masses. After the convection, the newly formed water spreads into the Ionian Sea, replaced by the less oxygenated water from the Ionian Sea, restoring the typical vertical profile (i.e., a profile characterised by a surface minimum, a subsurface maximum and a minimum in deeper layers). The inflow of the Northern Adriatic dense Water (NAddW) can furtherly increase the oxygen content in the Southern Adriatic pit, in the deepest layers of the pit due to the "saw tooth" mechanism (involving alternating long-lasting mixing processes and sudden density increases due to the intrusion of very dense Northern Adriatic water, e.g. in 2012, Querin et al., 2016) and at intermediate depths due to more sporadic events, like the double salinity maximum, as indicated in Kokkini et al., 2020 for 2015-2016 years.

Also following similar suggestions by Reviewer#2 and Reviewer#3, we added at the beginning of the section Results (lines 139-141) a brief comment on this cyclicity, being the investigation of the drivers the object of the correlation analysis with EOF time series:

*Dissolved oxygen in the southern Adriatic area (Fig. 1a) shows in the subsurface layers an alternation between periods of enrichment (in 2004-2006, 2010-2013, 2016-2017) and sharp declines that impacted the Oxygen Minimum Layer (OML), located between 100 and 300 m. Low concentration values are observed also in the years between 1999 and 2003.*

Considering 2021 as requested from OSR guidelines, its behaviour was only partially explained by our analysis conducted with EOF and correlation with drivers. In fact, the negative anomaly of oxygen, associated with the negative anomaly of the EOF first mode, is only correlated with one of its drivers (i.e., negative anomaly in subsurface chlorophyll). In addition, in the last 3 years we observed the entrance of water masses that are much saltier than usual (as reported in Fig. R1.1 and in Mihanović et al., 2021; Menna et al. 2022b, as indicated e.g. in Fig. R1.2) and we suppose that a regime shift is ongoing and requires further future monitoring, as mentioned in the Discussion section.

We did not enter in more details on this part because our analysis was mainly focused on the identification and analysis of the principal modes of variability displayed by the area and also because of the short length of the paper (the recommended number of figures and words of the paper was indicatively limited to a maximum of 4 figures and 3000 words). A more complete investigation on the origin of such waters will be the object of future work.

However, we agree with Reviewer#1 that the paper can be improved by providing more details on this particular aspect. Thus, including also a suggestion by Reviewer#2, we showed the time series of temperature and salinity at the surface and intermediate layer through the Otranto Strait in a new appendix and summarised more clearly the previously published results, to support such a preliminary hypothesis about the strong anomaly starting from 2019.

In particular, in the "Appendix B: Time series of surface and intermediate temperature and salinity at the Otranto Strait" we included the following figure:

[Figure]

*Figure B1: Time series of temperature (a) and salinity (b), averaged in the vertical layers 0 - 150 m (red lines) and 150-600 m (black lines) of the Otranto Strait (39.8°N, 18.5° - 19.5° E) in the 1999-2021 time period. In the top panel, light red and dark red indicate data before and after de-seasonalization, respectively. Data are provided by Copernicus physical reanalysis (Prod3, Table 1).*

Moreover, in the revised manuscript we replaced the lines 153-157 of the submitted version by this part (lines 198-211):

*One of the causes of the decrease in total oxygen concentration in the SAdr could be due to the exceptional salinization observed in the SAdr since 2017 (Mihanović et al., 2021, Menna et al., 2022b). This increase was related to the inflow of new, warmer and noticeably saltier water masses from the northeastern Ionian Sea (Mihanović et al., 2021, Menna et al. 2022b).*
*The inflow of saltier and warmer water masses is also evident by observing the temporal evolution of these parameters through the Strait of Otranto (Fig. B1). In particular, in the upper layer (0-150 m) both temperature and salinity show an overall positive trend throughout the period 1999-2021, whereas the decrease observed in 2006-2011 and 2017-2018 can be associated with the inflow of less saline AW, triggered by the anticyclonic circulation of the NIG (Fig. 2f). In the intermediate layer (150-600 m), salinity shows a positive trend in 1999-2021, while no clear trend is observed for temperature. Moreover, a sharp increase in salinity (~ 0.1) is observed in 2019. This increase occurred after the NIG inversion from anticyclonic to cyclonic (Fig. 2f), resulting in a further increase in salinity due to both the decrease in AW advection and the increase in LIW inflow.*

And we better specified in the Abstract (lines 26-27):
*we observe a sharp increase in salinity favoured by …*

Civitarese, G., Gačić, M., Lipizer, M., & Eusebi Borzelli, G. L.: On the impact of the Bimodal Oscillating System (BiOS) on the biogeochemistry and biology of the Adriatic and Ionian Seas (Eastern Mediterranean). Biogeosciences, 7(12), 3987- 3997. https://doi.org/10.5194/bg-7-3987-2010, 2010.

Kokkini, Z., Mauri, E., Gerin, R., Poulain, P. M., Simoncelli, S., & Notarstefano, G. (2020). On the salinity structure in the South Adriatic as derived from float and glider observations in 2013–2016. Deep Sea Research Part II: Topical Studies in Oceanography, 171, 104625.

Mauri E., Menna M., Garić R., Batistić M., Libralato S., Notarstefano G., Martellucci R., Riccardo Gerin R., Pirro A., Hure M., Poulain P.M. in von Schuckmann et al., Copernicus Marine Service Ocean State Report, Issue 5, Journal of Operational Oceanography, 14:sup1, 1-185, DOI: 10.1080/1755876X.2021.1946240, 2021.

Menna M., Martellucci R., Notarstefano G., Mauri E., Gerin R., Pacciaroni M., Bussani A., Pirro A., Poulain P.M. Record-breaking high salinity in the South Adriatic Pit in 2020 in Copernicus Ocean State Report, issue 6, Journal of Operational Oceanography, 15:sup1, 1-220, DOI: 10.1080/1755876X.2022.209516, 2022b.

Mihanović, H., Vilibić, I., Šepić, J., Matić, F., Ljubešić, Z., Mauri, E., ... & Poulain, P. M.: Observation, Preconditioning and Recurrence of Exceptionally High Salinities in the Adriatic Sea. Frontiers in Marine Science, 8, 834. https://doi.org/10.3389/fmars.2021.672210, 2021.

[Figure]

**Figure R1.1: Hovmöller diagram of seawater salinity east of Otranto Strait in the period 1999-2021 according to Mediterranean physical reanalysis (Prod3 indicated in the manuscript).**

[Figure]

**Fig. R1.2: T-S diagram of water in the southern Adriatic Pit during the 2013-2020 period (colour scale indicates time) according to BGC-Argo float observations, from Menna et al., 2022b (Figure 4.7.3.d of the reference).**

**Specific points**

line 19, 46: I don't understand "alternate" in the description of the North Ionian Gyre.

The Northern Ionian Gyre is a circulation structure that shows a reversal from cyclonic to anticyclonic and vice versa at multiannual scales (e.g., Civitarese et al., 2010; Menna et al., 2019, Gačić et al. 2021). In particular, circulation was cyclonic in 1999-2005, 2011-2016, 2019-2021 and anticyclonic in 2006-2010, 2017-2019. It can be also visualised from the vorticity time series in Fig. 2f, where positive (negative) vorticity indicates cyclonic (anticyclonic) circulation. We added more details about Northern Ionian Gyre circulation in the Introduction of the revised manuscript.
In particular, we replaced the sentence at lines 49-52 with (lines 52-62):

*The SAdr is strongly influenced by the inflow of water masses from the northern Adriatic Sea (i.e., North Adriatic Dense Water, Querin et al., 2016) and the Ionian Sea. In particular, the inflow of southern water masses is triggered by the periodic reversal of Northern Ionian Gyre circulation (Gačić et al., 2002; Civitarese et al., 2010, Menna et al., 2019). This oscillating system, called the Adriatic - Ionian Bimodal Oscillating System (BiOS), changes the circulation of the Northern Ionian Gyre from cyclonic to anticyclonic and vice versa, modulating the advection of water masses in the Adriatic Sea (Gačić et al., 2010, Rubino et al., 2020). The cyclonic circulation of the Northern Ionian Gyre causes the advection of saline water masses of Levantine origin (i.e., Levantine Intermediate Water, Cretan Intermediate Water, Ionian Surface Water and Levantine Surface Water, Manca et al., 2006), while the anticyclonic circulation favours the inflow of Atlantic water and a relative decrease of salinity in the SAdr (Gačić et al., 2011; Menna et al., 2022a). This feature has a strong influence on the biogeochemical properties of the SAdr, affecting nutrient availability (Civitarese et al., 2010), phytoplankton blooms (Gačić et al., 2002; Civitarese et al., 2010), and species composition (Batistić et al., 2014, Mauri et al., 2021).*

and added the following references to the bibliography:

*Batistić, M., Garić, R., & Molinero, J. C. . Interannual variations in Adriatic Sea zooplankton mirror shifts in circulation regimes in the Ionian Sea. Climate research, 61(3), 231-240, 2014.*

*Gačić, M.; Borzelli, G.E.; Civitarese, G.; Cardin, V.; Yari, S.; Can internal processes sustain reversals of the ocean upper circulation? The Ionian Sea example. Geophys. Res. Lett. 37 (9) :L09608, DOI:10.1029/2009JC005749, 2010.*

*Gačić, M.; Civitarese, G.; Eusebi Borzelli, G.L.; Kovačević, V.; Poulain, P.M.; Theocharis, A.; Menna, M.; Catucci, A.; Zarokanellos, N.; On the relationship between the decadal oscillations of the northern Ionian Sea and the salinity distributions in the eastern Mediterranean.; J. Geophys. Res. Oceans, 116, C12, https://doi.org/10.1029/2011JC007280, 2011.*

*Gačić, M.; Ursella, L.; Kovačević: V., Menna, M.; Malačič, V.; Bensi, M.; Negretti, M.E.; Cardin, V.; Orlić, M.; Sommeria, J.; Barreto, R.V.; Viboud, S.; Valran, T.; Petelin, B.; Siena, G.; Rubino, A.; Impact of dense-water flow over a sloping bottom on open-sea circulation: laboratory experiments and an Ionian Sea (Mediterranean) example. Ocean Sci., 17, 975–996, https://doi.org/10.5194/os-17-975-2021, 2021.*

*Manca, B. B., Ibello, V., Pacciaroni, M., Scarazzato, P., & Giorgetti, A. Ventilation of deep waters in the Adriatic and Ionian Seas following changes in thermohaline circulation of the Eastern Mediterranean. Climate Research, 31(2-3), 239-256, 2006.*

*Rubino, A.; Gačić, M.; Bensi, M.; Kovačević, V.; Malačič, V.; Menna, M.; Negretti, M.E.; Sommeria, J.; Zanchettin, D.; Barreto, R.V.; et al. Experimental evidence of long-term oceanic circulation reversals without wind influence in the North Ionian Sea. Sci. Rep., 10, 1905, 10.1038/s41598-020-57862-6, 2020.*

line 22-23 "We ascribe the lower content of DO in 2021 to a negative anomaly of the subsurface production in the same year, in agreement with the previous correlation analysis, but not to heat fluxes": this point made in abstract is not stated explicitly in the main body of the paper (though it is implied).

We agree with Reviewer#1. We explicitly indicated this point in the revised paper in the section dedicated to 2021 anomaly (at lines 194-197) as follows:

*The time series of the first mode (Fig. 3a) is actually negative from 2019 and corresponds to the negative anomaly of only one of its drivers (Table 2), i.e. subsurface chlorophyll (Fig. 2d), and not heat fluxes (Fig. 2a). In particular, we estimated a mean negative anomaly approximately equal to 6% with respect to the climatological mean (1999-2020) for subsurface chlorophyll in 2021.*

line 23-25 "we observe the entrance of warmer and exceptionally saltier waters favored by the cyclonic circulation of NIG from 2019 onwards": cyclonic circulation is also observed in 1999-2005 and most of 2011-2016; no evidence is presented about the temperature and salinity of the water.

We thank Reviewer#1 for this comment, which allowed us to improve our manuscript.

As we indicated in our reply to the general comments, we added in Appendix B of the revised manuscript the time series of salinity and temperature at the surface and intermediate layer through the Otranto Strait in the considered period (1999-2021) and summarised more clearly the previously published results.

line 67-74: regarding the bias correction, was the bias consistent across the three periods where float data was available? Did the bias correction allow for uncertainty in the in situ observations?

We applied the bias correction as a statistical procedure over all available data. We used only delayed mode oxygen data from BGC-Argo whose Quality Control procedure is described in Product 2 (https://doi.org/10.13155/75807). The bias correction procedure does not explicitly account for uncertainty in observations.

line 97: the dimensions of the SAdr box are much smaller than the region from which DO data was taken (i.e. the 0.9 autocorrelation contour). Why was a smaller region used for the forcing indices?

We thank Reviewer#1 for raising this point that allowed us to clarify an aspect of our method.

All the boxes used to average the time series of the meteo-marine drivers were chosen in order to obtain clear signals. In particular, for mixed layer depth (and heat fluxes and chlorophyll concentrations) we needed to limit the considered area in the Southern Adriatic Sea in order to include the whole volume of the pit.

In addition, we also specified that the biogeochemical reanalysis data considered in the spatial autocorrelation analysis are dissolved oxygen, nitrate and chlorophyll concentrations at surface and, also accounting for a comment by Reviewer#2, we corrected at line 100 and in the caption of Figure 1 that the correlation analysis is actually a "cross-correlation" (between the data in the central point of the pit and the ones at every spatial gridpoint in the domain, as in Jones et al., 2015 and Martellucci at al., 2021) instead of an "autocorrelation".

We acknowledged that additional details concerning the points above should be added to the Data and methods in the revised manuscript.

In particular, we added at line 128-129:

*to consider the whole volume of the pit*

Moreover, we added at lines 101-104:

*Specifically, we considered the cross-correlation between the surface data of DO, nitrate and chlorophyll concentrations in the central point of the pit and those at each spatial grid point in the domain, to identify the area that displayed the same dynamics at the surface from a phenomenological perspective.*

and, in order to make clearer the text, we moved the sentence from lines 104-105 to 84-85 and deleted the sentence at lines 98-99, by modifying lines 94-97 as:

*In our application, we adapted the code publicly provided by Beyer et al. (2020) at https://doi.org/10.17605/OSF.IO/8AXW9 and included available in situ data of daily DO (Fig. 1c) within a representative area (Fig. 1b) of the southern Adriatic in the period 2014-2020, and DO reanalysis data for the same days of measurements.*

Jones E.M., Doblin M.A., Matear R. , King E. Assessing and evaluating the ocean-colour footprint of a regional observing system J. Mar. Syst., 143 (2015), pp. 49-61, https://doi.org/10.1016/j.jmarsys.2014.10.012

line 119 Fig.3c-e: This should be 3c,e if it's referring to the temporal modes only or 3c-f if it's referring to both temporal and spatial modes.

We agree. We clarified this part (line 157) as:

*The second and third modes (Figs. 3c-d and 3e-f, respectively)*

line 129 "the first mode explains the variability (e.g. seasonal) connected with solubility": I'm not sure why the third mode is not involved too – it has a correlation of 0.51 with heat flux and hence, presumably, to temperature and solubility.

We agree with Reviewer#1, since also the third mode has a correlation with heat fluxes in the area. However, the first mode explains a higher percentage of total variance, i.e., we ascribe the variability connected with solubility mainly with the first mode.
We slightly rephrased this part (lines 169-171) as:

*Analysing the four modes in order of decreasing explained variance, we ascribe the seasonal variability connected with solubility mainly to the first mode, whereas we associate the biological contribution to oxygen dynamics to multiple interacting modes. In fact,* …

line 167-8 "We do not recognise a clear deoxygenation trend in the subsurface layer": there seems to be a very clear decline for 2010-2021. Is the point that there is no overall decline in the period 1999-2021 (though there is a rise then a fall)?

The oxygen time series (Fig. 1d) does not show a negative trend in the subsurface layer in the whole considered period (1999-2021) and its dynamics are characterised mainly by the inter-annual variability related to the drivers, rather than to an overall trend.
In the revised manuscript, we added a description of the succession of subsurface oxygen enrichment and reduction periods (please see our reply to the general comments on pages 3-4).

Moreover, we also improved Figure 1d to favour a better readability of the figure.

line 168-169 "the multiannual variability is characterized by a sort of cyclicity": I'm not sure what is meant here.

We thank Reviewer#1 for this comment. Here we referred to the cyclical pattern (i.e., alternation of periods with low and high values) of the oxygen concentrations in the subsurface layers. In fact, concentrations in the periods 2004-2006, 2010-2013, 2016-2017 are higher than those in the other years (please see our reply to the general comments on pages 3-4 about the cyclical pattern of the oxygen concentration in the Southern Adriatic pit, that we included in the revised version of the manuscript).

To better clarify this expression, we specify at line 223-224:

*the multiannual variability is characterised by an alternation of enrichment and reduction phases*

line 170-172: I'm not sure that I understand the point being made here – is it that the SAdr responds quickly to change because of the small water volume and residence time?

We thank Reviewer#1 for this comment that allowed us to clarify the text of the manuscript. The Southern Adriatic pit is actually characterised by a short water residence time, due to its small volume and to the strength of its meteo-marine drivers. This feature allows to identify possible recurrent behaviours and also results in a rapid response to changes in the drivers themselves. We slightly rephrased this part in the revised paper (lines 225-229) as:

*The possibility to observe such cyclic signals is enhanced by the relatively small volume and short residence time of the SAdr pit waters (Querin et al., 2016) with respect to other Mediterranean areas (Coppola et al., 2018). This feature makes the SAdr a potential efficient probe to detect a rapid response to changes in its meteo-marine drivers, i.e., circulation and atmospheric patterns.*

Overall I think this is a good paper and I enjoyed reading it. But I advise the authors to go through the text carefully to make sure that their main findings are presented really clearly and backed up by enough information. I also recommend an English language edit – the paper is generally well written but in a few places I was unclear about the meaning.

We thank Reviewer#1 for the feedback. We modified the manuscript as indicated in our replies and carefully checked the English language by means of a dedicated tool.

**Response to Reviewer#2's comments**

The manuscript investigates the interannual variability of dissolved oxygen (DO) over the last 23 years driven by multiple mechanisms (atmospheric, circulatory, and biological processes) in the southern Adriatic Sea. The aim is to demonstrate the importance of DO as indicator of current changes and environmental status.

The study is based on modelled and observational data from the Copernicus Marine Service: the biogeochemical reanalysis for the Mediterranean Sea covering the 1999-2021 period and BGC-Argo float measurements for the 2014-2020 period. DO estimated by BGC-Argo floats are used to bias-correct the modelled DO along the entire reanalysis time series using a quantile mapping technique.

The bias-corrected reanalysis signal is then decomposed using EOF analysis. Then a correlation analysis between the first four EOF temporal modes of variability and key drivers allows evaluating the relative importance of the different drivers to explain DO variability. Finally, year 2021 is compared to the 1999-2020 climatology to identify the main contributor(s) in 2021.

This is a very interesting study which has multiple interests in the context of actual changes. And DO is a good candidate to detect and monitor changes. However, I have some concerns about the foundations of the study. They are detailed below.

We thank Reviewer#2 for the consideration and evaluation of our manuscript. We have carefully revised it according to the suggestions. Our answers are in blue and the changes to the manuscript are in red italics. Line numbers refer to the revised version of the manuscript in the track-change version.

1. The ability of the QM method in correcting the bias of modelled DO.

QM covers a variety of methods that do not necessarily have a similar ability to correct model bias for a given domain or a given variable. Some QM methods may even deteriorate the model outputs compared to the non-corrected form. So it is therefore important to choose an appropriate bias correction method.

- Have you compared different QM methods? Does the performance vary significantly from each other? Please describe strengths and weaknesses of the QM method used for this study.

We thank Reviewer#1 for this comment, that allowed us to clarify the method we used.

We adopted a bias correction procedure to integrate observations and modelled data of oxygen in the area, with the aim of improving the representation of oxygen concentrations at the local scale. In fact, the biogeochemical reanalysis does not assimilate oxygen observations, whose availability from BGC-Argo profiles is particularly large in the area in a short period (2014-2020), and the validation of the biogeochemical reanalysis (Cossarini et al., 2021 and Teruzzi et al., 2021) showed the presence of a bias error.

Therefore, we applied the bias correction procedure as commonly done in previous studies (e.g., as illustrated in the review by Teutschbein and Seibert, 2012). We tested several methods for bias correction: (i) Additive delta change and (ii) Multiplicative delta change (e.g., Hempel et al., 2013; Maraun and Widmann, 2018); (iii) Variance scaling (e.g., Rocheta et al., 2014); (iv) Quantile

mapping (e.g., Hopson and Webster, 2010; Themeßl et al., 2011; Gudmundsson et al., 2012, with references indicated in the bibliography of the manuscript) and we performed visual inspection of all the resulting Hovmöller diagrams. We chose the Quantile Mapping because it allows us not only to characterise the bias over the entire distribution of the variable under study (e.g., Li et al., 2010), but also to not modify the main dynamics related to the dissolved oxygen, that are quite satisfactorily reproduced by the model in the considered period.

Due to the imposed length limitation for the Ocean State Report contributions (the recommended number of figures and words of the paper was indicatively limited to a maximum of 4 figures and 3000 words), this paper can not deepen the comparison among different bias correction methods. Nevertheless, we added an appendix that includes more details on the adopted method. In particular, in the "Appendix A: Quantile Mapping bias correction of DO concentration profiles" we included the Hovmöller diagrams of dissolved oxygen concentration before the bias correction by Quantile Mapping (Fig. A.1a) and after the procedure (Fig. A.1b) and the frequency histograms of oxygen concentrations before (Fig. A2a) and after the correction (Fig. A2b), compared with the frequency histogram of BGC-Argo float oxygen data (Fig. A2c). Moreover, we added the following text at lines 248-258:

*Figures A1 and A2 show the modelled DO concentration profiles and histogram distributions before and after the Quantile Mapping bias correction, respectively, conducted by using the BGC-Argo float measurements available in 2014-2020 (Fig. 1c). The Quantile Mapping, better than other methods (i.e. Additive Delta Change, Multiplicative Delta Change and Variance Scaling; results not shown), acted on the profiles by modifying the values of the concentrations (as indicated by the different colorbars in Figs. A1a and A1b) but, at the same time, maintaining the main dynamics observed before the correction: mixing and stratification at the surface during the year, subsurface oxygen maximum onset in spring and development in summer, and interannual variability related to the mixed layer depth dynamics in the intermediate layers. The distributions of the values of the model output before and after the Quantile Mapping bias correction and the values from BGC-Argo floats are displayed in Fig. A2. The correction actually changed the modelled values (Fig. A2a) to reproduce the shape of the distribution of the observations (Fig. A2c). In particular, after the correction (Fig. A2b) the modelled data show higher variability and a more skewed distribution toward the higher values, similarly to the observations.*

and the following sentence at lines 105-106:

*Further details on the Quantile Mapping bias correction are included in Appendix A.*

[Figure]

*Figure A1: Hovmöller diagram of the modelled oxygen concentrations spatially averaged within the area of autocorrelation equal to 0.9 indicated in Fig 1b, before the bias correction by Quantile Mapping (a) and after the procedure (b).*

[Figure]

*Figure A2: Frequency histogram of modelled oxygen concentrations before the bias correction by Quantile Mapping (a) and after the procedure (b), compared with BGC-Argo observations (c).*

Furthermore, in order to have an independent validation of the procedure, we also made a comparison between the model output (before and after the bias correction) with independent observations in the area, coming from the "EMODnet_int" dataset, defined in Cossarini et al., 2021 as EMODnet collection of in situ data (1999-2016 time period) integrated with additional oceanographic cruises (references in Cossarini et al., 2015; Lazzari et al., 2016). As reported in Fig.

R2.1, the bias correction procedure by Quantile Mapping allowed us to reduce the differences between modelled and observed oxygen concentrations at surface and under 100 m, without modifying the depth of subsurface oxygen maximum. Despite a residual bias under 400 m depth, that we ascribe to some underestimated processes in the model (related to the biological respiration), this result further supports the choice of the Quantile Mapping as a proper bias correction procedure in our case study.

In conclusion, it appears that the model could be too energetic in the vertical processes, and it produces too homogeneous oxygen vertical profiles. Indeed, Quantile Mapping successfully improves the distribution of oxygen data and its comparison with independent observations without modifying the temporal succession of oxygen maxima and minima in the southern Adriatic area.

[Figure]

**Figure R2.1: Vertical profile of modelled oxygen concentrations before the bias correction by Quantile Mapping (left panel, blue line) and after the procedure (right panel, black line), with values horizontally averaged in the southern Adriatic Sea (within the cross-correlation area r>0.9, Fig.1 of the manuscript), compared with observations from Emodnet_int dataset (Cossarini et al., 2021) in the same area, vertically averaged in the layers 0-25 m, 25-50 m, 50-75 m, 75-100 m, 100-125 m, 125-150 m, 150-200 m, 200-400 m, 400-600 m (red dots). Dashed lines for model and observations profiles indicate the standard deviation of data.**

Cossarini, G., Lazzari, P., and Solidoro, C. Spatiotemporal variability of alkalinity in the Mediterranean Sea. Biogeosciences 12, 1647–1658, 2015.

Cossarini, G., Feudale, L., Teruzzi, A., Bolzon, G., Coidessa, G., Solidoro, C., Di Biagio, V., Amadio, C., Lazzari, P., Brosich, A., and Salon, S.: High-Resolution Reanalysis of the Mediterranean Sea Biogeochemistry (1999–2019), Front. Mar. Sci., 8, 1537, https://doi.org/10.3389/fmars.2021.741486, 2021.

Di Biagio, V., Salon, S., Feudale, L., and Cossarini, G.: Subsurface oxygen maximum in oligotrophic marine ecosystems: mapping the interaction between physical and biogeochemical processes, Biogeosciences, 19, 5553–5574, https://doi.org/10.5194/bg-19-5553-2022, 2022.

Hempel, S., Frieler, K., Warszawski, L., Schewe, J., and Piontek, F.: A trend-preserving bias correction – the ISI-MIP approach, Earth Syst. Dynam., 4, 219–236, https://doi.org/10.5194/esd-4-219-2013, 2013.

Lazzari, P., Solidoro, C., Salon, S., and Bolzon, G. Spatial variability of phosphate and nitrate in the Mediterranean Sea: a modelling approach. Deep Sea Res. 108, 39–52. doi: 10.1016/j.dsr.2015.12.006, 2016.

Li, H. Sheffield J., Wood E.F. Bias correction of monthly precipitation and temperature fields from Intergovernmental Panel on Climate change AR4 models using equidistant quantile mapping. J. Geophys. Res. 115: D10101, doi: https://doi.org/10.1029/2009JD012882, 2010

Maraun, D., & Widmann, M. . Statistical downscaling and bias correction for climate research. Cambridge University Press., DOI: 10.1017/9781107588783, 2018

Rocheta E., Evans J.P. and Sharma A. Assessing atmospheric bias correction for dynamical consistency using potential vorticity, Environ. Res. Lett. 9 124010, DOI 10.1088/1748-9326/9/12/124010

Teruzzi, A., Di Biagio, V., Feudale, L., Bolzon, G., Lazzari, P., Salon, S., Coidessa, G., and Cossarini, G.: Mediterranean Sea Biogeochemical Reanalysis (CMEMS MED- Biogeochemistry, MedBFM3 system) (Version 1) Copernicus Monitoring Environment Marine Service (CMEMS) [data set], https://doi.org/10.25423/CMCC/MEDSEA_MULTIYEAR_ BGC_006_008_MEDBFM3, 2021.

Teutschbein, C., & Seibert, J. Bias correction of regional climate model simulations for hydrological climate-change impact studies: Review and evaluation of different methods. Journal of hydrology, 456, 12-29, https://doi.org/10.1016/j.jhydrol.2012.05.052, 2012.

Thrasher B., Maurer E.P., McKellar C., Duffy P.B. Technical Note: Bias correcting climate model simulated daily temperature extremes with quantile mapping. Hydrol. Earth Syst. Sci. 16:3309–3314. doi: 10.5194/hess-16-3309-2012, 2012.

- In the current draft, it is unclear to me how the reanalysis product is corrected with the BGC-Argo profiles. More details are needed on this correction.

The Copernicus reanalysis has been corrected with BGC-Argo profiles only in retrospect, using the Quantile Mapping as a bias correction procedure on the oxygen concentration in the Southern Adriatic Sea.

In fact, the current version of the Copernicus reanalysis (Cossarini et al., 2021 and Teruzzi et al., 2021) only features the assimilation of surface chlorophyll concentration (observations from Ocean Colour product based on ESA-CCI data). The data assimilation is performed once a week during the reanalysis run through a variational scheme (3DVarBio, see details in Teruzzi et al., 2014, 2018, 2019) to correct the four phytoplankton functional groups (17 state variables including carbon, chlorophyll, nitrogen phosphorus and silicon internal quotas) of the BFM biogeochemical model.

However, we added more details on the used Quantile Mapping method as bias correction procedure in Appendix A of the revised manuscript (please see our reply to the previous comment).

- Are there no other in-situ data (other than Copernicus In-Situ TAC) to evaluate the bias-corrected DO time series (before BGC-Argo area) and confirm the ability of the QM method to correct the modelled DO ?

Please see our reply to a previous comment at pages 14-15 about the comparison of model output before and after the bias correction by Quantile Mapping with an independent dataset of in situ oxygen data.

- To validate the QM method, it would be very helpful to provide the Hovmoller diagram of modelled DO (same as Figure 1d) before applying the bias-correction procedure.

We agree. Please see our reply at pages 13-14 about the Appendix A that we added to the manuscript.

- The bias-corrected DO time series (Fig 1d) shows that the deep layers are progressively enriched with oxygen over a period of 7-8 years (1999 to 2006), and then a mechanism re-initialize the oxygen at depth. Another cycle starts again (2007 to 2013), and then a third one seems to start and then perhaps disarmed by the bias-correction. This cyclic deep enrichment in oxygen is not really discussed in the paper. Is it an observed feature ? Is it already present in the non-corrected time series ? Is it the result of the bias-correction ? It could even be interpreted like a drift of the modelled DO, with a relaxation to initial conditions each 7-8 years. A lot of questions can arise from this cyclic enrichment in DO… So more clarification is needed.

As explained in our reply to Reviewer#1's comments, the initial draft of this contribution followed the guideline of the Ocean State Report in terms of number of figures, length of the text and focus on the last available year. However, we agree that the multiyear sequence of DO enrichments and reductions is an interesting feature worth to be discussed in the manuscript.

As we replied also to Reviewer#1 (pag. 3), the alternation of periods with low and high oxygen concentrations in the water column is mainly associated with the mixed layer depth (Fig. 2b) on the annual scale (leading to a deep enrichment in oxygen) and with the NIG circulation (Fig. 2f) on the multiannual scale (generally an enrichment during anticyclonic phase and reduction during the cyclonic phase). These features make the statistical analysis in terms of EOF particularly suitable for our study. Moreover, the from-year-to-year varying inflow of the Northern Adriatic dense Water (NAddW) can further increase the oxygen content in the Southern Adriatic pit, in the deepest layers of the pit due to the "saw tooth" mechanism (involving alternating long-lasting

mixing processes and sudden density increases due to the intrusion of very dense Northern Adriatic water, e.g. in 2012, Querin et al., 2016) and at intermediate depths due to more sporadic events, like the double salinity maximum, as indicated in Kokkini et al., 2020 for 2015-2016 years.

The higher values of oxygen concentration in subsurface layers in years 2004-2006, 2010-2013, 2016-2017 with respect to the other years is clearly visible also in the Hovmöller plot of oxygen concentration before the bias correction, as indicated in Figure A1 in Appendix A related to the Quantile Mapping bias correction method.

Since the investigation of the drivers is the object of the correlation analysis with EOF time series, we added at the beginning of the Results Section (lines 139-141) a comment on oxygen cyclicity:

*Dissolved oxygen in the southern Adriatic area (Fig. 1a) shows in the subsurface layers an alternation between periods of enrichment (in 2004-2006, 2010-2013, 2016-2017) and sharp declines that impacted the Oxygen Minimum Layer (OML), located between 100 and 300 m. Low concentration values are observed also in the years between 1999 and 2003.*

Querin, S., Bensi, M., Cardin, V., Solidoro, C., Bacer, S., Mariotti, L., ... & Malačič, V. (2016). Saw-tooth modulation of the deep-water thermohaline properties in the southern Adriatic Sea. Journal of Geophysical Research: Oceans, 121(7), 4585-4600.

Kokkini, Z., Mauri, E., Gerin, R., Poulain, P. M., Simoncelli, S., & Notarstefano, G. (2020). On the salinity structure in the South Adriatic as derived from float and glider observations in 2013–2016. Deep Sea Research Part II: Topical Studies in Oceanography, 171, 104625.

2. The necessity to bias-correct the modelled DO

Modelled DO is corrected ? But why is it necessary?

As we replied to a comment at pages 12 and 16-17, modelled dissolved oxygen concentration has been corrected only in retrospect (i.e., the Copernicus product does not include such a correction). In fact, since this study has been conducted on a local scale in which there is a large availability of BGC-Argo floats, we also integrated observations from floats to reduce the model bias (Cossarini et al., 2021; Teruzzi et al., 2021) and to better reproduce local dynamics. Please refer to our reply to the comment on line 68 of the submitted manuscript (page 22 of this document) for the sentence added in the revised manuscript to better explain this part.

It is difficult to appreciate the bias correction. Need to see the temporal evolution of modelled DO before applying the bias-correction (hovmoller diagram).

As explained above, we included in Appendix A additional details on the Quantile Mapping method, including the Hovmöller diagram of dissolved oxygen before the bias correction.

Bias-correction modifies the signal and may alter the link between DO and its drivers and thus reduce the correlation between the decomposed DO time series and forcing indexes... In fact, all correlations are quite "low", at best "moderate". Can't it come from this bias-correction? Have you tried the same analysis without bias-correction ?Or maybe, the correction does not modify the signal... but in this case, more clarification is needed in the text.

We applied the bias correction in order to "align" model results to the observations, thus correcting model errors. Then, the EOF analysis has been applied to the best time series available, that is, indeed, the timeseries corrected by the bias correction technique.

Furthermore, as shown in the figures A2 and R2.1, the bias correction increases variability of data (i.e., correcting the high positive excess kurtosis of the model data distribution) without modifying the temporal succession of maxima and minima and the shape of vertical oxygen profiles. Thus, EOF analysis should not be substantially impacted by the bias correction. Indeed, for sake of curiosity and completeness, we applied the same EOF analysis on the Hovmöller diagram before the bias correction (Figure A1a). Results are analogous to the ones obtained with the bias correction.

In particular, both EOF vertical patterns and EOF time series of the modes of dissolved oxygen before the bias correction (Fig. R2.2b,d,f,h) are quite similar to the analysis done on the bias corrected output (Fig. 3b,d,f,h in the manuscript). We can spot some differences in the intermediate part of the EOF vertical pattern of mode 1 only. We believe that this reflects the lower variability of data in the model results before the bias correction (e.g., model profiles before bias correction are vertically more homogeneous in the subsurface part, Fig. R2.1).

[Figure]

**Figure R2.2: EOF time series (a, c, e, g) and vertical patterns (b, d, f, h) of the first four modes computed on the dissolved oxygen concentration in the southern Adriatic area shown in Fig. A1a. The explained variances of the four modes are: 41.7%, 30.3%, 18.9%, 4.5%.**

|  | mode 1 | mode 2 | mode 3 | mode 4 |
|---|---|---|---|---|
| **HFlux (SAdr)** | -0.19 | 0.35 | 0.63 | 0.31 |
| **MLD (SAdr)** | 0.41 | -0.18 | -0.15 | -0.2 |
| **surf chl (SAdr)** | 0.62 | -0.14 | -0.25 | n.s. |
| **subsurface chl (SAdr)** | n.s. | 0.25 | 0.49 | 0.35 |
| **Hflux NAdr (2-months lagged)** | -0.71 | 0.26 | 0.30 | 0.13 |
| **NIG vorticity (NIon)** | n.s | -0.40 | 0.27 | -0.39 |

**Table RT2.1: Correlations between the EOF time series of the first four modes (Fig.R2.2a,c,e,g) and the forcing fields (Fig. 2 of the manuscript, with heat fluxes in the northern Adriatic Sea time-lagged by two months). Not significant correlations are identified by a significance level higher than 0.05 and indicated by "n.s." acronym in the table.**

Correlation analysis (Table RT2.1 and Table 2 in the manuscript) shows that the signs of the relationship between EOF time series and drivers remain the same for modes 2, 3 and 4. Moreover, the correlation values are pretty similar. Only mode 1 shows some differences: it captures mostly the dynamics of the most superficial layer while the remaining part of the profile shows very small variability (Fig. R2.4b) and correlations become higher for northern Adriatic heat fluxes, surface chlorophyll and mixed layer depth drivers. The difference in the intermediate and deeper layers between the two first modes (Fig. 3 of the original manuscript and Fig. R2.2) could be ascribed to the model bias. In fact, the model appears to be too energetic in the subsurface layer producing too homogeneous vertical profiles below 50-100m (Fig. R2.1). Thus, we think that (i) the bias correction is essential to correct a potential model inconsistency; (ii) the EOF analysis applied to the bias corrected model results provided clearer signals given the higher variability of the corrected data.

As a conclusion, since the output of the statistical analysis is not modified in a relevant way, and the bias correction validation produced encouraging results (please see our reply at pages 13-15), we consider that the method that we applied is scientifically valid.

3. The focus on the interannual variability

The draft states that this study focuses on DO interannual variability related to multiple drivers.

But, 5 over the 6 forcing indexes presented on Figure 2 mainly highlight a seasonal pattern. Seasonal pattern is also a main feature of the 3 first EOF temporal modes. And a major part of the results (section 3.2) aims to characterize the seasonal variability.

If the paper is intended to address DO inter-annual variability and long-term dynamics, shouldn't the seasonal cycle be removed from the signal before starting the analysis ? Have you tried this ?

We thank Reviewer#2 for having raised this point, that allowed us to clarify how we applied the EOF decomposition.

We have actually tried to identify all the temporal components present in the profile time series (i.e., Hovmöller diagram) of oxygen concentrations by EOF decomposition, rather than decompose the time series by classical time series decomposition (e.g., as T = interannual + seasonal + irregular by means of methods like X-11 (e.g. Colella et al., 2016)) and then extract the interannual component and compare it with the de-seasonalized time series of the drivers.

As noticed also by Reviewer#1, the time series shows a cyclicity over a time period of some years, which might not be revealed by classical time series decomposition methods. Thus, we preferred to apply EOF analysis on the not de-seasonalized time series. In fact, we considered that, on one hand, the seasonality would have emerged in the EOF time series and, on the other hand, that we could verify if multiannual cyclicity could have affected the seasonal cycle.

Colella, S., Falcini, F., Rinaldi, E., Sammartino, M., & Santoleri, R. (2016). Mediterranean ocean colour chlorophyll trends. PloS one, 11(6), e0155756.

4. The interpretation of the correlation analysis

The study is based on an analysis of correlation between the first four EOFs temporal modes and the six main forcing in the area. On the 24 correlations calculated, only 4 are greater than 0.5 (with a maximum correlation of 0.68), 15 are less than 0.5 and 5 are not significant. I find the link between EOFs modes and forcing indexes a bit week, and the message sometimes confusing, with the term "significant correlation". A statistically significant correlation must not be confused with relevant correlation. Please make a clear distinction between the two.

We agree. We carefully revised the used terms as indicated below and, in particular, we specified how we tested the significance of the correlation coefficients (line 133-134) :

*Moreover, we tested the significance of the correlation coefficients between EOF and driver time series using a parametric t-test (with a reference significance level equal to 0.05).*

in order to distinguish a statistically significant correlation from a relevant one.

The scale used to classify correlations is not really appropriated. It is somewhat exaggerated to mention a correlation of r=-0.61 as "strongly correlated".

A reasonable classification would be:    r < 0.5 à low correlation

0.5 < r < 0.7 à moderately correlated

0.7 < r < 0.9 à highly/strongly correlated

We agree. We modified the used terminology by following the Reviewer#2's suggestion and by using in the text expressions like: "a statistically significant but moderate correlation", "moderately correlated", "moderately associated".

Maybe, scatter plots (EOF temporal mode vs forcing index) would be helpful to appreciate the relationship between the 2 signals.

We appreciate this suggestion by Reviewer#2, but, since we have considered the correlations between the EOF time series of (the first) four modes (Fig. 3a,c,e,g) and six drivers (Fig. 2), it would lead to adding 24 plots (or panels). In the perspective of a short paper (in line with the general OSR7 guidelines), we think that there would be too much detailed information to show and comment.

 Specific Comments

L14: "we used DO modelled by the latest Copernicus Marine biogeochemical reanalysis", please add "for the Mediterranean Sea".

We agree. We added it.

L68: why is it necessary to correct the reanalysis? please add a few words

We agree. In the revised manuscript, we better explained why it is necessary to introduce the correction (lines 88-90):

*In fact, the biogeochemical reanalysis does not include BGC-Argo float DO assimilation and displays an average RMSD of 15 mmol $m^{-3}$ for DO in the 0-600 m depth layer with respect to the observations in the area (Cossarini et al., 2021, Teruzzi et al., 2021a-b).*

and we added the references for Teruzzi et al., 2021a-b in the bibliography.

L77: what does "auto correlation" mean ?

We thank Reviewer#2 for this comment, that allowed us to clarify the used terminology.

The "autocorrelation" is actually a cross-correlation between the central point of the Southern Adriatic pit and every spatial gridpoint in the domain (Jones et al., 2015, Martellucci et al., 2021). As we replied to Reviewer#1, we also specified in the revised manuscript that the biogeochemical reanalysis data considered in the spatial cross-correlation analysis are dissolved oxygen, nitrate and chlorophyll concentrations at surface.

In particular, we added at lines 101-104:

*Specifically, we considered the cross-correlation between the surface data of DO, nitrate and chlorophyll concentrations in the central point of the pit and those at each spatial grid point in the domain, to identify the area that displayed the same dynamics at the surface from a phenomenological perspective.*

Jones E.M., Doblin M.A., Matear R. , King E. Assessing and evaluating the ocean-colour footprint of a regional observing system J. Mar. Syst., 143 (2015), pp. 49-61, https://doi.org/10.1016/j.jmarsys.2014.10.012

Martellucci, R., Salon, S., Cossarini, G., Piermattei, V., & Marcelli, M. Coastal phytoplankton bloom dynamics in the Tyrrhenian Sea: Advantage of integrating in situ observations, large-scale analysis and forecast systems. Journal of Marine Systems, 218, 103528, https://doi.org/10.1016/j.jmarsys.2021.103528, 2021.

L85: Is Pearson correlation used ?

Yes, we used Pearson correlation. We specified it in the revised manuscript (line 112).

L97: why is the SAdr box used to average forcing (41.6-42.1°N / 17.6-18.1°E) different from the SAdr area used to average modelled DO (area of autocorrelation 0.9) ?

As also replied to Reviewer#1, we recognise that an explanation about this choice was missing in the submitted manuscript.

All the boxes used to average the time series of the meteo-marine drivers were chosen in order to obtain clear signals. In particular, for mixed layer depth (and heat fluxes and chlorophyll concentrations) we needed to limit the considered area in the Southern Adriatic Sea in order to include the whole volume of the pit.
In the revised manuscript we modified the text in order to clarify this aspect.

In particular, we added at line 128-129:

*to consider the whole volume of the pit*

L102-103: why has the vorticity been filtered ? why other forcing indexes have not been filtered ? please clarify

The 13-month moving average is a procedure commonly done on NIG vorticity (e.g., Menna et al., 2019) to highlight the cyclonic and anticyclonic periods in the northern Ionian Sea circulation.

However, we recognise that the use of not filtered time series should be applied also in case of NIG vorticity, also in the perspective of including all the temporal components as illustrated in our reply to a previous comment at page 21.

We applied the correlation analysis also to the not filtered NIG vorticity time series and we obtained values similar to the ones reported in the manuscript: -0.40 for the second mode and    -0.37 with the fourth mode. The main difference was that the correlation with the third mode was not significant.

We replaced the values in the last row of Table 2 and at line 166 with the ones here indicated, deleted lines 134-136 and added this sentence (lines 124-125):

*In particular, the temporal phases of the NIG are defined as cyclonic and anticyclonic, respectively, when the vorticity field is positive and negative, as highlighted by the de-seasonalized time series in Fig. 2f.*

Moreover, we modified the following part of the caption of Fig.2:

*(d) subsurface chlorophyll concentration (30-80 m layer in which deep chlorophyll maximum (DCM) is located, Prod1), (e) net heat fluxes in NAdr (Prod6), (f) NIG current vorticity (gray line) and de-seasonalized time series as obtained by applying a low-pass filter of 13 months (black thick line) (Prod4 and Prod5).*

and Fig. 2f displaying NIG vorticity time series:

[Figure]

(new version of Fig. 2)

L117: change "significantly correlated…" to "statistically significant but moderate correlation with the heat flux (r=0.56) and low correlation with the subsurface chlorophyll concentration (r=0.43)"

We agree, but we changed "low" with "lower".

L123: "strongly correlated" for r=-0.61 and r=0.68 is somewhat exaggerated… please change to "moderately correlated"

We did it.

L139-141: I would not say that Fig 1d shows 2 distinct periods 2005-2006 and 2012-2014, but rather 2 periods of 7-8 years each with progressive oxygenation of the deep layers (1999-2006 and 2007-2013). Is this the third mode that explains this deep oxygenation ? Which mechanism re-initialize the oxygen content at depth in 2007 and 2014 ? Is it confirm by analysis of DO advection from northern Adriatic sea ? here a time series of DO advection would confirm the hypothesis.

As explained at pages 17-18, the alternation of enrichment and reduction in the oxygen concentration in the subsurface layers is related to multiple factors: the mixed layer depth dynamics (Fig. 2b), the NIG circulation (Fig. 2f), and in deeper layers to deep water inflow from Northern Adriatic Sea (heat fluxes in Northern Adriatic Sea in Fig. 2e).

Regarding the inflow of NAddW, it causes an enrichment of oxygen in the intermediate (400-600 m depth) and deep layer (below 600 m). The inflow of NAddW in the Southern Adriatic pit can be visualised at BB mooring site (Fig. R2.3, data from Paladini de Mendoza et al., 2022), that shows potential density higher than 29.2 kg m$^{-3}$ in 2012-2013 and in 2017-2018. Since dense waters coming from the Northern Adriatic Sea are rich in oxygen, such density values support our interpretation of enrichment in oxygen in the deeper layers due to this forcing: in Table 2, r=0.68 for heat fluxes in the Northern Adriatic Sea (Fig. 2e) and the third mode of EOFs (Fig. 3e,f), even if the enrichment in years 2017-2018 (recognisable in Fig. 1d) is not clearly captured by the time series of the third mode of EOFs.

Furthermore, the pronounced NAddW formation that occurred in 2005-2006 is well documented by observations illustrated in other studies (e.g. Socal et al., 2008).

[Figure]

**Figure R2.3: Potential density of seawater recorded in the BB mooring site in the Southern Adriatic Sea from 2012 to 2020.**

On the other hand, other processes (e.g., entrance of less oxygenated water from Otranto) can be the most relevant drivers in specific years causing the observed decrease of concentration in the intermediate layer. For example, as shown by our analysis results (e.g., positive values of EOF time series of mode 4 in 2015-2016 and 2019-2021 and correlation of mode 4 with NIG) we hypothesise

that the mechanism of oxygen decrease in the intermediate layer is associated to the NIG and the inflow of less oxygenated waters through the Otranto Strait.

Unfortunately, oxygen advection flows were not part of the output of the Copernicus Marine Service reanalysis of Mediterranean Sea and a reconstruction in retrospect of the advection from the model output can be very demanding. Therefore, we proposed to identify the relevant processes and drivers indirectly with our EOF and correlation analysis.

Paladini de Mendoza F., Schroeder K., Langone L., Chiggiato J., Borghini M., Giordano P., Verazzo G., & Miserocchi S.. Moored current and temperature measurements in the Southern Adriatic Sea at mooring site BB and FF, March 2012-June 2020 (1.0) [Data set]. Zenodo. https://doi.org/10.5281/zenodo.6770202, 2022.

Socal, G., Acri, F., Bastianini, M., Bernardi Aubry, F., Bianchi, F., Cassin, D., Coppola, J., De Lazzari, A., Bandelj, V., Cossarini, G. and Solidoro, C. Hydrological and biogeochemical features of the Northern Adriatic Sea in the period 2003–2006. Marine Ecology, 29: 449-468, https://doi.org/10.1111/j.1439-0485.2008.00266.x, 2008.

L142: The analysis performed on detrended DO time series provides pretty similar results.But did you try to perform the analysis on de-seasonalized DO time series ?

Please see our reply to a previous comment on de-seasonalized time series at pages 21.

L151-152: I am sorry, I am not able to see the anomaly (lower than average) in subsurface chlorophyll in Fig 2d… Could you give more details please?

We recognise that the negative anomaly of subsurface chlorophyll is not so evident from Fig. 2d. We estimated that this negative anomaly averaged during 2021 year is equal to 6% with respect to the climatological mean (1999-2020) and we modified the sentence by adding the following paragraph at lines 194-197:

*The time series of the first mode (Fig. 3a) is actually negative from 2019 and corresponds to the negative anomaly of only one of its drivers (Table 2), i.e. subsurface chlorophyll (Fig. 2d), and not heat fluxes (Fig. 2a). In particular, we estimated a mean negative anomaly approximately equal to 6% with respect to the climatological mean (1999-2020) for subsurface chlorophyll in 2021.*

L152: please add the reference to Fig. 2d at the end of the sentence.

We did it.

L153-155: 2021 anomaly is an important result of the paper, and a central aspect for the Ocean State Report. I think that figure(s) is/are missing to support the text of Section 3.3. For exemple, a time serie of Temperature, salinity and oxygen content at entrance of the SAdr trough the Otranto Strait would confirm the regime shift mentioned in the text, with the entrance of new water masses (warmer, saltier and less oxygenated) well reproduced by the physical and biogeochemical models.  And the reasons why circulation has changed could be developed further.

We thank Reviewer#2 for this comment, that allowed us to better present our results.

As we replied also to Reviewer#1, we included in a new appendix (i.e., Appendix B) a plot to show the time series of temperature and salinity of water entering the Adriatic Sea through the Otranto Strait:

[Figure]

*Figure B1: Time series of temperature (a) and salinity (b), averaged in the vertical layers 0 - 150 m (red lines) and 150-600 m (black lines) of the Otranto Strait (39.8°N, 18.5° - 19.5° E) in the 1999-2021 time period. In the top panel, light red and dark red indicate data before and after de-seasonalization, respectively. Data are provided by Copernicus physical reanalysis (Prod3, Table 1).*

Moreover, since an increase in salinity in the Southern Adriatic pit has been already detected and described in previous studies (Mihanović et al., 2021; Menna et al., 2022b), in the reviewed version of the manuscript we also summarized more clearly these results. In particular, we replaced the lines 153-157 of the submitted version by the text hereafter (lines 198-211):

*One of the causes of the decrease in total oxygen concentration in the SAdr could be due to the exceptional salinization observed in the SAdr since 2017 (Mihanović et al., 2021, Menna et al., 2022b). This increase was related to the inflow of new, warmer and significantly saltier water masses from the northeastern Ionian Sea (Mihanović et al., 2021, Menna et al. 2022b).*

*The inflow of saltier and warmer water masses is also evident by observing the temporal evolution of these parameters through the Strait of Otranto (Fig. B1). In particular, in the upper layer (0-150 m) both temperature and salinity show an overall positive trend throughout the period 1999-2021, whereas the decrease observed in 2006-2011 and 2017-2018 can be associated with the inflow of*

*less saline AW, triggered by the anticyclonic circulation of the NIG (Fig. 2f). In the intermediate layer (150-600 m), salinity shows a positive trend in 1999-2021, while no clear trend is observed for temperature. Moreover, a sharp increase in salinity (~ 0.1) is observed in 2019. This increase occurred after the NIG inversion from anticyclonic to cyclonic (Fig. 2f), resulting in a further increase in salinity due to both the decrease in AW advection and the increase in LIW inflow.*

L169-170: "the cyclicity" is briefly mentioned here while it is an obvious feature in figure 1. More discussion about this point is necessary.

We agree. Please see our reply at pages 17-18. To be clearer also in this part, we specified:

*an alternation of enrichment and reduction phases*

in the revised text of the manuscript (line 224).

Figure 1a: please add the bathymetry.

We agree. We provided the bathymetry in a new version of Fig. 1a.

Figure 2: is Product 3 the forcing of Product 1 ? and are Products 4, 5, 6 used as atmospheric forcing or data assimilation in Product 3 ? please mention the links between the products.

We confirm that Product 3 (The Mediterranean Sea physical reanalysis, Escudier et al., 2021) is the forcing of Product 1 (The Mediterranean Sea biogeochemical reanalysis, Cossarini et al., 2021; Teruzzi et al., 2021) and that Product 6 (Global climate and weather analysis, ERA5, Hersbach et al., 2018) is the forcing of Product 3. On the other hand, Products 4 and 5 (Reprocessed and near real-time altimeter satellite gridded Sea Level Anomalies, SEALEVEL_EUR_PHY_L4_MY_008_068 and SEALEVEL_EUR_PHY_L4_NRT_OBSERVATIONS_008_060) are not forcing or assimilated in Product 3. Indeed, Product 3 assimilates a quite different product, i.e. SEALEVEL_EUR_PHY_L3_REP_OBSERVATIONS_008_061.

We mentioned the link among the products in the caption of the new version of Table 1 that we included in the manuscript:

*Prod3 is a forcing for Prod1 and Prod6 is a forcing for Prod3.*

Table 2: please change "Not significant correlations…" to "Not statistically significant correlations…"

We did it.

**Response to Reviewer#3's comments**

GENERAL COMMENTS

In this study, the authors combine various datasets (in situ and remote observations, and numerical simulations) and various variables (physical and biogeochemical data) to analyse the dissolved oxygen variability over 1999-2021 in the Southern Adriatic Sea through an EOF decomposition. In order to estimate the contribution of a set of drivers, the correlations between four first modes of variability and the drivers are computed. The study is interesting and the authors show that the dissolved oxygen is an relevant indicator of multiple drivers of the marine ecosystem.

The manuscript is very nice example of ocean data integration and provides a new relevant indicator for the Copernicus Ocean State Report. The manuscript is relatively well written although some corrections are required (but I am not English native). Indications about the datasets and computations of some derived variables are strongly missing but the authors could easily add them. Precisions on methodology are also required. The figures appear with low quality and most of them need improvements. I would recommend the publication of this manuscript after some clarifications and improvements (see my comments below).

We thank Reviewer#3 for the consideration and feedback on our manuscript. We carefully revised it according to the suggestions. In particular, we carefully checked the English language by means of a dedicated tool and we provided figures at higher resolution. We replied to the specific points about dataset, computations and methodology below.
Our answers are in blue and the changes to the manuscript are in red italics. Line numbers refer to the revised version of the manuscript in the track-change version.

MAJOR COMMENTS

Period of bias-correction

Could the authors explain why applying a bias correction using profiling floats over 2014.-2020? Why not over the whole period? Why excluding the last year? Which is the impact of such temporal sub-sampling in bias-correction on the bias-corrected reanalysis time series? In addition, fig. 1c shows several long temporal gaps? Which impacts?

We used all BGC-Argo float data available in a qualified mode (i.e., delayed mode after a PI quality check analysis) at the time of the manuscript preparation. In particular, in 2021 only few profiles were available and, thus, we excluded them from the analysis.

Anyway, the available float profiles cover all seasons, i.e., potentially reproduce the typical annual dynamics of the oxygen in the water column (i.e., mixing and stratification cycle at surface and subsurface oxygen maximum) and the distribution of oxygen values on a basis of four years, and the bias correction procedure has been applied considering the model output in the same days in which observations were available (i.e., there was a direct correspondence between model outputs and observations).

The temporal gaps that can be noted in Fig. 1c did not allow us to have a complete reference for the multiannual dynamics in 1999-2019, but, as we have replied to Reviewer#2, we tested

different bias correction procedures and also validated the output of the chosen method (i.e., Quantile Mapping) obtaining encouraging results.

Analysis and results

In the introduction, the authors state that they analyse the DO interannual variability (l.60). But in section 3., they also dedicate a sub section in the 2021 anomaly. Why? Because of the OSR7 that also focuses on 2021 event? Or this year has a specificity highlighted thanks to the analyses over 1999-2021? Please, clarify.

The analysis conducted for the 2021 year is actually a request for the Copernicus Ocean State Report 7 (OSR7). In fact, according to the OSR7 guidelines: (i) the core-period to be covered is 1993-2021 or earlier/later, depending on product availability and limitations; (ii) the inclusion of data during the year 2021 is mandatory (see for example the explanation of the scheme of OSR in https://marine.copernicus.eu/access-data/ocean-state-report/ocean-state-report-6) .

In our case, (i) we started the analysis from 1999, since Mediterranean biogeochemical reanalysis is available from that year, and (ii) we analysed 2021 year with respect to the 1999-2020 climatology providing a discussion of the 2021 anomaly with respect to the climatology.
In particular, the analysis of the year 2021 revealed an anomaly with respect to the framework described by the EOFs and correlations with drivers. As we described in more detail in the revised paper, a possible regime shift signal started in 2019.

Correlation

- I don't agree with the level of correlation in general. For example, 0.5 is not a high correlation. Please rephrase/moderate/modify your statements (l.117, l.123).

  We agree. We modified the statement in the revised manuscript, also accounting for the suggestions provided by Reviewer#2 (please see page 21 of this document).

- Could you detail the method used (and provide reference) for testing the significance of the correlation coefficient?

  We used the function *corrcoef* in Matlab, that provides as output both (Pearson) correlation coefficients and p-values for testing the hypothesis (parametric t test) that there is no relationship between the observed phenomena (null hypothesis). If the p-value is smaller than the significance level (that we fixed equal to 0.05), then the corresponding correlation coefficient is considered significant. We specified the test of significance in the revised manuscript at lines 133-134:

  *Moreover, we tested the significance of the correlation coefficients between EOF and driver time series using a parametric t-test (with a reference significance level of 0.05).*

Datasets

All products have to be described (briefly) in section 2., not only prod1 and prod2. For each driver of the study, indicate which product is used and how the indicator is computed (in particular, MLD, which criteria is used).

We added the reference to the used products in the revised manuscript at lines 115-123 in the list of the used drivers of oxygen dynamics in the SAdr. Moreover, for the mixed layer depth we added at lines 126-127:

*Mixed layer depth (computed in Prod3 considering the 0.03 kg m$^{-3}$ density difference with respect to the near-surface value at 10 m depth) and*

In addition, it is important to provide to complete references and DOIs (as indicated in the Copernicus Marine Service website, how to cite: https://help.marine.copernicus.eu/en/articles/4444611-how-to-cite-or-reference-copernicus-marine-products-and-services). For the dataset, do not write "the latest" in the text. I recommend to provide the complete references and to indicate the date of access in the references associated with dataset [access Month Day, Year]. Please find below the information:

Prod1: The Mediterranean Sea biogeochemical reanalysis at 1/24° of horizontal resolution and daily temporal resolution (Cossarini et al., 2021, Teruzzi et al., 2021)

- Cossarini, G., Feudale, L., Teruzzi, A., Bolzon, G., Coidessa, G., Solidoro C., Amadio, C., Lazzari, P., Brosich, A., Di Biagio, V., and Salon, S., 2021. High-resolution reanalysis of the Mediterranean Sea biogeochemistry (1999-2019). Frontiers in Marine Science.
- Teruzzi, A., Feudale, L., Bolzon, G., Lazzari, P., Salon, S., Di Biagio, V., Coidessa, G., & Cossarini, G. (2021). Mediterranean Sea Biogeochemical Reanalysis INTERIM (CMEMS MED-Biogeochemistry, MedBFM3i system) (Version 1) Data set. Copernicus Monitoring Environment Marine Service (CMEMS) https://doi.org/10.25423/CMCC/MEDSEA_MULTIYEAR_BGC_006_008_MEDBFM3I (accessed November 17, 2022)

Prod2: The Mediterranean Sea in situ quality-controlled observations, distributed by Copernicus In Situ TAC

https://doi.org/10.48670/moi-00044

Prod 3: The Mediterranean Sea physical reanalysis at 1/24° of horizontal resolution and daily temporal resolution (Escudier et al., 2021)

Escudier, R., Clementi, E., Omar, M., Cipollone, A., Pistoia, J., Aydogdu, A., Drudi, M., Grandi, A., Lyubartsev, V., Lecci, R., Cretí, S., Masina, S., Coppini, G., & Pinardi, N. (2020). Mediterranean Sea Physical Reanalysis (CMEMS MED-Currents) (Version 1) Data set. Copernicus Monitoring Environment Marine Service (CMEMS). https://doi.org/10.25423/CMCC/MEDSEA_MULTIYEAR_PHY_006_004 (accessed November 17, 2022)

Prod4 and prod 5:Reprocessed and near real-time altimeter satellite gridded Sea Level Anomalies (SLA) computed with respect to a twenty-year 1993, 2012 mean (prod4 and prod5, respectively). The product gives additional variables (i.e. Absolute Dynamic Topography and geostrophic currents).

Prod4: DOI (product): https://doi.org/10.48670/moi-00141 (accessed November 17, 2022)

Prod 5: DOI (product): https://doi.org/10.48670/moi-00142 (accessed November 17, 2022)

Prod6: Global climate and weather analysis (ERA5, Hersbach et al., 2018) with ¼º of horizontal resolution and hourly/monthly (?) temporal resolution.

We thank Reviewer#3 for these suggestions.

We included in the revised manuscript a new version of Table 1, including name, type and references for the products, following the Copernicus Marine Service data catalogue and guide in https://help.marine.copernicus.eu/en/articles/4444611-how-to-cite-or-reference-copernicus-marine-products-and-services. The complete citations for the reference papers and the Quality Information Documents of the products are included in the bibliography. For the In situ TAC product we followed the citation indicated in https://archimer.ifremer.fr/doc/00646/75807/. For ERA 5, we followed the citation included in https://cds.climate.copernicus.eu/cdsapp#!/dataset/reanalysis-era5-single-levels?tab=doc. Regarding the use of "the latest" referring to the products, we deleted this expression from the text and added the data access in Table 1.

Anyway, we are available to follow further suggestions to be aligned with the journal official standard for the product citation.

| Ref. no. | Product name & type | Documentation |
|---|---|---|
| 1 | Copernicus Marine MEDSEA_MULTIYEAR_BGC_006_008 Mediterranean Sea Biogeochemistry Reanalysis | Cossarini et al., (2021) Dataset: Teruzzi et al., (2021a, 2021b) https://doi.org/10.25423/CMCC/MEDSEA_MULTIYEAR_BGC_006_008_MEDBFM3 (Accessed on 6-3-2023) https://doi.org/10.25423/CMCC/MEDSEA_MULTIYEAR_BGC_006_008_MEDBFM3I (Accessed on 6-3-2023) |
| 2 | Copernicus Marine INSITU_MED_NRT_OBSERVATIONS_013_035 Mediterranean Sea-In-Situ Near Real Time Observations | Copernicus Marine in situ TAC (2021). Copernicus Marine In Situ TAC quality information document for Near Real Time In Situ products (QUID and SQO). https://doi.org/10.13155/75807 (Accessed on 6-3-2023) |
| 3 | Copernicus Marine MEDSEA_MULTIYEAR_PHY_006_004 Mediterranean Sea Physics reanalysis | Escudier et al., (2021) Dataset: Escudier et al., (2020) https://doi.org/10.25423/CMCC/MEDSEA_MULTIYEAR_PHY_006_004_E3R1 (Accessed on 6-3-2023) |
| 4 | Copernicus Marine SEALEVEL_EUR_PHY_L4_MY_008_068 | https://doi.org/10.48670/moi-00141 (Accessed on 6-3-2023) |

| | | |
|---|---|---|
| | *European Seas Gridded L 4 Sea Surface Heights And Derived Variables Reprocessed 1993 Ongoing* | |
| 5 | *Copernicus Marine SEALEVEL_EUR_PHY_L4_NRT_OBSERVATIONS_008_060 European Seas Gridded L 4 Sea Surface Heights And Derived Variables Nrt* | *https://doi.org/10.48670/moi-00142 (Accessed on 6-3-2023)* |
| 6 | *Copernicus Climate ERA5 Global climate and weather reanalysis* | *Hersbach at el., 2018 https://doi.org/10.24381/cds.adbb2d47 (Accessed on 6-3-2023)* |

*Table 1: Products used in the present work. Prod3 is a forcing for Prod1 and Prod6 is a forcing for Prod3. Complete references for Prod1, Prod3 and Prod6 are reported in the bibliography.*

Figures

All the figures have low resolution and too small characters (xlabel, ylabel and colorbar). Please improve and enlarge the characters of all the figures.

We provided figures at higher resolution and higher readability in the revised manuscript.

MINOR COMMENTS

In my opinion, the abstract is too much detailed (in particular with percentage of variance, correlation numbers).

We agree that the abstract should not be too detailed and we deleted the correlation number from the text (line 17 of the submitted manuscript) in the revised manuscript. On the other hand, we left the percentage of variance, to show that the first four modes capture a great part of variance (i.e., 94%), and that mode 4 explains a percentage lower than 10%.

In all the manuscript:

- Replace all "associated to" by "associated with"

  We did it.

- Replace "Levantine/Modified Atlantic Waters" by "Levantine and Modified Atlantic Waters"

  We agree with the formal correction. However, we indicated Modified Atlantic Waters simply as "Atlantic Waters" in the revised manuscript, following the guidelines on Mediterranean water mass acronyms formulated by CIESM C2 Committee in 2022 (https://ciesm.org/MWM_Acronyms/MedWaterMassAcronyms.pdf). We applied this correction through the whole manuscript.

- I would write in situ (in italic, and without "–")

We agree. We changed it.

- Avoid "/" in the text. To be replaced by a word.

  We agree. In particular, we replaced "past/future" by

  *past or future*

  (line 93 of the revised manuscript), "mixing/stratification" by

  *mixing and stratification*

  (line 115), "production/consumption" by

  *production and consumption*

  (line 179).

- In the text, do not indicate "first/second/etc column". Only reference to the figure

  We agree. In particular, we replaced "Fig. 3 (left and right columns, respectively)" by:

  *Figs. 3a,c,e,g and Figs. 3b,d,f,h, respectively*

  (line 145-146) and "first column in Fig.3" by

  *Figs. 3a,c,e,g*

  at lines 148-149 of the text and in the caption of Table 2, respectively.

Abstract

l.14: Precise Mediterranean Sea reanalysis

We did it.

l.15: satellite chlorophyll concentration

We did it.

l.15: I would prefer profiling floats that Argo floats

We used the expression "Argo floats" to explicitly make reference to the Argo program (https://argo.ucsd.edu/), whose data in the Mediterranean Sea are delivered by Copernicus In Situ Thematic Assembly Centre. Therefore, we would like to maintain such a reference, unless Reviewer#3 still suggests replacing it.

l.16 and 20: of total variance

We did it.

l.23: biological production?

Yes. We added the term "biological" as suggested.

l.31: I would replace "i.e." by "such as"

We replaced the term as suggested and, more in general, we carefully checked the English language of the manuscript.

Introduction

- Add the reference to IPPC 2021?

We thank Reviewer#3 for this suggestion. We added the reference "Pörtner et al., 2019" in the text (lines 34-35):

*Indeed, DO is currently being studied under the global warming scenarios by climate and marine ecological scientific communities (e.g. Pörtner et al., 2019 …*

and in the bibliography:

*Pörtner, H. O., Roberts, D. C., Masson-Delmotte, V., Zhai, P., Tignor, M., Poloczanska, E., & Weyer, N. M. The ocean and cryosphere in a changing climate. IPCC Special Report on the Ocean and Cryosphere in a Changing Climate. Cambridge University Press, Cambridge, UK and New York, NY, USA, 755 pp., https://doi.org/10.1017/9781009157964, 2019.*

- 38: I would delete "Despite being a marginal Sea". Not necessary and "despite" introduce a negative aspect…

We agree. We deleted such an expression.

- 2nd paragraph: the authors could reorganize the geographical description: first the Adriatic Sea, secondly the Southern part.

We thank Reviewer#3 for this comment, that allowed us to improve the text of the manuscript.

In this point of the submitted manuscript, we had introduced before the Southern Adriatic Sea and after the Adriatic Sea since the Southern Adriatic Sea (i.e., the domain under study) is an example of an area in which "oceanic processes connect the surface and deep layers", as written in the previous sentence. However, we recognise that the text should be revised to make it clearer. We modified this part in the revised manuscript (lines 38-43) as:

*so this parameter is of primary interest especially in those areas where oceanic processes connect the surface and deep layers.*

*The southern Adriatic Sea (SAdr, Fig. 1a) is one of these areas, as it is a site of deep water formation (Gačić et al., 2002; Pirro et al., 2022) and represents the deep engine of the eastern Mediterranean thermohaline circulation (Malanotte-Rizzoli et al. 1999), which is crucial for the eastern basin ventilation.*

and we added the reference for Malanotte-Rizzoli et al., 1999 in the bibliography:

*Malanotte-Rizzoli, P., Manca, B. B., d'Alcala, M. R., Theocharis, A., Brenner, S., Budillon, G., & Ozsoy, E. The Eastern Mediterranean in the 80s and in the 90s: the big transition in the intermediate and deep circulations. Dynamics of Atmospheres and Oceans, 29(2-4), 365-395,1999.*

53:" Marine Strategy Framework Directives (MSFD)" rather than "sensu MFSD"

We did it.

- 53: refer to IPPC 2021?

We thank Reviewer#3 for this suggestion. We added the citation "Pörtner et al., 2022" (lines 69-70):

*for understanding anthropogenic impacts on the marine environment (Pörtner et al., 2022).*

and in the bibliography:

*Pörtner H.O. , Roberts D.C. , Tignor M., Poloczanska E.S. , Mintenbeck K., Alegría A. , Craig M., Langsdorf S., Löschke S., Möller V., Okem A., Rama B. Climate Change 2022: Impacts, Adaptation and Vulnerability. Contribution of Working Group II to the Sixth Assessment Report of the Intergovernmental Panel on Climate Change. Cambridge University Press. Cambridge University Press, Cambridge, UK and New York, NY, USA, 3056 pp., doi:10.1017/9781009325844, 2022*

- 54: replace "Marine Strategy Framework Directives" by " MSFD"

We did it.

- 57: replace "i.e." by "distributed by the"

We modified this part as (lines 74-75):

*the present work integrates the state-of-the-art approach of in situ measurements (in 2014-2020, distributed by Copernicus In Situ TAC)*

- 58: reanalysis in the Mediterranean Sea

We did it.

- 58: I would replace "at 1/24º horizontal resolution" by "high-resolution" (such precision is given in section 2)

We agree. We included the suggested expression.

Data and method

See my previous comment concerning the dataset descriptions (references and DOIs) + all datasets have to be introduced.

We accounted for it.

- 65: reanalysis in the Mediterranean Sea

We modified this part (lines 81-82) as:

*by combining data from the Copernicus reanalysis in the Mediterranean Sea*

- 65: replace "in" by "over"

We modified this expression as:

*available for the time period 2014-2020*

- 66 I would replace "in the time" by "over the"

Please see our reply to the previous comment.

- 67: replace CMEMS by Copernicus

Please see our reply to the next comment.

- I suggest to rewrite the beginning of this section, combining l.64-66 and 67-69 (to avoid repetitions)

We thank Reviewer#3 for this comment, that allowed us to improve the manuscript. We would prefer to leave at first a general sentence about the integration of in situ data and modelled ones and then another sentence introducing more details about the method. Nevertheless, we recognise that there are some repetitions. Thus, we modified the part at lines 86-88 of the revised manuscript as:

*In particular, we used the BGC-Argo float measurements of in situ DO to compute a bias correction to the daily DO concentrations simulated by the biogeochemical reanalysis at 1/24° horizontal resolution.*

- 70: what is the reference to ThemeXX et al.?

We thank Reviewer#3 for having noticed this oversight. In fact, we had forgotten to indicate the reference in the bibliography and we added it in the revised manuscript:

*Themeßl J., M., Gobiet, A. and Leuprecht, A. Empirical-statistical downscaling and error correction of daily precipitation from regional climate models. Int. J. Climatol., 31: 1530-1544. https://doi.org/10.1002/joc.2168, 2011.*

- 71: past/future: past and future

We replaced this expression by:

*past or future*

- 89: delete "the"

We did it (in line 116).

- 90 and 96: at surface and in subsurface, (removing level)

We did it.

- 94 Levantine Water should be Levantine Intermediate Water (LIW)

Here we indicated a more general "Levantine Water", since it actually includes Levantine Intermediate Water, Levantine Surface Water, Surface Ionian Water and Cretan Intermediate Water (Civitarese et al., 2010; Schroeder et al., 2017; Table 2 in Menna et al., 2021). Therefore, we would prefer to maintain this expression, or, if Reviewer#3 suggests us to be more precise, to include the four water masses.

Schroeder, K.; Chiggiato, J.; Josey, S.A.; Borghini, M.; Aracri, S.; Sparnocchia, S. Rapid response to climate change in a marginal sea. Sci. Rep. 7, 4065, https://doi.org/10.1038/s41598-017-04455-5, 2017.

Menna, M., Gerin, R., Notarstefano, G., Mauri, E., Bussani, A., Pacciaroni, M., & Poulain, P. M. On the Circulation and Thermohaline Properties of the Eastern Mediterranean Sea. Frontiers in Marine Science, 903, https://doi.org/10.3389/fmars.2021.671469, 2021.

103: could you explain why 13 months?

The 13-month moving average is applied in order to remove the seasonal and intra-annual variations and it is a commonly done procedure to highlight the sense of rotation of the

Northern Ionian Gyre (NIG), i.e., cyclonic or anticyclonic (e.g. Menna et al., 2019). The use of 13 months instead of 12 is related to the use of a centred average, since odd orders (i.e., number of periods, here years, over which the moving average is calculated) allows to avoid phase shift phenomena in the smoothed time series.

Menna, M., Suarez, N. R., Civitarese, G., Gačić, M., Rubino, A., & Poulain, P. M. Decadal variations of circulation in the Central Mediterranean and its interactions with mesoscale gyres. Deep Sea Research Part II: Topical Studies in Oceanography, 164, 14-24, https://doi.org/10.1016/j.dsr2.2019.02.004, 2019.

110: "vertical pattern" rather than "spatial pattern"

We did it.

Results

- I suggest to delete the 3.1 Subtitle and move this paragraph in section 3.2. Then, Section 3.2 will become 3.1, and 3.3 will become3.2.

We thank Reviewer#3 for this suggestion. We modified the structure of the manuscript as suggested.

- l.115: replace "almost 50%" by "48.9%"

We did it.

- l-119: 20%, be precise here, indicate 19.7 and 17.7 % respectively. In general, be more precise in section 3. results

We did it.

-l.123-125: such results raise the questions about the dependency between the drivers. I may be not a problem. Do you have an opinion about that?

The analysis of the correlation among time series of EOFs and meteo-marine drivers gave worthwhile information about dynamics influencing oxygen concentration in the Southern Adriatic pit, but drivers acting on the oxygen concentration in the area are unavoidably interrelated.

In the specific case of mode 3, to which lines 123-125 of the submitted manuscript are referred, it partly explains the intensity of the vertical gradient between surface and subsurface layers (Fig. 3f). Thus, the intensity of the gradient is influenced by drivers impacting either the surface and subsurface dynamics.

-l.138: multiple modes

We did it.

- l. 140: where the seasonal signal is strong, you could remove it to better highlight and quantify the interannual variability, no? Have you test it?

As we replied to Reviewer#2 (page 21 of this document), we have actually tried to identify all temporal components in the profile time series (i.e. Hovmöller diagram) of oxygen concentrations

by EOF decomposition, rather than conducting a classical time series analysis (e.g., X-11 method after de-seasonality; Colella et al., 2016).

Indeed, the oxygen time series shows a sort of cyclicity on a time period of some years (as also noticed by Reviewer#1) and thus we believed that EOF analysis on the non-de-seasonalized time series should be the most appropriate approach to reveal different frequencies of the time series signal (e.g., seasonal and multiannual). In fact, we considered that, on one hand, the seasonality would have emerged in the EOF time series and, on the other hand, that we could verify if multiannual cyclicity could have affected the seasonal cycle.

Colella, S., Falcini, F., Rinaldi, E., Sammartino, M., & Santoleri, R. (2016). Mediterranean ocean colour chlorophyll trends. PloS one, 11(6), e0155756.

2021 Anomaly

l.152: r=?

As we also indicated in our replies to Reviewer#1 and Reviewer#2, we clarified this sentence in lines 194-197 as:

*The time series of the first mode (Fig. 3a) is actually negative from 2019 and corresponds to the negative anomaly of only one of its drivers (Table 2), i.e. subsurface chlorophyll (Fig. 2d), and not heat fluxes (Fig. 2a). In particular, we estimated a mean negative anomaly approximately equal to 6% with respect to the climatological mean (1999-2020) for subsurface chlorophyll in 2021.*

1. 152 "seems to be connected"… coincide with? With what it is written in the literature?

   This sentence illustrates a first hypothesis about the 2021 anomaly that should be carefully verified and monitored in future works. As replied also to Reviewer#1 and Reviewer#2, we did not enter in more details on this part in the submitted manuscript because of the short length of the paper (the recommended number of figures and words of the paper was indicatively limited to a maximum of 4 figures and 3000 words). However, we recognised that the paper could be improved by providing more details on this particular aspect (i.e., the link of the 2021 oxygen anomaly with the already observed and described incoming of saltier Levantine water). Thus, we added in Appendix B a figure on salinity and temperature through the Otranto Strait:

[Figure]

*Figure B1: Time series of temperature (a) and salinity (b), averaged in the vertical layers 0 - 150 m (red lines) and 150-600 m (black lines) of the Otranto Strait (39.8°N, 18.5° - 19.5° E) in the 1999-2021 time period. In the top panel, light red and dark red indicate data before and after de-seasonalization, respectively. Data are provided by Copernicus physical reanalysis (Prod3, Table 1).*

and we summarized more clearly the results of previously published papers (e.g., Menna et al., 2022b), where the salinity evolution in the southern Adriatic pit in the latest years has been investigated. In particular, we replaced the lines 153-157 of the submitted version by this part (lines 198-211):

*One of the causes of the decrease in total oxygen concentration in the SAdr could be due to the exceptional salinization observed in the SAdr since 2017 (Mihanović et al., 2021, Menna et al., 2022b). This increase was related to the inflow of new, warmer and significantly saltier water masses from the northeastern Ionian Sea (Mihanović et al., 2021, Menna et al. 2022b).*

*The inflow of saltier and warmer water masses is also evident by observing the temporal evolution of these parameters through the Strait of Otranto (Fig. B1). In particular, in the upper layer (0-150 m) both temperature and salinity show an overall positive trend throughout the period 1999-2021, whereas the decrease observed in 2006-2011 and 2017-2018 can be associated with the inflow of less saline AW, triggered by the anticyclonic circulation  of the NIG (Fig. 2f). In the intermediate layer (150-600 m), salinity shows a positive trend in 1999-2021, while no clear trend is observed for temperature. Moreover, a sharp increase in salinity (~ 0.1) is observed in 2019. This increase occurred after the NIG inversion from anticyclonic to cyclonic (Fig. 2f), resulting in a further increase in salinity due to both the decrease in AW advection and the increase in LIW inflow.*

2. 159: remove "latest" (see my comment above)

We did it.

l.160: rephrase

If Reviewer#3 refers to the mention of 2021 year analysis in addition to the interannual variability characterisation, we agree and we added the focus on 2021 in the revised manuscript at line 216 as:

*and the 2021 anomaly with respect to the mean over 1999-2020*

l.161: long-term

We did it.

l.165: that we conducted ?

We did it.

Figures

See my comments above.

We accounted for them in the revised manuscript.

Fig 3: add the 0-value line in all panels (a,c,e,g). In legend replace "spatial" by "vertical", and ",8.4%" by "and 8.4%".

We agree with the changes indicated between quotation marks.

On the other hand, in Fig. 3 we had already indicated the 0-value line in a, c, e, and g panels and maybe it could be a problem of graphical visualisation. Anyway, we provided a new version of Fig. 3 at higher quality in the revised manuscript.

Fig 4: legend: "climatological mean […] in 1999-2020 " could be replace by "Mean over 1999-2020" and "reference period" by "1999-2020 period".

We did it.

Table 1: see my previous comments

We accounted for them in the revised manuscript.

Table 2: EOFs of DO

We did it.

---

## Referee Report (RR1)

**Comment on "Dissolved oxygen as indicator of multiple drivers of the marine ecosystem: the Southern Adriatic Sea case study"**

Anonymous Referee

**General Comments**

The manuscript investigates the interannual variability of dissolved oxygen (DO) over the last 23 years driven by multiple mechanisms (atmospheric, circulatory, and biological processes) in the southern Adriatic Sea. The aim is to demonstrate the importance of DO as indicator of current changes and environmental status.

The study is based on modelled and observational data from the Copernicus Marine Service: the biogeochemical reanalysis for the Mediterranean Sea covering the 1999-2021 period and BGC-Argo float measurements for the 2014-2020 period. DO estimated by BGC-Argo floats are used to bias-correct the modelled DO along the entire reanalysis time series using a quantile mapping technique.

The bias-corrected reanalysis signal is then decomposed using EOF analysis. Then a correlation analysis between the first four EOF temporal modes of variability and key drivers allows evaluating the relative importance of the different drivers to explain DO variability. Finally, year 2021 is compared to the 1999-2020 climatology to identify the main contributor(s) in 2021.

The authors have responded seriously and thoroughly to the first phase of the review. The unclear points have been reworked and clarified. They answered all questions carefully. The study is now complete and detailed.

This is a very interesting study which has multiple interests in the context of actual changes. And DO is a good candidate to detect and monitor changes.